# *Swarm* field-aligned currents during a severe magnetic storm of September 2017

Renata Lukianova[1,2]

[1] Space Research Institute, 117997 Moscow, Russia

[2] Institute of Earth Science, Saint Petersburg State University, 199034 Saint Petersburg, Russia

*Correspondence to*: Renata Lukianova  (renata@aari.ru)

**Abstract.** *Swarm* satellites observations are used to characterize the extreme behavior of large- and small-scale field-aligned currents (FACs) during the severe magnetic storm of September 2017. Evolutions of the current intensities
and the equatorward displacement of FACs are analyzed while the satellites cross the pre-midnight, pre-noon, dusk and dawn sectors on both hemispheres. The equatorward boundaries of FACs mainly follow the dynamics of ring current as monitored in terms of the SYM-H index. The minimum latitude of the FAC boundaries is limited to 50° MLat. The FAC densities are very variable and may increase dramatically, especially on the nightside ionosphere during the storm-time substorms. At the peak of substorm, the average FAC densities reach >3 $\mu$A/m$^2$. The dawn–
dusk asymmetry is manifested in the enhanced dusk-side R2 FACs on both hemispheres. In the 1 Hz dat,a filamentary high-density structures are always observed. In the pre-noon sector, the bipolar structures (7.5 km width FACs of opposite polarities adjacent to each other) dominate, while at the other local times the upward and downward FACs tend to be latitudinally separated. The most intense small-scale FACs, up to ~80 $\mu$A/m$^2$, is observed just in the post-midnight sector. Simultaneous magnetic and plasma perturbations indicate that this structure is likely a current system
of a mesoscale auroral arc.

**Keywords:** Ionosphere-magntosphere interaction, Field-alighned currents, Storms and substorms, Auroral arcs electrodynamics

## 1 Introduction

Field-aligned currents (FACs) provide electrodynamic coupling of the solar wind-magnetosphere-ionosphere system. FACs flow along the high-conducting geomagnetic field lines between different magnetospheric domains and the high-latitude ionosphere. The current system is driven by the internal
magnetospheric circulation of plasma and magnetic field within the global reconnection cycle (Dangey, 1961; Cowley and Lockwood, 1992) and by additional viscous-like interaction at the flanks of magnetosphere (Axford, 1964). Configuration of FACs is primarily controlled by the interplanetary magnetic field (IMF) orientation (Bythrow et al., 1984; Potemra et al, 1984). Other parameters of the solar

wind (velocity, density, IMF strength) and the ionospheric conductivity also play a role (e.g. Christiansen et al., 2002; Ridley 2007; Korth et al., 2002).

Schematic distribution of large-scale FACs has been established by Iijima and Potemra (1976) based on the Triad satellite data. Subsequent space missions allowed constructing comprehensive empirical models of FAC parameterized by the IMF direction and strength, by season, and by hemisphere (Weimer, 2001; Papitashvili et al., 2002; Green at al., 2009). The ionospheric projection of the 3D FAC system consists of a pair of sheets elongated approximately along the magnetic latitude, namely, Region 1 (R1) and Region 2 (R2), with opposite current flow directions in the morning and evening local time sectors and additional current sheets (R0) located on the dayside poleward of R1/R2. R1 FAC flows into the ionosphere (downward current) and from the ionosphere (upward current) on the dawn and dusk side, respectively. R1 currents, if reside on closed field lines of the Earth's magnetic field, are believed to originate in either the boundary layer or in the plasma sheet (Ganushkina et al., 2015). R2 FAC is considered to be a diversion of the partial ring current to the ionosphere driven by pressure gradients in the inner magnetosphere (Cowley, 2000). R0 current is connected to the dayside magnetopause and its polarity strongly depends on the IMF By component. On the Northern Hemisphere, the R0 current flows predominantly out of the ionosphere for positive IMF By and into the ionosphere for negative IMF By (Papitashvili et al., 2002; Lukianova et al., 2012). Additional (NBZ) current associated with the sunward ionospheric flow may appear inside the polar cap, if IMF Bz is northward (Iijima et al., 1984; Vennerstrøm et al., 2002).

While average large-scale (>150 km) current densities typically are of units of $\mu A/m^2$ or less, instantaneous small-scale FACs may reach several hundred $\mu A/m^2$ (Neubert and Christiansen, 2003). The smaller-scale structures are often associated with auroral arcs which are accompanied by ionospheric conductivity and electric field perturbations (Aikio e al., 2002; Juusola et al., 2016). In particular, it was shown that in the evening (morning) sector, there is downward FAC equatorward (poleward) of the arc and upward FAC above the arc. These two FAC regions are connected by a poleward (equatorward) horizontal current. Recent studies also confirmed that the cusp plasma injections are accompanied by pairs of FACs, upward at lower latitude and downward at higher latitude (Marchaudon et al., 2006).

Significant differences in the characteristics of FACs at different scales, especially near noon and midnight have been found (Gjerloev et al., 2011; Luhr et al., 2015; McGranaghan et al., 2017). Under stationary conditions the FAC system is evolved in accordance with the reconnection rate, which is controlled primarily by the solar wind. If a substorm occurs, additional FACs form a current wedge connecting the cross-tail current and the nightside westward ionospheric electrojet (Akasofu, 1964; Lui 1996). The magnitude of existing large-scale FACs also increases (Iijima and Potemra, 1978; Coxon et al., 2014). The dayside R1 currents are found to be stronger than their nightside counterpart during the substorm growth phase, at the same time the R1 region moves equatorward. After expansion phase onset, the nightside R1 currents dominate and their location moves to higher latitudes (Clausen et al., 2013). Recent studies have also suggested that the substorm current wedge could also include a R2 current system (Ritter and Lühr, 2008).

Magnetic storms are characterized by a dramatic enhancement of energy deposition to the Earth's atmosphere. During a magnetic storm, FACs become highly dynamic because of the enhanced solar-wind-magnetosphere interaction, release of energy stored previously in the magnetotail, particle precipitation and ring current build up. Storm-time FACs are stronger and more variable compared to non-storm FACs predicted by the climatological models. Since the intensity and time evolution of FACs vary from storm to storm, it is of interest to analyze their unique characteristics. However, relatively few papers focus on observed storm-time FACs. For example, utilizing the magnetic field measurements by *CHAMP* satellite Wang et al. (2006) investigated the northern and southern hemisphere dayside and nightside FAC characteristics during the extreme October and November 2003 magnetic storms. It was shown that as Dst decreases, the FAC region expand equatorward, with the shift of FACs on the dayside controlled by the southward IMF. For both case studies, on the southern (late spring) hemisphere the minimum latitude of the FACs is limited to 50° magnetic latitude (MLat) for large negative values of Bz (The minima are the same, although in October the IMF Bz drops dawn to -28 nT, while in November it reaches -50 nT.) On the northern (late autumn) hemisphere the equatorward boundaries of the FAC region are located at 55-60° MLat. Using the global maps from the *Iridium* constellation Anderson et al. (2005) studied the FACs intensities during severe magnetic storms which occurred during the solar cycle 23 with a particular attention to the evolution of FACs in the course of the storm of August 2000. The results revealed the dawn–dusk asymmetry of the R1/R2 current sheets, with an increase primarily found on the duskside. It was also shown that under disturbed conditions the total current intensity constrained to be below 20 MA (Anderson and Korth, 2007).

Since 2014, comprehensive studies of FAC distributions were carried out based on high precision observations onboard of *Swarm* constellation (e.g. Dunlop et al., 2015; Juusola et al., 2016; McGranaghan et al., 2017). However, the *Swarm* data have not yet been fully utilized for the storm-time FAC analysis. It is the purpose of this paper to characterize the magnitude and position of the large- and smaller scale FACs as their response to the magnetic storm development. The *Swarm* observations are used in order to identify various characteristics of the storm-time FACs for the event of 6–9 September 2017, which was one of the two most severe magnetic storms of the recent solar cycle 24 (the previous event was the St-Patrick storm on 17 March 2015). The September 2017 event is of particular interest because it was a two-step storm during which two major substorms occurred and the FAC system is affected by the storm-substorm interplay. In this paper we investigate the time evolution of the large scale FAC intensities, the displacement of the FAC equatorward boundaries and the extreme small scale currents.

## 2 *Swarm* satellites orbit

### 2.1 Instrumentation

The ESA *Swarm* mission is a constellation consisting of the three identical satellites (hereafter SwA, SwB and SwC, respectively), all are at the low-altitude polar orbits (Friis-Christensen et al., 2008). The *Swarm* constellation was launched in the end of 2013 and entered the operational phase in April 2014. The initial orbit altitude is 465 km (SwA and SwC) and ~520 km (SwB) and the inclination is 87.5°. By September 2017 the orbit altitude decreases down to ~440 and 505 km, respectively. SwA and SwC fly in a tandem separated by 1-1.4° in longitude and the differential delay in orbit is ~3 s. The orbit period is about 93 min (the speed of the satellites is about 7.5 km/s) and slightly different between SwA/SwC and the upper satellite SwB, so that their along-orbit separation in local time gradually changes. Their orbital planes also gradually drift apart and the separation angle increases by ~20° longitude per year. Slowly drifting in longitude, the orbits cover all the local time sectors over about 130 days.

The mission has a multi-instrument payload. The main module is the high-sensitivity vector (fluxgate type) and scalar magnetometers for determining the magnitude and direction of the total vector and variations of the geomagnetic field with an accuracy of more than 0.5 nT (Merayo et al., 2008). Magnetometers make it possible to carry out measurements in a wide range, including the Earth's main magnetic field and the variations of external magnetic field generated by FACs. FACs are detected by their magnetic perturbations in the orthogonal plane which are obtained after subtracting the main magnetic field model from the total

measured values. From single spacecraft the FAC density can be estimated based on one magnetic component with a techniques invoking Ampere's law under assumptions about the infinite current sheet geometry and the orthogonal crossing of the current sheet. This method was used for the previous one-satellite missions, such as Magsat and Ørsted (Christiansen et al., 2002). It is also applied to each *Swarm* satellite separately. The dual-satellite estimation method calculates current density from curl(B) measured quasi-simultaneously at 4 locations is adapted for SwA and SwC data, where measurements separated along-track are used to create a 'tetrahedron' (Ritter and Lühr, 2006). The curl(B) method provides more reliable current density estimates, as it does not require any assumptions on current geometry and orientation. The FAC output of both a dual-satellite and a single satellite methods are considered to be in a reasonable agreement (Ritter et al., 2013). However, a high degree of coherence is typical at auroral latitudes, while in the polar cap the results based ondual-spacecraft technique as more reliable (Luhr et al., 2016). Both algorithms are implemented to generate the *Swarm* products that are produced automatically by ESA's processing center as soon as all input data are available. The products are provided using the dual-satellite method on the lower pair of satellites SwA and SwC, and the single-satellite solution for each of the Swarm spacecraft individually. The 1 s values (1 Hz sampling rate) of FAC densities are available via the on-line *Swarm* data portal (ftp://swarm-diss.eo.esa.int) as Level 2 data products (Swarm Level 2 Processing System Consortium, 2012). In the present study the single-satellite FACs are used in order to apply the similar method to SwB and SwA/SwC data.

Each satellite is also equipped with the Electric Field Instrument which includes the Langmuir probe to provide measurements of ionospheric plasma parameters: electron density, electron temperature and spacecraft potential (Knudsen et al., 2003). The plasma data are available at 2 Hz sampling rate as the standard product of the *Swarm* data base. Unfortunately, due to technical problems, measurements of the electric field and ions are rather rare. Nevertheless, the combination of data provided by a magnetometer and a plasma analyzer on electrons makes it possible to identify perturbations associated with FACs. In each Level 2 data file the location of the satellite is presented in an geographic coordinate system NEC ($x$ - North, $y$ - East, $z$ - Center), where the $x$ and $y$ components lie in the horizontal plane, pointing northward and eastward, respectively, and $z$ points to the centre of gravity of the Earth. For the purpose of present study all projections of the passes are shown in the magnetic local time (MLT) and magnetic latitude (MLat) domain. For this the coordinates are available via the on-line Swarm Data Visualisation Tool (VirES).

.

**2.2 Orbits on 6-9 September 2017**

The polar projection of the satellite orbits (14-15 trajectories per day) as of September 6-9, 2017 on the northern and southern hemispheres is shown in **Fig. 1**. For mid-September 2017 the passes are centered in the pre-midnight, pre-noon, pre-dusk and pre-dawn sectors. The satellite SwA (orbits of SwC are very similar) enters the region of MLat>50° between ~09 and 12 MLT, and leave this region between ~21 and 23 MLT. The entry (exit) points of the SwB orbit are between ~15 and 17 (02 and 04) MLT. On the southern hemisphere the direction of the tracks in the MLT-MLat framework is opposite. During a day, the successive projections are systematically shifted almost parallel to each other, however, in auroral latitudes, they stay mainly within the same sectors. The MLT ranges covered by the tracks are presented in **Table 1**.

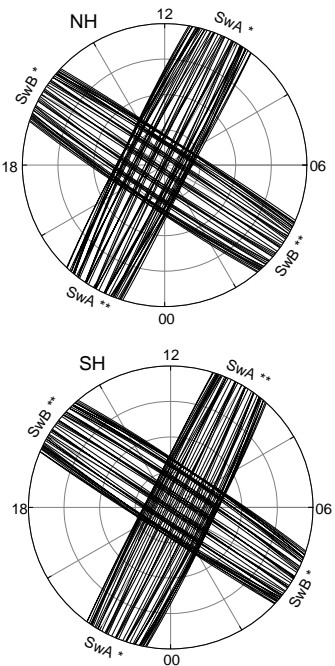

**Figure 1:** Polar maps of the SwA and SwB orbits on the northern and southern hemispheres on 6-9 September 2017 in the MLT-Mlat framework. Circles are drawn every 10° down to 50° MLat. Symbol * and ** indicates, respectively, the entry and exit crossing of the boundary MLat=50°.

**Table 1.** MLT range of the tracks in the northern and southern polar regions

| Satellite | MLT range within which the satellite cross the boundary of 50° (70°) MLat, (hh:mm)* | Center of the MLT range | |
|---|---|---|---|
| | | hh:mm | hh |
| Northern hemisphere | | | |
| SwB | 02:50-04:30  (01:30-05:10) | 03:40 | 04 |
| SwA (SwC) | 09:20-11:30  (08:40-12:50) | 10:30 | 10 |
| SwB | 15:00–16:50  (14:20-18:10) | 16:00 | 16 |
| SwA (SwC) | 21:00–22:50  (19:40-23:30) | 22:00 | 22 |
| Southern hemisphere | | | |
| SwB | 03:10-05:00  (01:50-06:20) | 04:00 | 04 |
| SwA (SwC) | 09:10-11:00  (08:30-12:20) | 10:00 | 10 |
| SwB | 14:50–16:40  (14:10-18:00) | 15:50 | 16 |
| SwA (SwC) | 21:20–23:10  (20:00-23:50) | 22:10 | 22 |

* with accuracy of 10 min

**3 Space weather conditions on 6–9 September 2017**

At the declining phase of solar cycle 24, starting from 6 September 2017, strong multiple solar flares occurred. The associated interplanetary coronal mass ejections collided with Earth's magnetosphere and caused the most intense magnetic storm of the recent solar cycle. The storm produced strong geomagnetic disturbances, ionospheric effects, magnificent auroral displays, elevated hazards to power systems and unstable HF radio wave propagation (e.g. Chertok et al., 2018; Clilverd et al., 2018; Curto et al., 2018;
Yasyukevich et al., 2018).

Evolution of the solar wind (SW) parameters and geomagnetic activity is presented in **Fig. 2** showing (from top to bottom): the IMF Bz and By, the SW proton speed (Vsw) and density (Nsw), the auroral AL and the equatorial SYM-H geomagnetic indices from the OMNI-web service (https://omniweb.gsfc.nasa.gov/).
Two SW shock events impact the magnetosphere. The arrival of the first shock late on 6 September (23:50 UT) results in a sudden increase in all parameters except the AL index. Since at that time the IMF Bz turns northward, the initial disturbance is only weakly geoeffective as a result. At 20:40 UT, 7 September, IMF Bz turns southward that triggers a substorm growth phase and a ring current build up. The second shock

arrived at ~23:40 UT on 7 September, with the SW speed up to 800 km/s and strongly negative Bz and By. This shock causes an abrupt drop of SYM-H down to -150 nT and a spike-like decrease of AL down to -2200 nT. After 03 UT, 8 September, the IMF Bz becomes positive, AL gradually approaches to zero and SYM-H starts to recover until the next southward turn of Bz. At ~06 UT on 8 September another strongly negative Bz period is seen, and the SW speed remains high (>700 km/s). This causes the second substorm (AL is -2000 nT) and ring current intensification (SYM-H is -100 nT). A steady recovery occurs in the AL index throughout 9 September, while the SYM-H gradually increases from -75 to -35 nT. The SW parameters are not available for this day.

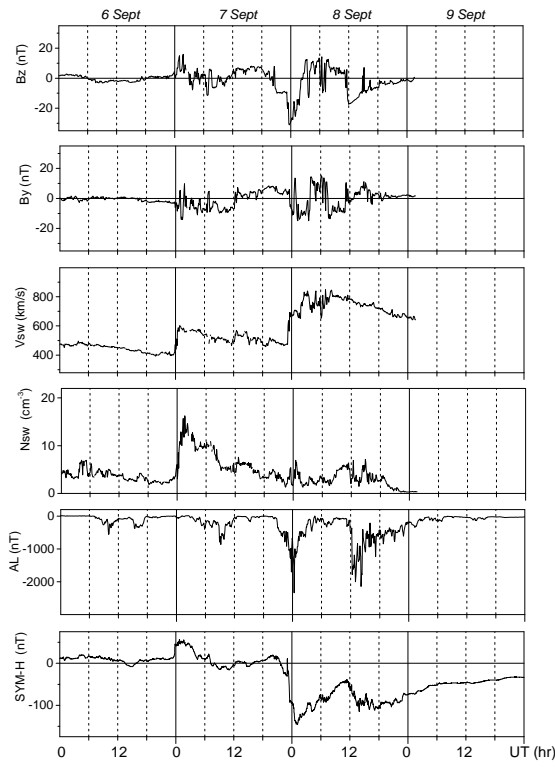

**Figure 2:** From top to bottom: IMF Bz and By, SW speed and density, AL and SYM-H indices on 6-9 September 2017 (5-min values).

## 4 Data analysis

### 4.1 FAC densities

Statistically the large-scale R1 and R2 FAC densities are peaked at the dawn-dusk meridian. In this regard, satellites orbits on September 2017 are not optimal for identifying the R1/R2 extremes, since they are deviated from this meridian. However, the local times of satellite paths are representative enough to assess the evolution of these FACs. On dusk, the orbits of SwB are centered at about 16 MLT that is not far from the region, where the current density is expected to be maximal. On the night side, the orbits are centered at 04 MLT, where they overlap the ionospheric westward electrojet, which is greatly enhanced when a substorm occurs. SwA and SwC cross the pre-noon sector at about 10 MLT, where both the downward R1 and upward R2 are often identified. These satellites also cross the pre-midnight sector, where disturbances associated with substorms are expected.

An example of the FACs measured along the SwB track is shown in **Fig. 3**. The 1 s values presented in **Fig. 3a** provide clear evidence for strong bursts in the auroral latitudes (55-75° MLat), while the near-pole region is almost empty of FACs. The auroral FACs exhibit large-amplitude spike-like structures, thus confirming the existence of filamentary current sheets embedded into the large scale current sheets. The intensities of these small-scale FACs vary from units to tens $\mu A/m^2$. **Fig. 3b** depicts the 51-point smoothed curve. It can be seen that the satellite approaching the pole from the dusk observes first the downward (positive) R2 and then the upward (negative) R1 current, both are of ~1 $\mu A/m^2$ density. Above approximately 70° MLat FACs become marginal. When the satellite moves equatorward in the early morning local times, a structure is observed, in which the poleward currents are positive, so they may be associated with the downward R1 FAC. The most equatorward currents are negative and thus represent the R2 FAC.

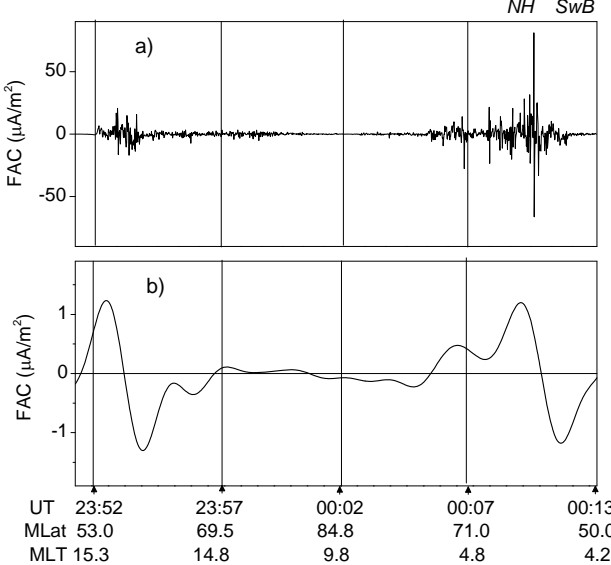

**Figure 3: (a)** 1 s and **(b)** smoothed FACs measured by SwB in the northern polar region between 23:50 UT, 7 September, and 00:13 UT, 8 September. Downward (upward) current is positive (negative).

To demonstrate the global temporal evolution of FACs, in **Fig. 4** the current intensities for the four MLT sectors are presented separately for the northern (**Fig. 4 a, c, e, g**) and southern (**Fig. 4 b, d, f, h**) hemispheres. Each red (blue) point is determined by averaging the 1 s downward (upward) current densities, when the satellite crosses the region filled with FACs. The upper (**a - d**) and lower (**e - h**) plots represent the data from the day side (10 and 16 MLT) and night side (04 and 22 MLT), respectively. For easier visual association of the evolution of FACs with the storm development, the SYM-H and AL indices are added in the plots (**a, b**) representing the day side and in the plots (**e, f**) representing the night side, respectively. During 6-9 September, FACs shown in **Fig. 4,** exhibit three pronounced enhancements, which are of different intensity depending on the MLT sectors. (Note that the FAC densities do not show any systematic changes associated with the orbit ocsilllation during the day.) All FACs start to increase in the very beginning of September 7 in association with the SW dynamic pressure front impinges the magnetosphere causing a positive excursion of SYM-H. The dayside FACs increase abruptly (this is especially well seen in **Fig. 4 b - c**, i.e. at 10 MLT, north, and at 16 MLT, south), while the nightside FACs

(**Fig. 4 e - h**) respond to the shock with a considerable delay. The nightside FACs are peaked in the middle of September 7, when a moderate substorm occurs.

In the very beginning of September 8, in association with the first deep drops of SYM-H and AL, a step-like increase is seen at all MLTs except the prenoon sector. The peak of the day- and nightside FACs reaches 2.5 and 3.5 $\mu A/m^2$, respectively. For a particular crossing the average density exceeds 5-6 $\mu A/m^2$ as seen from the standard deviation. The dayside FACs (**Fig. 4 a - d**) stay enhanced during the whole day of 8 September. The nightside FACs (**Fig. 4 e - h**) more closely follow the evolution of AL, so that the current intensities decrease in accordance with the first storm-time substorm recovery. The next increase in the nightside FACs occurs at ~12 UT on September 8, when the second major substorm occurs and the second drop in SYM-H is observed. On the day side the response of FACs to this substorm is marginal, although the current densities remain elevated throughout the day. All FACs fall to pre-storm levels by September 9.

Comparison of the evolution of FAC intensity with the SW and geomagnetic parameters during the period of 6-9 September reveals that the storm-time FACs are, on average, by several times larger than the quiet-time ones. Better correspondence exists between the nightside FACs (compared to the dayside ones) and the substorm activity as monitored by AL index.

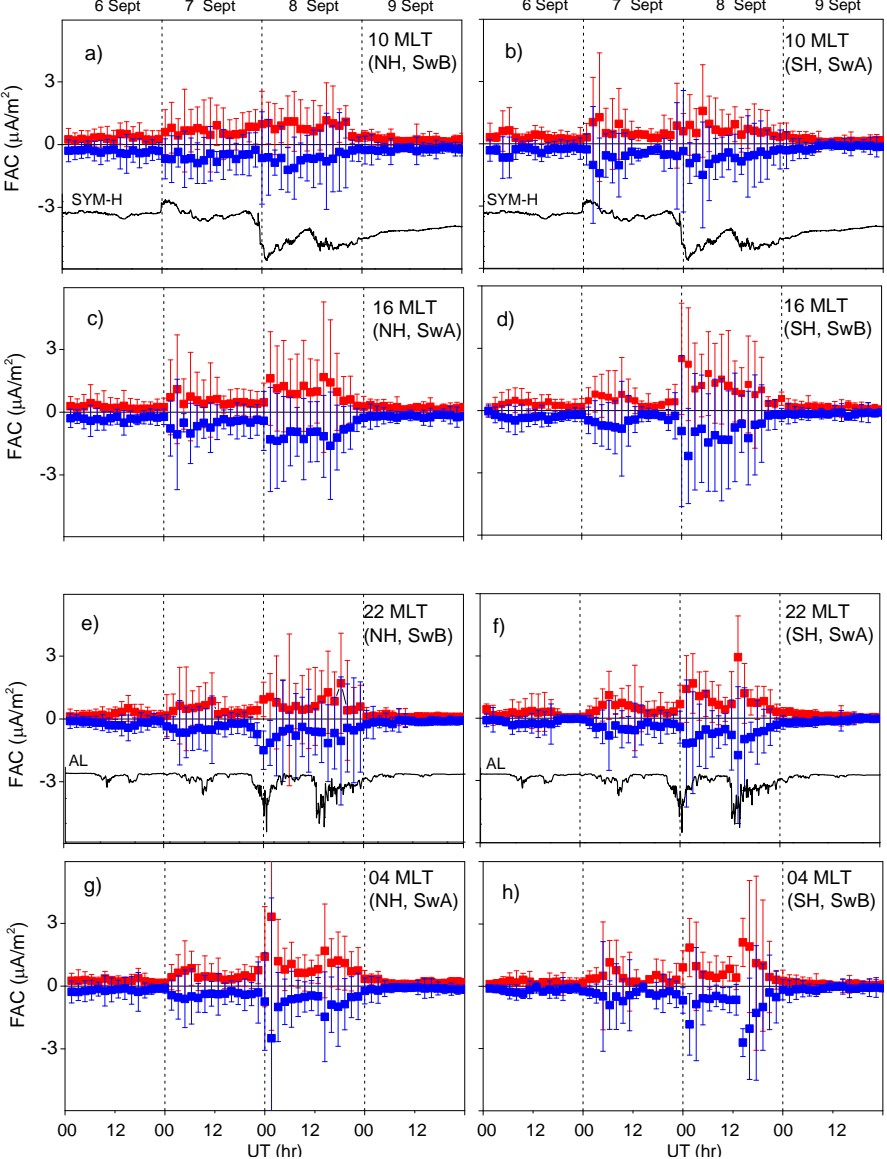

**Figure 4:** Average FAC densities in the four local time sectors covered by the *Swarm* data on 6-9 September, 2017. The left column of plots corresponds to the northern hemisphere (NH) and the right column corresponds to the southern hemisphere (SH). The upper plots (a-d) and the lower plots (e-h) show the dayside and nightside FACs, respectively. The SYM-H and AL indices are added in the plots (a, b) and (e, f), respectively. The centered MLT (10, 16, 22 and 04) is shown in the right upper corner of each plot. The downward and upward FACs (and the corresponding error bars) are shown in red and blue, respectively.

## 4.2 Dawn-dusk asymmetry

During the event of September 2017 a signature of the dawn-dusk asymmetry in the storm-time FACs is revealed. Although the estimate is based on a limited number of crossings and does not allow calculating the total FAC, the current densities summed separately over the dawn and dusk sides may serve as a proxy.

Those parts of the tracks which fall into the 00-12 MLT (12-00 MLT) sector are considered to be related to the dawn (dusk) side, MLat for 50°–90° is accounted. The summed up- and downward FAC densities in each MLT sector as well as those for dawn and dusk sides are presented in **Table 2**. The summation is over the entire 4-day interval.

For any given pass, the net summed FAC density is frequently nonzero. As seen in **Table 2** the net FACs summed in a particular MLT range, is also nonzero. The difference between upward (negative) and downward (positive) current densities varies from 1 to 15%. For the sectors centered at ~04 and 10 MLT this difference is relatively small (-0.7 and +0.1 $\mu A/m^2$ in the Northern hemisphere, and 1.6 $\mu A/m^2$ and -0.4 $\mu A/m^2$ in the Southern hemisphere), i.e the R1 and R2 densities are of comparable values. For the sectors

centered at 16 and 22 MLT the downward (R2) FACs exceed the upward (R1) FACs by 5.2 $\mu A/m^2$ and 3.1 $\mu A/m^2$ in the Northern hemisphere, and by 3 $\mu A/m^2$ and 3.4 $\mu A/m^2$ in the Southern hemisphere.

If the MLT sectors are combined in pairs in order to obtain FACs summed over the dawn and dusk sides, the prevalence of the dusk-side downward current is revealed. From the values presented in the two last

columns of **Table 2** one can see that on both hemispheres the duskside downward current (+60 and +58.9 $\mu A/m^2$ on the Northern and Southern hemisphere, respectively) is stronger than all the other currents. Although the numbers in **Table 2** contain uncertainties related to the lack of global observations, the estimate based on the summed FAC densities from in-situ *Swarm* measurements indicate the existence of the storm-time dawn-dusk asymmetry. Even a limited number of crossings show a clear tendency of the

prevalence of the dusk-side R2. This predominance implies an additional amplification of the storm-time R2 FAC on the dusk side, which is related to the partial ring current. This shift may result from a strong dusk side ion pressure leading to asymmetric dusk-side inflation of the magnetic field consistent with a partial, dusk side, ring current during storm main phase (Liemohn et al., 2001; Anderson and Korth, 2007).

**Table 2**. Summed upward (negative) and downward (positive) FAC densities in for all passes on 6-9 September. In the last two columns the conventional FAC regions (R1 or R2) are indicated in brackets. The largest values are shown in bold.

| Side | MLT range (as at 50° MLat) * | FAC densities (μA/m²) | | | |
|---|---|---|---|---|---|
| | | up | down | up | down |
| Northern Hemisphere | | | | | |
| dawn | 09:20 – 11:30 | -23.3 | +23.4 | -54.6  (R2) | +53  (R1) |
| dawn | 02:50 – 04:30 | -31.3 | +29.6 | | |
| dusk | 21:00 – 22:50 | -27 | +30.1 | -52.7  (R1) | **+60**  (R2) |
| dusk | 15:00 – 16:50 | -24.7 | +29.9 | | |
| Southern Hemisphere | | | | | |
| dawn | 09:10 – 11:00 | -27.8 | +27.4 | -51.5  (R2) | +52.7  (R1) |
| dawn | 03:10 – 05:00 | -23.7 | +25.3 | | |
| dusk | 21.20 – 23.10 | -23.8 | +27.2 | -52.5  (R1) | **+58.9**  (R2) |
| dusk | 14:50 – 16:40 | -28.7 | +31.7 | | |

* with accuracy of 0.5 hr

## 4.3 Dynamics of the equatorward boundary of the FAC region

It is well established that the enhanced SW input and the pile-up of open magnetic flux during a geomagnetic storm results in the equatorward expansions of the polar cap and the auroral oval as a whole
(e.g. Milan et al., 2004). Following the magnetospheric dynamics FACs also move equatorward. **Fig. 5** shows the evolution of the equatorward boundary (EqB) of FACs on 6-9 September. For the comparison the SYM-H and AL indices are added. The EqB parameter is determined as the lowest MLat at which FACs are terminated. The procedure of the 20-point sliding window (the scale is about 150 km) moving along a track from the equator to the pole is applied to the 1 s FAC values and the corresponding MLats.
EqB is selected as the lowest MLat of the window if 90% of FAC values within the window exceed |0.1| μA/m². Then the results are checked visually in order to avoid the erroneously calculated latitudes, that may happen, e.g., if a significant latitudinal gap between R1 and R2 occur. When calculating EqB, no separation between the up- and downward FACs is made.

Even visual comparison of the SYM-H and EqB evolutions in **Fig. 5** reveals generally coherent behavior of these two parameters. In particular, during a period preceding the storm main phase (before 8 September, when SYM-H is mainly positive) EqB is located much lower than during the end of recovery phase (after ~12 UT on 9 September, when SYM-H is still negative). Before the SYM-H attains the negative values

below -20 nT at 22:00 on 7 September, FACs are observed mainly poleward of 60° MLat on both hemispheres. Moderate equatorward shifts of EqB are associated with the modest substorms occurred before the storm main phase in the middle of 6 and 7 September. Prior the main phase, on both hemispheres the prenoon (10 MLT) EqB is found considerably poleward compared to the EqB location at other MLTs. The effect is well seen during the two time intervals: from ~22 UT, September 6 till 06 UT,

September 7 and at 12-24 UT, September 7. Both intervals are dominated by the northward IMF (sf. **Fig. 2**), so that a shrinking of the polar cap and a poleward shift of the auroral oval is expected. With regard to the position of FACs, the displacement of its equatorward boundary is the largest only in the pre-noon sector, while the other local times remain less affected.

Upon arrival of the SW shock at the very end of September 7, EqB is abruptly shifted equatorward, then tends to recover until the middle of September 8, and then drops again following the second intensification of the storm. At the very beginning of 8 September EqB is found at its lowest position at 50° MLat. A drop of EqB occurs simultaneously with the peak of the first substorm intensification and the lowest SYM-H (-160 nT). The second substorm reaches its peak slightly before the second minimum of SYM-H (at 12:50

and 13:55, respectively). During this second activation the EqB is shifted again as low as 50° MLat (although SYM-H is only -100 nT). As seen in **Fig. 5**, the evolution of EqB tends to follow the gradual change of SYM-H rather than abrupt drops of AL related to the substorm activations (see also **Fig. 2** for AL). Unlike the current density, which is enhanced throughout the storm and exhibits several spike-like increases in accordance with AL, the temporal variations of EqB are relatively smooth. Relatively small

difference in evolution on the day- and nightside EqBs is observed. At the peaks of the storm, EqB is at about 50° MLat, while during the late recovery phase, EqB is shifted poleward as high as 70° MLat. Possible expansion of the FAC region during the substorm growth phase, and then its contraction after onset are difficult to resolve with the *Swarm* data.

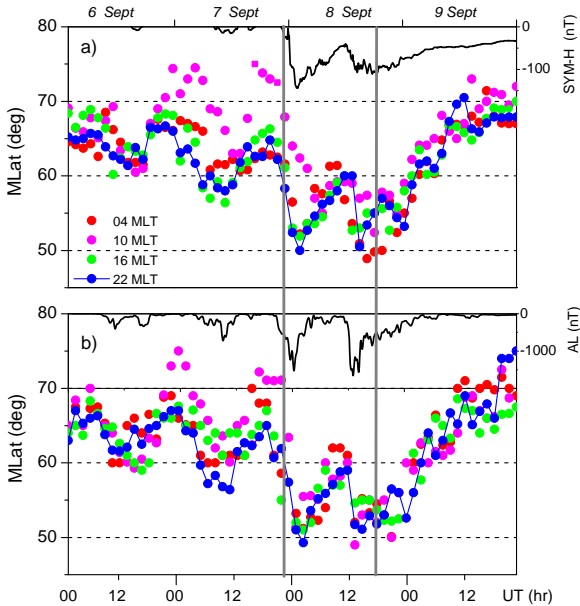

**Figure 5**: MLat of the FAC equatorward boundaries (EqB) on the Northern (a) and Southern (b) hemispheres for the sectors centered at around 04, 10, 16 and 22 MLT. EqB for each sector is shown by dots of different colors; blue dots representing the nightside (~22 MLT) EqB are connected by a line. The SYM-H and AL index (black line) is added to the upper and lower plot, respectively. The vertical lines mark the beginning of the main and recovery phases.

The equatorward displacement of FACs roughly correlates with the storm intensity as monitored by the SYM-H index, while the storm-time subsorms can modify this relationship. In **Fig. 6,** separately for the main and recovery phases, the correlations between SYM-H and the nightside EqB are shown. Data from both the northern and southern hemispheres are included. The correlation coefficients for the main and recovery phases are very similar (cc=0.88 and 0.87), while the corresponding regression equations are considerably different. During the storm main phase, the equatorward expansion of EqB is governed by the equation MLat=63.1+0.1·SYMH. When the recovery phase begins, the poleward shift of EqB is described by the expression MLat=79.5+0.3·SYMH. The faster poleward recovery of EqB comparing with its equatorward expansion is due to the fast decrease in substorm activity on September 9 .

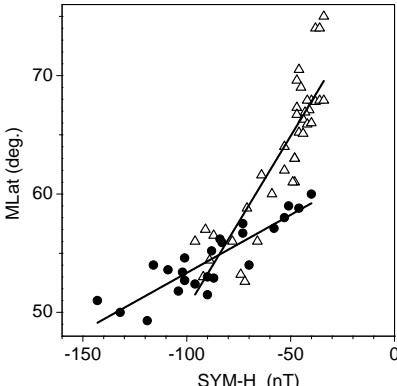

**Figure 6**: Correlations between the SYM-H index and the latitudinal position of the nightside (~22 MLT) EqB: black dots and open triangles correspond to the main and recovery phase, respectively.

### 4.4 Small-scale FACs

It is known that FACs appear on a wide range of scales from large-scale sheet-like currents of hundreds kilometers width to very small-scale filamentary currents of hundreds meters width. The quasi-instantaneous amplitudes of the small-scale component are often much larger than the stationary R1/R2

FACs. The current intensity vary inversely with scale so that large-scale currents are typically a few $\mu A/m^2$, whereas the smaller scale (down to 10 km) are a few tens $\mu A/m^2$ (Neubert and Christiansen, 2003; Luhr et al., 2015; McGranaghan et al., 2017). To obtain the time-series of the *Swarm* peak current densities on 6-9 September 2017, the largest positive and negative 1 s values were selected from each crossing in a given MLT time sector irrespective of the hemisphere. The obtained peak values are presented in **Fig. 7**. First of

all, from this figure one can see that the small-scale peaks may be more than an order of magnitude larger than the FACs averaged over a track (sf. **Fig. 4**). On September 6, only two outliers of about +20 $\mu A/m^2$ and -30 $\mu A/m^2$ are observed. Both are from the pre-midnight sector and are associated with a moderate substorm occurred in the middle of this day. During the disturbed period, starting with the compression of the magnetosphere on September 7, the amplitude of peaks tends to increase. Two intense substorms

occurring during the storm main phase cause an additional strengthening of small-scale FACs at all MLTs. At ~00 UT on September 8, the up- and downward currents located in early morning local times attain their extremes of 70-80 $\mu A/m^2$. The second major substorm occurred in the middle of September 8 is also

accompanied by the peaks, which are more pronounced in the dusk-side, where the upward FAC reaches about -50 μA/m². Note that some peaks can be missed due to the temporal and spatial gaps between the satellite tracks.

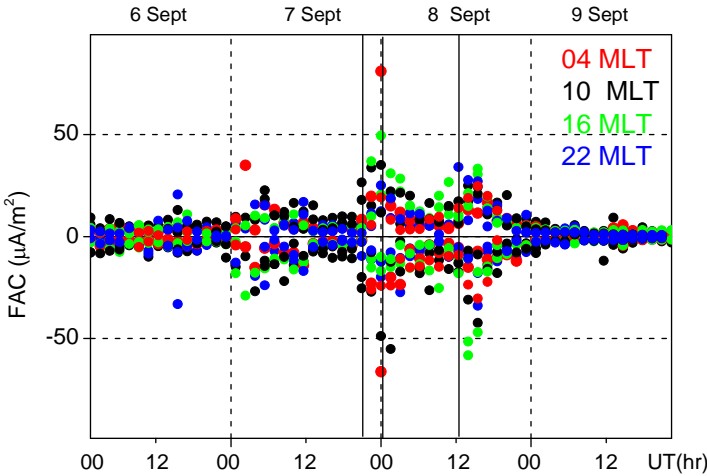

**Figure 7**: The largest downward (positive) and upward (negative) 1 s current densities for four MLT sectors on 6-9 September. The vertical solid lines mark the beginning of the storm main phase at 22:00 on 7 September (the time when SYM-H attains its stable negative values <-20 nT; the period of SYM-H<-20 lasts till the end of September 9), the peaks of the first and second major substoms (the time when AL attains its minimum).

When for each crossing within a certain MLT sector, the minimum (i.e. peak upward current) and the maximum (i.e. peak downward current) 1 s FACs are selected, it appears that in some cases these peaks are observed at very close latitudes, while in other cases the minimum and maximum are spaced in latitude. In **Fig. 8**, the correlations between the MLats, at which the most intense small-scale FACs of opposite polarities are observed, are presented for each MLT sectors. The x-axis (y-axis) corresponds to the MLat of the downward (upward) peak selected in each crossing. The magnitude of minima and maxima are not accounted. From **Fig. 8** one can see that correlation between the latitudinal positions of the up- and downward peaks varies with MLT. The highest correlation coefficient (cc=0.94) is found in the pre-noon sector (**Fig. 8b**). This is indicative of a large population of the paired, closely adjacent small-scale currents of opposite polarity (called hereafter the bipolar structure). In the dusk (**Fig. 8a**) the correlation coefficient

decreases down to 0.78. Almost the same correlation (cc=0.75) is observed in the pre-midnight sector (**Fig. 8c**). At the early morning hours (**Fig. 8d**) the correlation is much weaker (cc=0.53) implying that the extreme up- and downward currents appear less frequently in pair but rather are spatially (or temporary) separated. Different mechanisms of the small-scale FAC formation on the day- and night side can be the cause of this spatial distribution and variability.

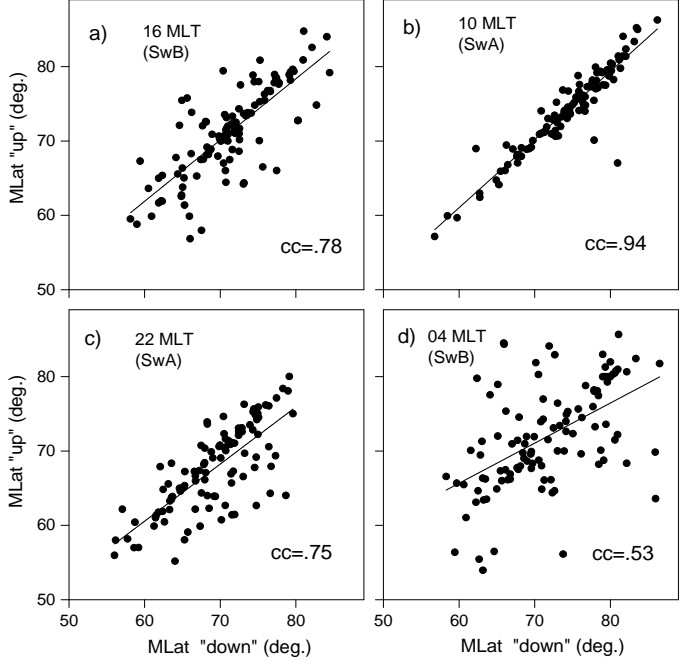

**Figure 8**: Correlations between magnetic latitudes, at which the up- and downward peak FACs are observed: (a) dusk, 16 MLT;  (b) pre-noon, 10 MLT; (c) pre-midnight, 22 MLT; (d) post-midnight/early morning, 04 MLT.

## 4.5 Small-scale FACs of extreme amplitudes

During the storm under consideration a pair of the most intense upward and downward small-scale FACs is revealed by SwB at around 00:10 UT on September 8, when the satellite traverses the auroral latitudes from north to south over the geographic area of the Barents Sea, about 20 degree magnetic longitude to the East

from the IMAGE magnetometer network (http://space.fmi.fi/image).  The network produces the IL index, which is simple estimate of the total westward currents crossing the IMAGE chain. The IL index (**Fig. 9**) shows that the extreme FACs are observed during the first period of the storm-time substorm intensifications, several minutes  before the IL drops from -1500 nT to -3700 nT.

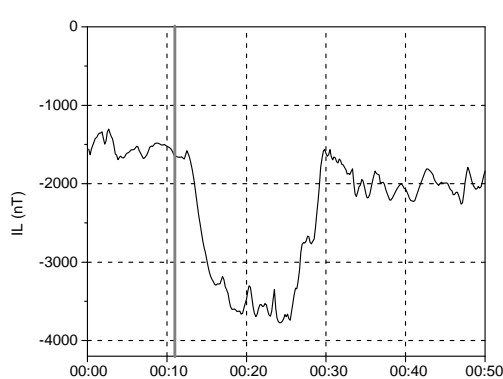

**Figure 9**: The 10 s IL index at 00:00-00:50 UT, September 8. Time of the extreme FAC observation is shown by grey line.

10 The 1 s FACs and plasma parameters (the electron density, Ne, temperature, Te, and the spacecraft electric potential, $U_{sc}$) measured by SwB at 00:08-00:12 UT on 8 September are shown in **Fig. 10**. As show in **Fig. 10a,** at 00:10:18-00:10:19 UT the satellite observes the bipolar current structure of extreme density consisting of the poleward downward (81 $\mu A/m^2$) and equatoward upward (-66 $\mu A/m^2$) FACs. The paired up- and downward FACs are of relatively comparable values, thus they are balanced and likely closed 15 locally. In **Fig. 10a** the original 1 s values are superimposed to the smoothed curve, which reveals that the bipolar structure is located at the edge of the mesoscale downward FACs.

The bipolar current structure is accompanied by plasma perturbations. A narrow peak in Ne up to $77 \cdot 10^3$ $cm^{-3}$ (**Fig. 10b**) and an increase of Te up to $\sim 10^4$ K on average (**Fig. 10c**), that is $\sim$50% above their ambient 20 values, are observed almost simultaneously with a pair of extreme FACs. (It should be noted that the Te values presented here are based on the current processing of the satellite data and may be still uncalibrated. However, it hardly affects the relatively small-scale perturbations.) The elevated Te is observed in a wider region slightly poleward of the enhanced Ne. The plasma disturbances are clearly seen in $U_{sc}$, which is

proportional to $-k \cdot Te$ ($k$ is the Boltzmann constant). Note that the level of noise for the $U_{sc}$ channel is much lower compared to that for the Te channel (0.4% and 2% for $U_{sc}$ and Te, respectively). **Fig. 10d** shows that a reduction of $U_{sc}$ starts at 00:09:56 UT, then peaks at 00:10:08 (-12 V) and 00:10:20 UT (-8 V), the average decrease is -5 V. The region, where the Te and $U_{sc}$ are perturbed is several times wider than region occupied by the pair of extreme FACs..

If the localized increase in Ne indicates conductance enhancement (likely due to precipitating electrons), the observed plasma and current perturbations are similar to those associated with auroral arcs (Opgenoorth et al. 1990; Lyons, 1992; Lewis et al., 1994; Johnson et al., 1998; Aikio et al., 1993; Juusola, et al., 2016). In particular, Aikio et al. (2002) studied the current system of arcs in the evening sector, where the background electric field is northward. It was shown that for arcs located within the northward convection electric field currents flow downward on the equatorward side of the arcs, then poleward, and upward from the arcs. The arcs are associated with an enhanced northward-directed electric field region on the equatorward side of the arc. An enhancement in the electric field starts already several tens km equatorward of the arc edge.

During the storm under consideration the bipolar FAC pattern observed at 00:10 UT is located in the morning sector, where the background electric field is expected to be southward. This is confirmed by the SuperDARN-based convection model (http://vt.superdarn.org/tiki-index.php?page=ASCIIData), which predicts in the region of the SwB observations the magnitude of the southward and westward component of about 6.5 mV/m and 0.5 mV/m, respectively. As mentioned in Section 2.1, unfortunately the in-situ *Swarm* electric field is unavailable. Only the reported characteristics of the electric field associated with arcs can be used for qualitative analysis. In particular, for morning side arcs an enhanced southward electric field on the poleward side of the arc is expected. In this case the current pattern consists of a downward FAC on the poleward side of the arc connected to an upward current above the arc by an equatorward ionospheric closure current. This is exactly what is seen in **Fig. 10a**: when SwB flies away the pole, it first observes a positive spike (downward FAC) and then a negative spike (upward FAC). Since the width of the region of enhanced Ne is ~30 km, the arc is relatively narrow. Comparing **Fig. 10a** and **Fig. 10b** one can see that the paired FACs is located on the poleward side of the region of enhanced Ne. Note, that in **Fig. 10b** a sharp increase in Ne up to ~80 $\cdot 10^3$ cm$^{-3}$ is preceded by a weaker spike-like drop down to ~30 $\cdot 10^3$ cm$^{-3}$. A

decrease in Ne (which is usually much less pronounced than an increase due to precipitating electrons) is associated with a downward FAC observed at the opposite boundary of the arc. Elevations of Te may be created by electric fields which can arise within narrow region adjacent to the northern side of the auroral arc as observed by Aikio et al. (2002).

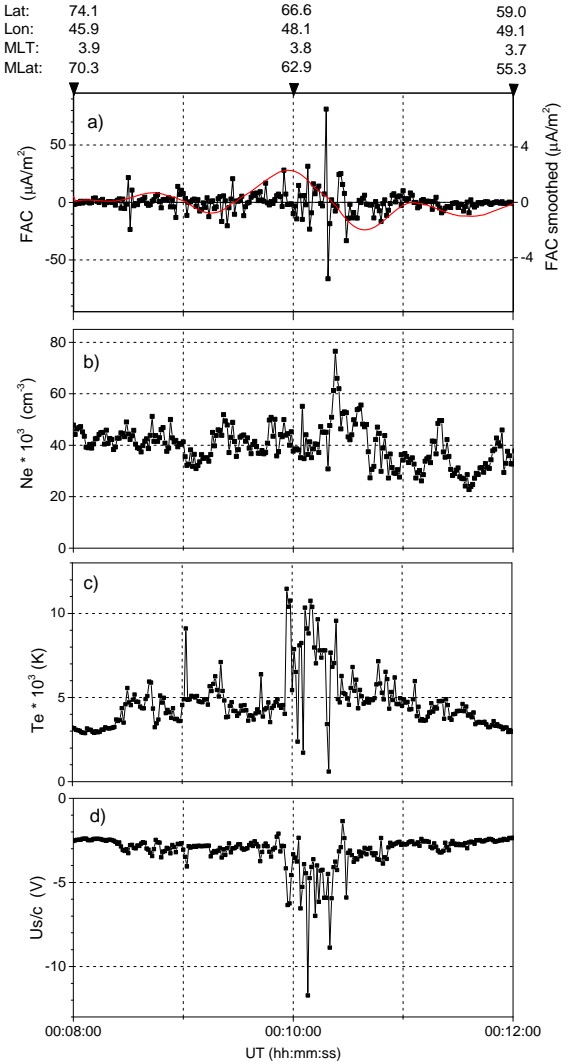

**Figure 10**: The 1 s values of (a) FAC density, (b) Ne, (c) Te, and (d) Us/c measured along the SwB track at 00:08:00 – 00:12:00 UT, 8 September. In the upper plot the 21-point smoothed FAC density is also shown. Geographic and geomagnetic coordinates are shown on the top.

## 5 Discussion

Observations of the LEO *Swarm* multi-satellite mission are used in order to identify various characteristics of the storm-time FACs for the severe event of 6–9 September 2017. During the storm main phase two major substorms occurred, so that the FAC system evolved under conditions of the storm-substorm interplay. In mid-September 2017 the separation between the upper and lower *Swarm* satellites was about 6 hours in local time. Within the sectors centered at 04, 10, 16 and 22 MLT the northern and southern polar regions were covered by about 60 tracks along which the 1 Hz measurements of FACs were carried out. These observations made it possible to reveal the evolution of the large scale FAC intensities, the displacement of the FAC equatorward boundaries and the some features of the extreme small-scale FACs.

### 5.1 Large-scale characteristics of FACs

Evolution of large-scale characteristics of FACs during the September 2017 storm is in general agreement with regularities observed previously by *CHAMP* during the intense 2003 geomagnetic storms (Wang et al., 2006). The common feature of all storm-times is the equatorward motion of FACs generally correlating with the storm intensity. During the September 2017 storm the global coverage of the high latitudes by the precise measurement onboard of *Swarm* satellites made it possible to reveal that the FACs was enhanced at all MLTs starting from the time of the first SW shock arrival at the very beginning of 7 September, although the northward IMF and the prolonged period of geomagnetic quietness lasted almost a day. After this quietness a storm abruptly commenced at ~22: UT on 7 September. During the two-step main phase FACs exhibit three pronounced enhancements, the evolution of FACs depends on the MLT sectors. On the dayside FACs strengthen after the sudden commencement and in response of the first drop of SYM-H, while the response to the second drop of SYM-H is relatively weak. On the night side the current intensities follow mainly the substorm dynamics as monitored in terms of the AL index, promptly respond to the onset of storm-time substorms and strengthen at the peak of substorm. At the same time, during the period between the major substorms, when AL is fully recovered, but SYM-H is not, FACs stay considerably enhanced.

The September 2017 storm is characteristics of a considerable equatorward expansion of the FACs region as low as 50° MLat on both hemispheres. The latitudinal displacement of FACs is more gradual and smooth than the changes in current intensity. For comparison, during the 2003 storms the minimum latitude of peak current density are limited to 52-56° MLat (Wang et al., 2006). It should be noted that these authors defined the latitudinal positions of peak current density but not the most equatoward boundary of the FAC region, thus the actual FAC region may expand to lower latitudes. The lowest latitudinal position of the storm-time FACs was found by Fujii (1992). For the storm of March 1989 the equatorward boundary of the FAC system reached as low as 48° MLat. Similar to the 2003 storms, in 2017 the latitudinal positions of EqB generally follow the SYM-H variations. FACs are shifted further equatorward during the storm-time substorms. Even a relatively minor substorm occurred prior the storm causes a considerable equatorward displacement of FACs. The lowest latitude of EqB is observed when both the SYM-H and AL indices reach their minimums.

Although the storm of September 2017 is considerably weaker (Dst≈-100 nT) than the storms occurred in 1989 (Dst≈-600 nT) and 2003 (Dst≈-400 nT), the FAC region expands approximately to the same latitudes. This effect may be interpreted in terms of saturation, when the FAC region does not expand lower than ~50° MLat independently of the storm severity. Linear dependence between latitudinal boundaries of the FAC sheets upon the dayside merging electric field, the AE and Dst indices has been reported by Xiong et al. (2014). It was also pointed out that toward high activity a saturation of equatorwards expansion seems to set in.

In September 2017, prior the storm main phase, when the IMF Bz is northward, the pre-noon EqB is located at higher latitudes (~75° MLat) compared to the other MLT sectors (~65° MLat). Surprisingly, in the course of the storm main phase, no considerable difference between the latitudinal positions of EqB in different MLT sectors is found. After ~12 UT on September 9, in the late recovery phase (SYM-H is -50 nT), both the day- and nightside EqB recover to their undisturbed position (about 70° MLat). The coherent behavior of EqB is rather unexpected because Wang et al (2006) found that the poleward recovery of FACs on the nightside is slower than on the dayside. Previous analysis of the latitudinal shift of the polar cap boundaries based on the *IMAGE* observations during a magnetic storm has also shown that, if the IMF Bz turns northward, the dayside boundary recovers much faster than the nightside boundary (Lukianova and

Kozlovsky, 2013). This is because dynamics of the nightside boundary depends on the energy accumulated in the magnetotail during the previous period of the storm main phase. However, it seems that the storm of September 2017 does not show the same regularity. The reason may be that during the storm main phase the two major substorms occurred, so that the energy stored in the tail was released faster. Comparing the evolution of the FAC densities and the equatorial boundary positions during the storm recovery, one can see that the densities decay much faster than the boundaries return to their quiet time positions.

High FAC intensity is associated with the auroral oval. Previous studies based on particle precipitation and optical observations have shown that the oval radius increases when the ring current is intensified during magnetic storms (e.g., Meng, 1982; Yokoyama et al., 1998). Significant variations in the location of the aurora take place during the substorm cycle. Substorms occurring on expanded auroral ovals during magnetic storms are most intense, since they close the most magnetospheric open magnetic flux and the presence of the enhanced ring current increases the open flux threshold at which substorm onset is favoured (Milan et al., 2009). It was also shown that changes in oval radius associated with dayside and substorm driving occur on timescales of minutes and hours, while changes associated with the ring current are more protracted as the ring current dissipates slowly (Milan, 2009).

The *Swarm* observations, although they are instantaneous, reveals a tendency of the dawn-dusk asymmetry FACs. The dawn-dusk asymmetry is revealed by comparing the up- and downward FACs, which are summed for all crossings over dusk and dawn separately. While the summed FAC intensities are comparable between the two hemispheres, the positive and negative densities on the dusk and dawn are slightly imbalanced and the net current is nonzero. It seems that the dusk-side downward (R2) FACs are larger than the dusk-side upward (R1) and the dawn-side R1 and R2 currents. The observed imbalance in FACs is likely related to an intensification of partial ring current, which is connected to R2 FAC on the dusk. Strengthening of partial ring current may also lead to asymmetric dusk-side inflation of the geomagnetic field lines. The dawn-dusk asymmetry in strength and the equatorward displacement of R1 and R2 at the peak of the major storm on August, 2000, has been reported by Anderson and Korth (2007). This study was utilized the global distributions of FACs generated at a 10 min cadence separately for the Northern and Southern Hemisphere by the AMPERE project which is based on the fleet of Iridium satellites. Although the *Swarm* observations unable to provide the instantaneous global FAC distribution, the responses of FACs in certain

MLT sectors on the dawn side are different from those on the dusk side. Note that the results in Table 2 are calculated by using the 1 Hz FAC values and their averages do not necessary represent the large-scale R1/R2 FACs. Nevertheless, for the storm of September 2017, the dawn–dusk asymmetry is manifested in the enhanced average density of the downward FACs on the dusk side. This feature is consistent with the global observations by AMPERE, from which the asymmetry of large-scale FACs can be identified. At the same time, almost no difference in the equatorward shift of the dusk and dawn side FACs is observed by *Swarm*.

## 5.2 Small-scale FACs

Due to their large amplitudes small-scale FACs play an important role for the energy input to the upper atmosphere. In several previous studies, the FACs associated with arcs were estimated as 1-10 $\mu A/m^2$ (Bythrow and Potemra, 1987; Elphic et al., 1998; Janhunen et al., 2000; Luhr et al., 2016). Larger range of current densities, varying between 4 and >40 $\mu A/m^2$, has been observed (Aikio et al., 2002) and even more intense small-scale FACs, up to hundred of $\mu A/m^2$, at the edges of arcs have been measured by MEO satellites (Marklund et al. 1982; Bythrow et al. 1984). Such a large range of the FACs estimates is likely related to its different scales (and different techniques), because for arcs with very sharp electron density gradients, the FACs associated with ionospheric currents flow in narrow regions at arc edges. If the real widths are smaller, the current densities are expected to be larger.

Filamentary structures of high densities are always presented in the *Swarm* observations. The narrow high-density currents are averaged out when integrated over a FAC region, so that multilayer structures of steady large-scale FACs of the R1/R2 type depicted by Iijima and Potemra (1978) can be revealed after a proper smoothing. From a statistical study of the temporal and spatial-scale characteristics of different FAC types derived with the *Swarm* satellites Luhr et al. (2015) have shown that small-scale, up to some 10 km FACs are carried predominantly by kinetic Alfven waves. A persistent period of small-scale FACs of order 10 s, while large-scale FACs can be regarded stationary for more than 1 min. Neubert and Christiansen (2003) studied the morphology of very small-scale FACs from a survey of *Oersted* satellite 25 Hz data. These FACs are distributed in a broad region around the pre-noon and cusp region, and in the pre-midnight sector. It was found that at the considered time scale, instantaneous currents may reach the largest values up to 1000 $\mu A/m^2$, while the average current densities reach a maximum of 10 $\mu A/m^2$. McGranaghan et al.

(2017) demonstrated a local time dependence in the relationships between large (>250 km) and small FAC scales (10–150 km width, density is up to 0.5 $\mu A/m^2$). It was found that linear relationships exist near dawn and dusk local times, while at noon and midnight local times no similar regularity is seen. The results are based at all available data from the *Swarm* satellites and the AMPERE irrespective of the level of geomagnetic activity.

During the September 2017 storm one of the *Swarm* satellites managed to observe a pair of the most intense small-scale 7.5 km width FACs of opposite polarity, the magnitude of which are approximately +80 and -70 $\mu A/m^2$. These up- and downward FACs are adjusted to each other and separated in a fraction of degree in MLat. The bipolar FAC structure occurs in the region approximately between R1 and R2, just prior of the abrupt substorm intensification in the vicinity of the newly developed ionospheric WEJ. The polarity reversal captured by the *Swarm* data for two consecutive seconds implies a quite localized current closure through the ionosphere mostly via Pedersen horizontal currents. Although without optical and electric field data one could not make a strict conclusion, the small-scale bipolar FAC pattern accompanied by a localized enhancements in Ne and Te are likely associated with mesoscale discrete aurora. One-to-one correspondence of small-scale FACs with localized electron precipitation events has been previously observed (e.g. Fukunishi et. al., 1991). The SwB observations are in agreement with the disturbances expected for the acrs occurred on the morning side, where the ambient electric field is southward. The observed features are resemble to those reported by Kozlovsky et al (2007) and Aikio et al. (2002) but bearing in mind that the latter are related to the evening sector, where the background electric field is northward. Based on *Swarm*/*THEMIS* ASI observations Wu et al. (2017) has associated multiple auroral arcs with up/down current pairs. For these arcs unipolar and multipolar FAC systems with current densities of about a few $\mu A/m^2$ have been observed. Arcs in unipolar FAC systems have a typical width of 10–20 km and a spacing of 25–50 km. Arcs in multipolar systems are wider and more separated. In the bipolar structure of extreme intensity observed by SwB in September 8, the current density exceeds the values observed by Wu et al. (2017) at least by a factor of ten, while the spatial extend of FACs is smaller. This difference implies to the existence of sharp electron density gradients at arc edges. Usually, the arcs consist of auroral rays and bright spots moving along the arcs and these spatial irregularities may produce the extreme small-scale FACs. This study has shown that under disturbed conditions, FACs forming the arc current system may reach hundred of $\mu A/m^2$ on the spatial scale of less than ten kilometers.

Statistically, the bipolar structures dominate in the pre-noon. In the post-midnight MLTs they are observed less frequently. While the interpretation of the bipolar structure in the terms of the meso-scale arc pattern seems reasonable, the small-scale FACs are often a result of reconnection processes distributed over the dayside magnetopause and even in the tail for negative Bz. In contrast to the post-midnight, in the pre-noon sector, where cusp/cleft currents are expected, the bipolar structures are quite frequent. This may be a signature of the plasma injections which are accompanied by pairs of FACs generated due to flux transfer event (FTE) formation (Southwood, 1987) or multiple reconnection at the magnetopause. Magnetic topologies associated with FTEs were previously observed by the MEO satellites (Marchaudon et al., 2004; 2006; Pu et al., 2013). The small-scale field-aligned currents are possibly a consequence of turbulence and instabilities associated with the process of opening previously closed magnetospheric field lines and merging them with the interplanetary magnetic field (Watermann et al., 2009). The regularity presented in **Fig. 8** shows that during the September 2017 magnetic storm the bipolar structures dominate exactly in the region where the signatures of FTEs and the reconnection lines formed at the magnetopause are expected. At the same time, a pair of the most intense FACs is observed on the night side.

## 6 Conclusion

Characteristics of FACs inferred from the 1 Hz *Swarm* observations during the severe magnetic storm of 6-9 September 2017 are presented. This storm is the two-step one with about 22-hr preliminary phase and the intense substorms occurred in the course of the storm main phase. The satellites cross the pre-midnight, pre-noon, pre-dusk and pre-midnight sectors. The following features of the storm-time FACs are found.

Evolution of the current intensities and the latitudinal position of the equatorward boundaries of the FAC region are mainly controlled by a storm-substorm interplay. The FACs become enhanced starting from the SW shock arrival despite of the prolonged period of the northward IMF. The night-time FAC densities primarily follow the substorm development while the dayside FACs are intensified in response to the SW shock and then stay enhanced. At the peak of substorm, the FAC densities averaged over a track within a given MLT sector, reach 3 $\mu A/m^2$, while the undisturbed level is about 0.2 $\mu A/m^2$.

The equatorward displacement of FAC sheets correlates with the storm intensity as monitored by the SYM-H index. The correlation coefficients for the main and recovery phases are about 0.9, while in the course of the main phase the rate of equatorward expansion of FACs is slower than their poleward displacement during the recovery phase. This is likely due to the relatively fast decrease in substorm activity. The minimum latitude of the equatorward FAC boundaries is limited to 49-50° MLat. Although the storm of September 2017 is relatively weak (Dst is about -100 nT), the FAC region expands approximately to the same latitudes as those observed for the much severe storms.

The filamentary structures of high-density FACs are always presented in the *Swarm* observations. A bipolar structure (i.e. the adjacent upward and downward small-scale FACs), ~80 $\mu A/m^2$, 7.5 km width, is observed in the vicinity of the newly developed westward electrojet just prior the substorm onset. Simultaneous plasma perturbations indicate that the FAC pattern is likely associated with mesoscale auroral arc.

*Data availability*: The data used for the publication of this research are freely available from the *Swarm* Science Team web site (ftp://swarm-diss.eo.esa.int). Data selected for the analysis are available upon request (RL).

*Competing interests*: The author declare that she has no conflict of interest concerning this paper.

**Acknowledgement**

*Swarm* data are available through the European Space Agency Online platform (ftp://swarm-diss.eo.esa.int), after registration. We acknowledge the *Swarm* Science Team for providing the level 2 data and the *Swarm* visualization tool (https://vires.services/). The OMNI data on the solar wind, interplanetary magnetic field and geomagnetic indices are obtained from NASA/GSFC's Space Physics Data Facility's CDAweb service (http://omniweb.gsfc.nasa.gov/). This research was supported by the RSF (grant 16-17-00121).

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

# *Swarm* field-aligned currents during a severe magnetic storm of September 2017

Renata Lukianova[1,2]

[1] Space Research Institute, 117997 Moscow, Russia
[2] Institute of Earth Science, Saint Petersburg State University, 199034 Saint Petersburg, Russia

*Correspondence to*: Renata Lukianova (renata@aari.nw.ru)

**Abstract.** *Swarm* satellites observations are used to characterize the extreme behavior of large- and small-scale field-aligned currents (FACs) during the severe magnetic storm of September 2017. Evolutions of the current intensities
and the equatorward displacement of FACs are analyzed while the satellites cross the pre-midnight, pre-noon, dusk and dawn sectors on both hemispheres. The equatorward boundaries of FACs mainly follow the dynamics of ring current as monitored in terms of the SYM-H index. The minimum latitude of the FAC boundaries is limited to 50° MLat. The FAC densities are very variable and may increase dramatically, especially on the nightside ionosphere during the storm-time substorms. At the peak of substorm, the average FAC densities reach >3 $\mu A/m^2$. The dawn–
dusk asymmetry is manifested in the enhanced dusk-side R2 FACs on both hemispheres. In the 1 Hz data, filamentary high-density structures are always observed. In the pre-noon sector, the bipolar structures (7.5 km width FACs of opposite polarities adjacent to each other) dominate, while at the other local times the upward and downward FACs tend to be latitudinally separated. The most intense small-scale FACs, up to ~80 $\mu A/m^2$, is observed just in the post-midnight sector. Simultaneous magnetic and plasma perturbations indicate that this structure is likely a current system
of a mesoscale auroral arc.

**Keywords:** Ionosphere-magntosphere interaction, Field-alighned currents, Storms and substorms, Auroral arcs electrodynamics

## 1 Introduction

Field-aligned currents (FACs) provide electrodynamic coupling of the solar wind-magnetosphere-ionosphere system. FACs flow along the high-conducting geomagnetic field lines between different magnetospheric domains and the high-latitude ionosphere. The current system is driven by the internal
magnetospheric circulation of plasma and magnetic field within the global reconnection cycle (Dangey, 1961; Cowley and Lockwood, 1992) and by additional viscous-like interaction at the flanks of magnetosphere (Axford, 1964). Configuration of FACs is primarily controlled by the interplanetary magnetic field (IMF) orientation (Bythrow et al., 1984; Potemra et al, 1984). Other parameters of the solar

wind (velocity, density, IMF strength) and the ionospheric conductivity also play a role (e.g. Christiansen et al., 2002; Ridley 2007; Korth et al., 2002).

Schematic distribution of large-scale FACs has been established by Iijima and Potemra (1976) based on the
Triad satellite data. Subsequent space missions allowed constructing comprehensive empirical models of FAC parameterized by the IMF direction and strength, by season, and by hemisphere (Weimer, 2001; Papitashvili et al., 2002; Green at al., 2009). The ionospheric projection of the 3D FAC system consists of a pair of sheets elongated approximately along the magnetic latitude, namely, Region 1 (R1) and Region 2 (R2), with opposite current flow directions in the morning and evening local time sectors and additional
current sheets (R0) located on the dayside poleward of R1/R2. R1 FAC flows into the ionosphere (downward current) and from the ionosphere (upward current) on the dawn and dusk side, respectively. R1 currents, if reside on closed field lines of the Earth's magnetic field, are believed to originate in either the boundary layer or in the plasma sheet (Ganushkina et al., 2015). R2 FAC is considered to be a diversion of the partial ring current to the ionosphere driven by pressure gradients in the inner magnetosphere (Cowley,
2000). R0 current is connected to the dayside magnetopause and its polarity strongly depends on the IMF By component. On the Northern Hemisphere, the R0 current flows predominantly out of the ionosphere for positive IMF By and into the ionosphere for negative IMF By (Papitashvili et al., 2002; Lukianova et al., 2012). Additional (NBZ) current associated with the sunward ionospheric flow may appear inside the polar cap, if IMF Bz is northward (Iijima et al., 1984; Vennerstrøm et al., 2002).

While average large-scale (>150 km) current densities typically are of units of $\mu A/m^2$ or less, instantaneous small-scale FACs may reach several hundred $\mu A/m^2$ (Neubert and  Christiansen, 2003). The smaller-scale structures are often associated with auroral arcs which are accompanied by ionospheric conductivity and electric field perturbations (Aikio e al., 2002; Juusola et al., 2016). In particular, it was shown that in the
evening (morning) sector, there is downward FAC equatorward (poleward) of the arc and upward FAC above the arc. These two FAC regions are connected by a poleward (equatorward) horizontal current. Recent studies also confirmed that the cusp plasma injections are accompanied by pairs of FACs, upward at lower latitude and downward at higher latitude (Marchaudon et al., 2006).

Significant differences in the characteristics of FACs at different scales, especially near noon and midnight have been found (Gjerloev et al., 2011; Luhr et al., 2015; McGranaghan et al., 2017). Under stationary conditions the FAC system is evolved in accordance with the reconnection rate, which is controlled primarily by the solar wind. If a substorm occurs, additional FACs form a current wedge connecting the cross-tail current and the nightside westward ionospheric electrojet (Akasofu, 1964; Lui 1996). The magnitude of existing large-scale FACs also increases (Iijima and Potemra, 1978; Coxon et al., 2014). The dayside R1 currents are found to be stronger than their nightside counterpart during the substorm growth phase, at the same time the R1 region moves equatorward. After expansion phase onset, the nightside R1 currents dominate and their location moves to higher latitudes (Clausen et al., 2013). Recent studies have also suggested that the substorm current wedge could also include a R2 current system (Ritter and Lühr, 2008).

Magnetic storms are characterized by a dramatic enhancement of energy deposition to the Earth's atmosphere. During a magnetic storm, FACs become highly dynamic because of the enhanced solar-wind-magnetosphere interaction, release of energy stored previously in the magnetotail, particle precipitation and ring current build up. Storm-time FACs are stronger and more variable compared to non-storm FACs predicted by the climatological models. Since the intensity and time evolution of FACs vary from storm to storm, it is of interest to analyze their unique characteristics. However, relatively few papers focus on observed storm-time FACs. For example, utilizing the magnetic field measurements by *CHAMP* satellite Wang et al. (2006) investigated the northern and southern hemisphere dayside and nightside FAC characteristics during the extreme October and November 2003 magnetic storms. It was shown that as Dst decreases, the FAC region expand equatorward, with the shift of FACs on the dayside controlled by the southward IMF. For both case studies, on the southern (late spring) hemisphere the minimum latitude of the FACs is limited to 50° magnetic latitude (MLat) for large negative values of Bz (The minima are the same, although in October the IMF Bz drops dawn to -28 nT, while in November it reaches -50 nT.) On the northern (late autumn) hemisphere the equatorward boundaries of the FAC region are located at 55-60° MLat. Using the global maps from the *Iridium* constellation Anderson et al. (2005) studied the FACs intensities during severe magnetic storms which occurred during the solar cycle 23 with a particular attention to the evolution of FACs in the course of the storm of August 2000. The results revealed the dawn–dusk asymmetry of the R1/R2 current sheets, with an increase primarily found on the duskside. It was also shown that under disturbed conditions the total current intensity constrained to be below 20 MA (Anderson and Korth, 2007).

Since 2014, comprehensive studies of FAC distributions were carried out based on high precision observations onboard of *Swarm* constellation (e.g. Dunlop et al., 2015; Juusola et al., 2016; McGranaghan et al., 2017). However, the *Swarm* data have not yet been fully utilized for the storm-time FAC analysis. It is the purpose of this paper to characterize the magnitude and position of the large- and smaller scale FACs as their response to the magnetic storm development. The *Swarm* observations are used in order to identify various characteristics of the storm-time FACs for the event of 6–9 September 2017, which was one of the two most severe magnetic storms of the recent solar cycle 24 (the previous event was the St-Patrick storm on 17 March 2015). The September 2017 event is of particular interest because it was a two-step storm during which two major substorms occurred and the FAC system is affected by the storm-substorm interplay. In this paper we investigate the time evolution of the large scale FAC intensities, the displacement of the FAC equatorward boundaries and the extreme small scale currents.

## 2 *Swarm* satellites orbit

### 2.1 Instrumentation

The ESA *Swarm* mission is a constellation consisting of the three identical satellites (hereafter SwA, SwB and SwC, respectively), all are at the low-altitude polar orbits (Friis-Christensen et al., 2008). The *Swarm* constellation was launched in the end of 2013 and entered the operational phase in April 2014. The initial orbit altitude is 465 km (SwA and SwC) and ~520 km (SwB) and the inclination is 87.5°. By September 2017 the orbit altitude decreases down to ~440 and 505 km, respectively. SwA and SwC fly in a tandem separated by 1-1.4° in longitude and the differential delay in orbit is ~3 s. The orbit period is about 93 min (the speed of the satellites is about 7.5 km/s) and slightly different between SwA/SwC and the upper satellite SwB, so that their along-orbit separation in local time gradually changes. Their orbital planes also gradually drift apart and the separation angle increases by ~20° longitude per year. Slowly drifting in longitude, the orbits cover all the local time sectors over about 130 days.

The mission has a multi-instrument payload. The main module is the high-sensitivity vector (fluxgate type) and scalar magnetometers for determining the magnitude and direction of the total vector and variations of the geomagnetic field with an accuracy of more than 0.5 nT (Merayo et al., 2008). Magnetometers make it possible to carry out measurements in a wide range, including the Earth's main magnetic field and the variations of external magnetic field generated by FACs. FACs are detected by their magnetic perturbations in the orthogonal plane which are obtained after subtracting the main magnetic field model from the total

measured values. From single spacecraft the FAC density can be estimated based on one magnetic component with a techniques invoking Ampere's law under assumptions about the infinite current sheet geometry and the orthogonal crossing of the current sheet. This method was used for the previous one-satellite missions, such as Magsat and Ørsted (Christiansen et al., 2002). It is also applied to each *Swarm*

satellite separately. The dual-satellite estimation method calculates current density from curl(B) measured quasi-simultaneously at 4 locations is adapted for SwA and SwC data, where measurements separated along-track are used to create a 'tetrahedron' (Ritter and Lühr, 2006). The curl(B) method provides more reliable current density estimates, as it does not require any assumptions on current geometry and orientation. The FAC output of both a dual-satellite and a single satellite methods are considered to be in a

reasonable agreement (Ritter et al., 2013). However, a high degree of coherence is typical at auroral latitudes, while in the polar cap the results based ondual-spacecraft technique as more reliable (Luhr et al., 2016). Both algorithms are implemented to generate the *Swarm* products that are produced automatically by ESA's processing center as soon as all input data are available. The products are provided using the dual-satellite method on the lower pair of satellites SwA and SwC, and the single-satellite solution for

each of the Swarm spacecraft individually. The 1 s values (1 Hz sampling rate) of FAC densities are available via the on-line *Swarm* data portal (ftp://swarm-diss.eo.esa.int) as Level 2 data products (Swarm Level 2 Processing System Consortium, 2012). In the present study the single-satellite FACs are used in order to apply the similar method to SwB and SwA/SwC data.

Each satellite is also equipped with the Electric Field Instrument which includes the Langmuir probe to provide measurements of ionospheric plasma parameters: electron density, electron temperature and spacecraft potential (Knudsen et al., 2003). The plasma data are available at 2 Hz sampling rate as the standard product of the *Swarm* data base. Unfortunately, due to technical problems, measurements of the electric field and ions are rather rare. Nevertheless, the combination of data provided by a magnetometer

and a plasma analyzer on electrons makes it possible to identify perturbations associated with FACs. In each Level 2 data file the location of the satellite is presented in an geographic coordinate system NEC ($x$ - North, $y$ - East, $z$ - Center), where the $x$ and $y$ components lie in the horizontal plane, pointing northward and eastward, respectively, and $z$ points to the centre of gravity of the Earth. For the purpose of present study all projections of the passes are shown in the magnetic local time (MLT) and magnetic latitude (MLat) domain.

For this the coordinates are available via the on-line Swarm Data Visualisation Tool (VirES).

.

**2.2 Orbits on 6-9 September 2017**

The polar projection of the satellite orbits (14-15 trajectories per day) as of September 6-9, 2017 on the northern and southern hemispheres is shown in **Fig. 1**. For mid-September 2017 the passes are centered in the pre-midnight, pre-noon, pre-dusk and pre-dawn sectors. The satellite SwA (orbits of SwC are very similar) enters the region of MLat>50° between ~09 and 12 MLT, and leave this region between ~21 and 23 MLT. The entry (exit) points of the SwB orbit are between ~15 and 17 (02 and 04) MLT. On the southern hemisphere the direction of the tracks in the MLT-MLat framework is opposite. During a day, the successive projections are systematically shifted almost parallel to each other, however, in auroral latitudes, they stay mainly within the same sectors. The MLT ranges covered by the tracks are presented in **Table 1**.

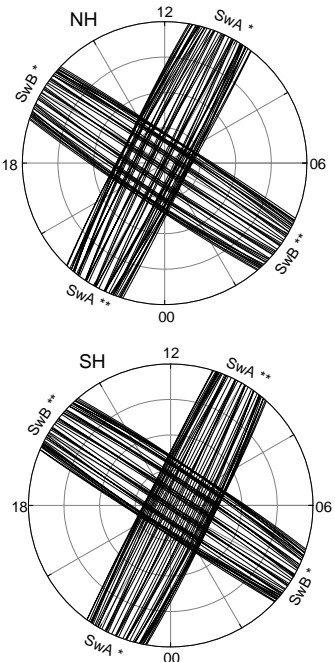

**Figure 1:** Polar maps of the SwA and SwB orbits on the northern and southern hemispheres on 6-9 September 2017 in the MLT-Mlat framework. Circles are drawn every 10° down to 50° MLat. Symbol * and ** indicates, respectively, the entry and exit crossing of the boundary MLat=50°.

**Table 1.** MLT range of the tracks in the northern and southern polar regions

| Satellite | MLT range within which the satellite cross the boundary of 50° (70°) MLat, (hh:mm)* | Center of the MLT range | |
|---|---|---|---|
| | | hh:mm | hh |
| Northern hemisphere | | | |
| SwB | 02:50-04:30   (01:30-05:10) | 03:40 | 04 |
| SwA (SwC) | 09:20-11:30   (08:40-12:50) | 10:30 | 10 |
| SwB | 15:00–16:50   (14:20-18:10) | 16:00 | 16 |
| SwA (SwC) | 21:00–22:50   (19:40-23:30) | 22:00 | 22 |
| Southern hemisphere | | | |
| SwB | 03:10-05:00   (01:50-06:20) | 04:00 | 04 |
| SwA (SwC) | 09:10-11:00   (08:30-12:20) | 10:00 | 10 |
| SwB | 14:50–16:40   (14:10-18:00) | 15:50 | 16 |
| SwA (SwC) | 21:20–23:10   (20:00-23:50) | 22:10 | 22 |

* with accuracy of 10 min

### 3 Space weather conditions on 6–9 September 2017

At the declining phase of solar cycle 24, starting from 6 September 2017, strong multiple solar flares occurred. The associated interplanetary coronal mass ejections collided with Earth's magnetosphere and caused the most intense magnetic storm of the recent solar cycle. The storm produced strong geomagnetic disturbances, ionospheric effects, magnificent auroral displays, elevated hazards to power systems and unstable HF radio wave propagation (e.g. Chertok et al., 2018; Clilverd et al., 2018; Curto et al., 2018;

Yasyukevich et al., 2018).

Evolution of the solar wind (SW) parameters and geomagnetic activity is presented in **Fig. 2** showing (from top to bottom): the IMF Bz and By, the SW proton speed (Vsw) and density (Nsw), the auroral AL and the equatorial SYM-H geomagnetic indices from the OMNI-web service (https://omniweb.gsfc.nasa.gov/).

Two SW shock events impact the magnetosphere. The arrival of the first shock late on 6 September (23:50 UT) results in a sudden increase in all parameters except the AL index. Since at that time the IMF Bz turns northward, the initial disturbance is only weakly geoeffective as a result. At 20:40 UT, 7 September, IMF Bz turns southward that triggers a substorm growth phase and a ring current build up. The second shock

arrived at ~23:40 UT on 7 September, with the SW speed up to 800 km/s and strongly negative Bz and By. This shock causes an abrupt drop of SYM-H down to -150 nT and a spike-like decrease of AL down to -2200 nT. After 03 UT, 8 September, the IMF Bz becomes positive, AL gradually approaches to zero and SYM-H starts to recover until the next southward turn of Bz. At ~06 UT on 8 September another strongly

5    negative Bz period is seen, and the SW speed remains high (>700 km/s). This causes the second substorm (AL is -2000 nT) and ring current intensification (SYM-H is -100 nT). A steady recovery occurs in the AL index throughout 9 September, while the SYM-H gradually increases from -75 to -35 nT. The SW parameters are not available for this day.

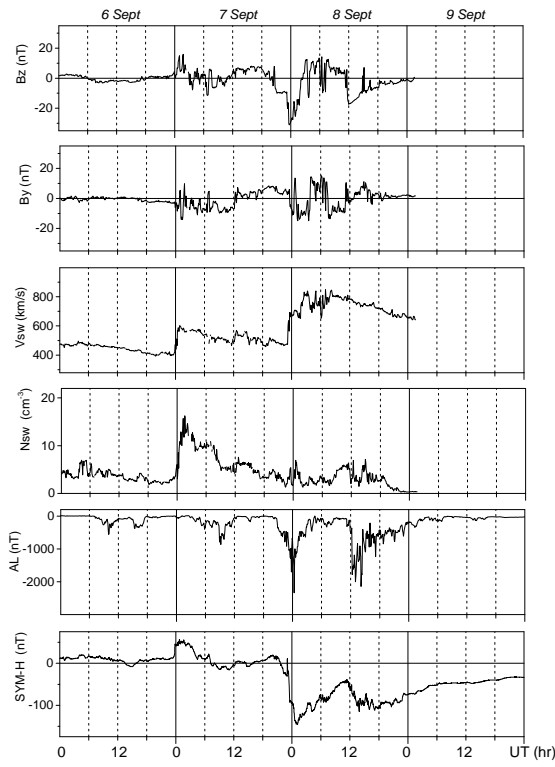

**Figure 2:** From top to bottom: IMF Bz and By, SW speed and density, AL and SYM-H indices on 6-9 September 2017 (5-min values).

## 4 Data analysis

### 4.1 FAC densities

Statistically the large-scale R1 and R2 FAC densities are peaked at the dawn-dusk meridian. In this regard, satellites orbits on September 2017 are not optimal for identifying the R1/R2 extremes, since they are deviated from this meridian. However, the local times of satellite paths are representative enough to assess the evolution of these FACs. On dusk, the orbits of SwB are centered at about 16 MLT that is not far from the region, where the current density is expected to be maximal. On the night side, the orbits are centered at 04 MLT, where they overlap the ionospheric westward electrojet, which is greatly enhanced when a substorm occurs. SwA and SwC cross the pre-noon sector at about 10 MLT, where both the downward R1 and upward R2 are often identified. These satellites also cross the pre-midnight sector, where disturbances associated with substorms are expected.

An example of the FACs measured along the SwB track is shown in **Fig. 3**. The 1 s values presented in **Fig. 3a** provide clear evidence for strong bursts in the auroral latitudes (55-75° MLat), while the near-pole region is almost empty of FACs. The auroral FACs exhibit large-amplitude spike-like structures, thus confirming the existence of filamentary current sheets embedded into the large scale current sheets. The intensities of these small-scale FACs vary from units to tens $\mu A/m^2$. **Fig. 3b** depicts the 51-point smoothed curve. It can be seen that the satellite approaching the pole from the dusk observes first the downward (positive) R2 and then the upward (negative) R1 current, both are of ~1 $\mu A/m^2$ density. Above approximately 70° MLat FACs become marginal. When the satellite moves equatorward in the early morning local times, a structure is observed, in which the poleward currents are positive, so they may be associated with the downward R1 FAC. The most equatorward currents are negative and thus represent the R2 FAC.

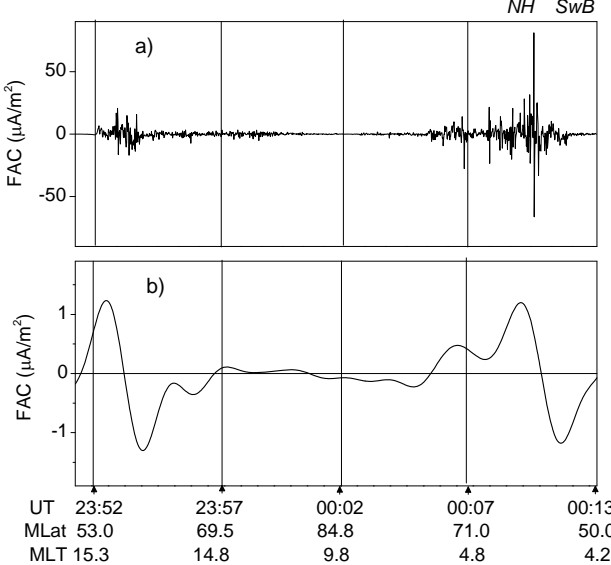

**Figure 3: (a)** 1 s and **(b)** smoothed FACs measured by SwB in the northern polar region between 23:50 UT, 7 September, and 00:13 UT, 8 September. Downward (upward) current is positive (negative).

To demonstrate the global temporal evolution of FACs, in **Fig. 4** the current intensities for the four MLT sectors are presented separately for the northern (**Fig. 4 a, c, e, g**) and southern (**Fig. 4 b, d, f, h**) hemispheres. Each red (blue) point is determined by averaging the 1 s downward (upward) current densities, when the satellite crosses the region filled with FACs. The upper (**a - d**) and lower (**e - h**) plots represent the data from the day side (10 and 16 MLT) and night side (04 and 22 MLT), respectively. For easier visual association of the evolution of FACs with the storm development, the SYM-H and AL indices are added in the plots (**a, b**) representing the day side and in the plots (**e, f**) representing the night side, respectively. During 6-9 September, FACs shown in **Fig. 4,** exhibit three pronounced enhancements, which are of different intensity depending on the MLT sectors. (Note that the FAC densities do not show any systematic changes associated with the orbit ocsilllation during the day.) All FACs start to increase in the very beginning of September 7 in association with the SW dynamic pressure front impinges the magnetosphere causing a positive excursion of SYM-H. The dayside FACs increase abruptly (this is especially well seen in **Fig. 4 b - c**, i.e. at 10 MLT, north, and at 16 MLT, south), while the nightside FACs

(**Fig. 4 e - h**) respond to the shock with a considerable delay. The nightside FACs are peaked in the middle of September 7, when a moderate substorm occurs.

In the very beginning of September 8, in association with the first deep drops of SYM-H and AL, a step-like increase is seen at all MLTs except the prenoon sector. The peak of the day- and nightside FACs reaches 2.5 and 3.5 $\mu A/m^2$, respectively. For a particular crossing the average density exceeds 5-6 $\mu A/m^2$ as seen from the standard deviation. The dayside FACs (**Fig. 4 a - d**) stay enhanced during the whole day of 8 September. The nightside FACs (**Fig. 4 e - h**) more closely follow the evolution of AL, so that the current intensities decrease in accordance with the first storm-time substorm recovery. The next increase in the nightside FACs occurs at ~12 UT on September 8, when the second major substorm occurs and the second drop in SYM-H is observed. On the day side the response of FACs to this substorm is marginal, although the current densities remain elevated throughout the day. All FACs fall to pre-storm levels by September 9.

Comparison of the evolution of FAC intensity with the SW and geomagnetic parameters during the period of 6-9 September reveals that the storm-time FACs are, on average, by several times larger than the quiet-time ones. Better correspondence exists between the nightside FACs (compared to the dayside ones) and the substorm activity as monitored by AL index.

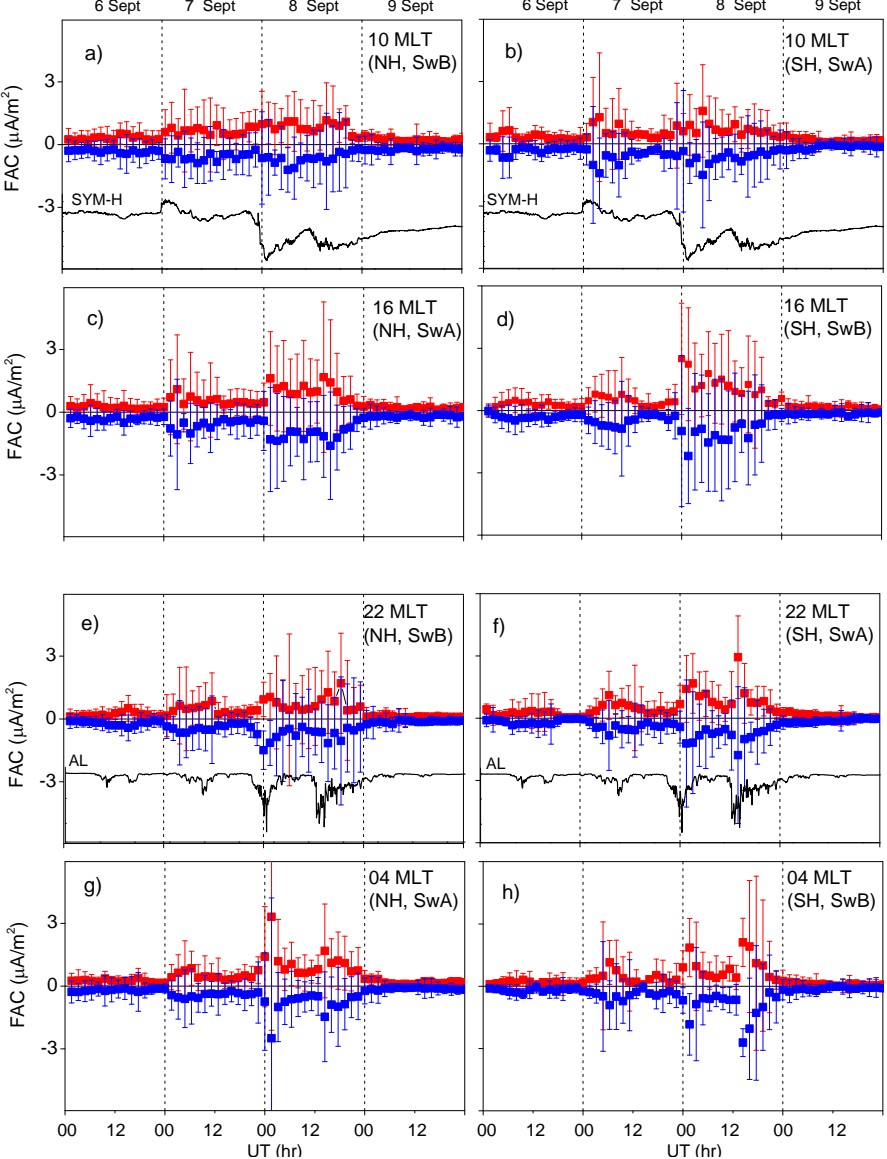

**Figure 4:** Average FAC densities in the four local time sectors covered by the *Swarm* data on 6-9 September, 2017. The left column of plots corresponds to the northern hemisphere (NH) and the right column corresponds to the southern hemisphere (SH). The upper plots (a-d) and the lower plots (e-h) show the dayside and nightside FACs, respectively. The SYM-H and AL indices are added in the plots (a, b) and (e, f), respectively. The centered MLT (10, 16, 22 and 04) is shown in the right upper corner of each plot. The downward and upward FACs (and the corresponding error bars) are shown in red and blue, respectively.

**4.2 Dawn-dusk asymmetry**

During the event of September 2017 a signature of the dawn-dusk asymmetry in the storm-time FACs is revealed. Although the estimate is based on a limited number of crossings and does not allow calculating the total FAC, the current densities summed separately over the dawn and dusk sides may serve as a proxy.

Those parts of the tracks which fall into the 00-12 MLT (12-00 MLT) sector are considered to be related to the dawn (dusk) side, MLat for 50°–90° is accounted. The summed up- and downward FAC densities in each MLT sector as well as those for dawn and dusk sides are presented in **Table 2**. The summation is over the entire 4-day interval.

For any given pass, the net summed FAC density is frequently nonzero. As seen in **Table 2** the net FACs summed in a particular MLT range, is also nonzero. The difference between upward (negative) and downward (positive) current densities varies from 1 to 15%. For the sectors centered at ~04 and 10 MLT this difference is relatively small (-0.7 and +0.1 $\mu A/m^2$ in the Northern hemisphere, and 1.6 $\mu A/m^2$ and -0.4 $\mu A/m^2$ in the Southern hemisphere), i.e the R1 and R2 densities are of comparable values. For the sectors

centered at 16 and 22 MLT the downward (R2) FACs exceed the upward (R1) FACs by 5.2 $\mu A/m^2$ and 3.1 $\mu A/m^2$ in the Northern hemisphere, and by 3 $\mu A/m^2$ and 3.4 $\mu A/m^2$ in the Southern hemisphere.

If the MLT sectors are combined in pairs in order to obtain FACs summed over the dawn and dusk sides, the prevalence of the dusk-side downward current is revealed. From the values presented in the two last

columns of **Table 2** one can see that on both hemispheres the duskside downward current (+60 and +58.9 $\mu A/m^2$ on the Northern and Southern hemisphere, respectively) is stronger than all the other currents. Although the numbers in **Table 2** contain uncertainties related to the lack of global observations, the estimate based on the summed FAC densities from in-situ *Swarm* measurements indicate the existence of the storm-time dawn-dusk asymmetry. Even a limited number of crossings show a clear tendency of the

prevalence of the dusk-side R2. This predominance implies an additional amplification of the storm-time R2 FAC on the dusk side, which is related to the partial ring current. This shift may result from a strong dusk side ion pressure leading to asymmetric dusk-side inflation of the magnetic field consistent with a partial, dusk side, ring current during storm main phase (Liemohn et al., 2001; Anderson and Korth, 2007).

**Table 2**. Summed upward (negative) and downward (positive) FAC densities in for all passes on 6-9 September. In the last two columns the conventional FAC regions (R1 or R2) are indicated in brackets. The largest values are shown in bold.

| Side | MLT range (as at 50° MLat) * | FAC densities ($\mu A/m^2$) | | | |
|---|---|---|---|---|---|
| | | up | down | up | down |
| Northern Hemisphere | | | | | |
| dawn | 09:20 – 11:30 | -23.3 | +23.4 | -54.6 (R2) | +53 (R1) |
| | 02:50 – 04:30 | -31.3 | +29.6 | | |
| dusk | 21:00 – 22:50 | -27 | +30.1 | -52.7 (R1) | **+60** (R2) |
| | 15:00 – 16:50 | -24.7 | +29.9 | | |
| Southern Hemisphere | | | | | |
| dawn | 09:10 – 11:00 | -27.8 | +27.4 | -51.5 (R2) | +52.7 (R1) |
| | 03:10 – 05:00 | -23.7 | +25.3 | | |
| dusk | 21.20 – 23.10 | -23.8 | +27.2 | -52.5 (R1) | **+58.9** (R2) |
| | 14:50 – 16:40 | -28.7 | +31.7 | | |

* with accuracy of 0.5 hr

## 4.3 Dynamics of the equatorward boundary of the FAC region

It is well established that the enhanced SW input and the pile-up of open magnetic flux during a geomagnetic storm results in the equatorward expansions of the polar cap and the auroral oval as a whole
(e.g. Milan et al., 2004). Following the magnetospheric dynamics FACs also move equatorward. **Fig. 5** shows the evolution of the equatorward boundary (EqB) of FACs on 6-9 September. For the comparison the SYM-H and AL indices are added. The EqB parameter is determined as the lowest MLat at which FACs are terminated. The procedure of the 20-point sliding window (the scale is about 150 km) moving along a track from the equator to the pole is applied to the 1 s FAC values and the corresponding MLats.
EqB is selected as the lowest MLat of the window if 90% of FAC values within the window exceed |0.1| $\mu A/m^2$. Then the results are checked visually in order to avoid the erroneously calculated latitudes, that may happen, e.g., if a significant latitudinal gap between R1 and R2 occur. When calculating EqB, no separation between the up- and downward FACs is made.

Even visual comparison of the SYM-H and EqB evolutions in **Fig. 5** reveals generally coherent behavior of these two parameters. In particular, during a period preceding the storm main phase (before 8 September, when SYM-H is mainly positive) EqB is located much lower than during the end of recovery phase (after ~12 UT on 9 September, when SYM-H is still negative). Before the SYM-H attains the negative values below -20 nT at 22:00 on 7 September, FACs are observed mainly poleward of 60° MLat on both hemispheres. Moderate equatorward shifts of EqB are associated with the modest substorms occurred before the storm main phase in the middle of 6 and 7 September. Prior the main phase, on both hemispheres the prenoon (10 MLT) EqB is found considerably poleward compared to the EqB location at other MLTs. The effect is well seen during the two time intervals: from ~22 UT, September 6 till 06 UT, September 7 and at 12-24 UT, September 7. Both intervals are dominated by the northward IMF (sf. **Fig. 2**), so that a shrinking of the polar cap and a poleward shift of the auroral oval is expected. With regard to the position of FACs, the displacement of its equatorward boundary is the largest only in the pre-noon sector, while the other local times remain less affected.

Upon arrival of the SW shock at the very end of September 7, EqB is abruptly shifted equatorward, then tends to recover until the middle of September 8, and then drops again following the second intensification of the storm. At the very beginning of 8 September EqB is found at its lowest position at 50° MLat. A drop of EqB occurs simultaneously with the peak of the first substorm intensification and the lowest SYM-H (-160 nT). The second substorm reaches its peak slightly before the second minimum of SYM-H (at 12:50 and 13:55, respectively). During this second activation the EqB is shifted again as low as 50° MLat (although SYM-H is only -100 nT). As seen in **Fig. 5**, the evolution of EqB tends to follow the gradual change of SYM-H rather than abrupt drops of AL related to the substorm activations (see also **Fig. 2** for AL). Unlike the current density, which is enhanced throughout the storm and exhibits several spike-like increases in accordance with AL, the temporal variations of EqB are relatively smooth. Relatively small difference in evolution on the day- and nightside EqBs is observed. At the peaks of the storm, EqB is at about 50° MLat, while during the late recovery phase, EqB is shifted poleward as high as 70° MLat. Possible expansion of the FAC region during the substorm growth phase, and then its contraction after onset are difficult to resolve with the *Swarm* data.

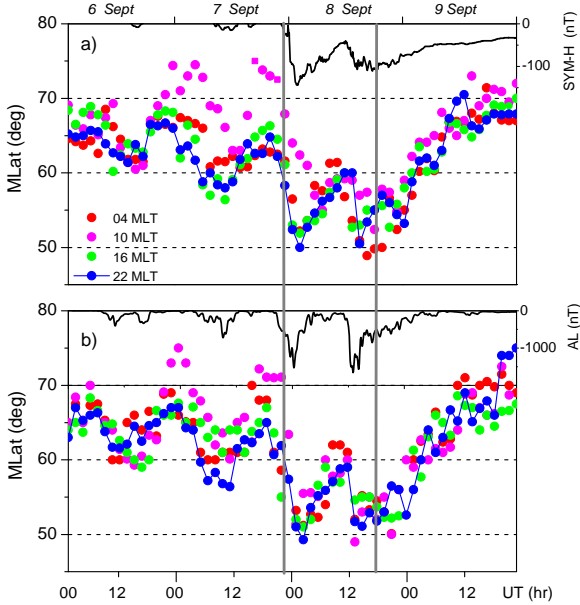

**Figure 5**: MLat of the FAC equatorward boundaries (EqB) on the Northern (a) and Southern (b) hemispheres for the sectors centered at around 04, 10, 16 and 22 MLT. EqB for each sector is shown by dots of different colors; blue dots representing the nightside (~22 MLT) EqB are connected by a line. The SYM-H and AL index (black line) is added to
5 the upper and lower plot, respectively. The vertical lines mark the beginning of the main and recovery phases.

The equatorward displacement of FACs roughly correlates with the storm intensity as monitored by the SYM-H index, while the storm-time subsorms can modify this relationship. In **Fig. 6,** separately for the
10 main and recovery phases, the correlations between SYM-H and the nightside EqB are shown. Data from both the northern and southern hemispheres are included. The correlation coefficients for the main and recovery phases are very similar (cc=0.88 and 0.87), while the corresponding regression equations are considerably different. During the storm main phase, the equatorward expansion of EqB is governed by the equation MLat=63.1+0.1·SYMH. When the recovery phase begins, the poleward shift of EqB is described
by the expression MLat=79.5+0.3·SYMH. The faster poleward recovery of EqB comparing with its equatorward expansion is due to the fast decrease in substorm activity on September 9 .

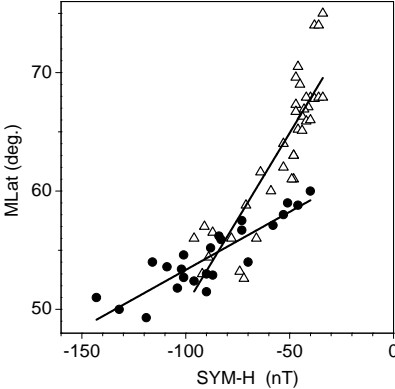

**Figure 6**: Correlations between the SYM-H index and the latitudinal position of the nightside (~22 MLT) EqB: black dots and open triangles correspond to the main and recovery phase, respectively.

### 4.4 Small-scale FACs

It is known that FACs appear on a wide range of scales from large-scale sheet-like currents of hundreds kilometers width to very small-scale filamentary currents of hundreds meters width. The quasi-instantaneous amplitudes of the small-scale component are often much larger than the stationary R1/R2
FACs. The current intensity vary inversely with scale so that large-scale currents are typically a few $\mu A/m^2$, whereas the smaller scale (down to 10 km) are a few tens $\mu A/m^2$ (Neubert and Christiansen, 2003; Luhr et al., 2015; McGranaghan et al., 2017). To obtain the time-series of the *Swarm* peak current densities on 6-9 September 2017, the largest positive and negative 1 s values were selected from each crossing in a given MLT time sector irrespective of the hemisphere. The obtained peak values are presented in **Fig. 7**. First of
all, from this figure one can see that the small-scale peaks may be more than an order of magnitude larger than the FACs averaged over a track (sf. **Fig. 4**). On September 6, only two outliers of about +20 $\mu A/m^2$ and -30 $\mu A/m^2$ are observed. Both are from the pre-midnight sector and are associated with a moderate substorm occurred in the middle of this day. During the disturbed period, starting with the compression of the magnetosphere on September 7, the amplitude of peaks tends to increase. Two intense substorms
occurring during the storm main phase cause an additional strengthening of small-scale FACs at all MLTs. At ~00 UT on September 8, the up- and downward currents located in early morning local times attain their extremes of 70-80 $\mu A/m^2$. The second major substorm occurred in the middle of September 8 is also

accompanied by the peaks, which are more pronounced in the dusk-side, where the upward FAC reaches about -50 μA/m$^2$. Note that some peaks can be missed due to the temporal and spatial gaps between the satellite tracks.

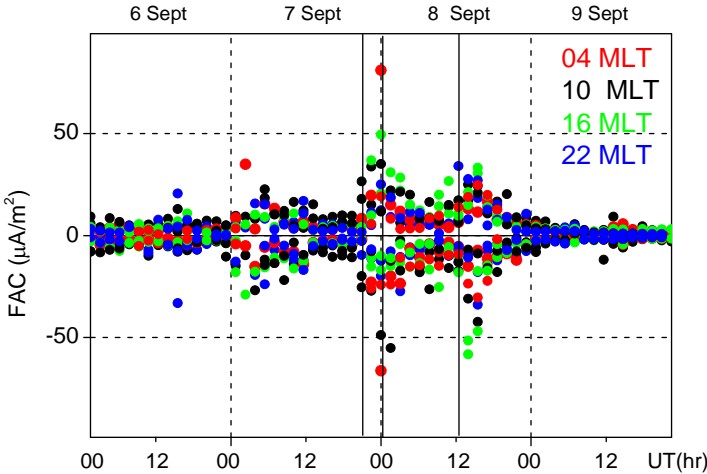

**Figure 7**: The largest downward (positive) and upward (negative) 1 s current densities for four MLT sectors on 6-9 September. The vertical solid lines mark the beginning of the storm main phase at 22:00 on 7 September (the time when SYM-H attains its stable negative values <-20 nT; the period of SYM-H<-20 lasts till the end of September 9), the peaks of the first and second major substoms (the time when AL attains its minimum).

When for each crossing within a certain MLT sector, the minimum (i.e. peak upward current) and the maximum (i.e. peak downward current) 1 s FACs are selected, it appears that in some cases these peaks are observed at very close latitudes, while in other cases the minimum and maximum are spaced in latitude. In **Fig. 8**, the correlations between the MLats, at which the most intense small-scale FACs of opposite polarities are observed, are presented for each MLT sectors. The x-axis (y-axis) corresponds to the MLat of the downward (upward) peak selected in each crossing. The magnitude of minima and maxima are not accounted. From **Fig. 8** one can see that correlation between the latitudinal positions of the up- and downward peaks varies with MLT. The highest correlation coefficient (cc=0.94) is found in the pre-noon sector (**Fig. 8b**). This is indicative of a large population of the paired, closely adjacent small-scale currents of opposite polarity (called hereafter the bipolar structure). In the dusk (**Fig. 8a**) the correlation coefficient

decreases down to 0.78. Almost the same correlation (cc=0.75) is observed in the pre-midnight sector (**Fig. 8c**). At the early morning hours (**Fig. 8d**) the correlation is much weaker (cc=0.53) implying that the extreme up- and downward currents appear less frequently in pair but rather are spatially (or temporary) separated. Different mechanisms of the small-scale FAC formation on the day- and night side can be the cause of this spatial distribution and variability.

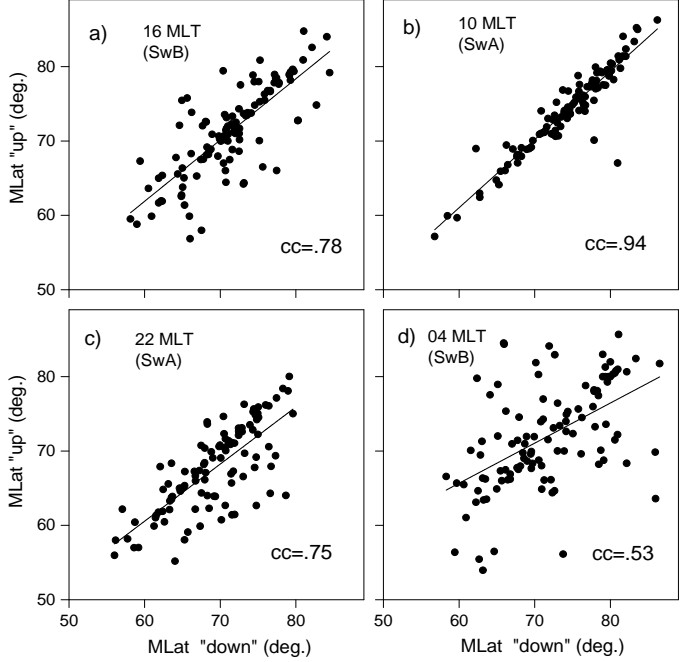

**Figure 8**: Correlations between magnetic latitudes, at which the up- and downward peak FACs are observed: (a) dusk, 16 MLT; (b) pre-noon, 10 MLT; (c) pre-midnight, 22 MLT; (d) post-midnight/early morning, 04 MLT.

## 4.5 Small-scale FACs of extreme amplitudes

During the storm under consideration a pair of the most intense upward and downward small-scale FACs is revealed by SwB at around 00:10 UT on September 8, when the satellite traverses the auroral latitudes from north to south over the geographic area of the Barents Sea, about 20 degree magnetic longitude to the East

from the IMAGE magnetometer network (http://space.fmi.fi/image). The network produces the IL index, which is simple estimate of the total westward currents crossing the IMAGE chain. The IL index (**Fig. 9**) shows that the extreme FACs are observed during the first period of the storm-time substorm intensifications, several minutes before the IL drops from -1500 nT to -3700 nT.

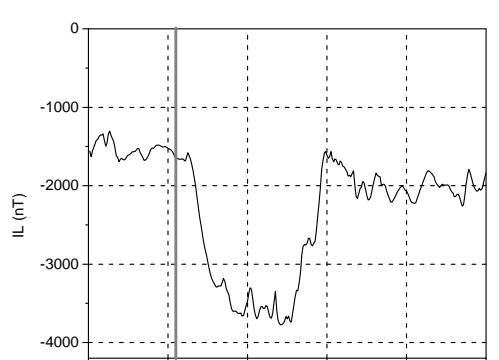

**Figure 9**: The 10 s IL index at 00:00-00:50 UT, September 8. Time of the extreme FAC observation is shown by grey line.

The 1 s FACs and plasma parameters (the electron density, Ne, temperature, Te, and the spacecraft electric potential, $U_{sc}$) measured by SwB at 00:08-00:12 UT on 8 September are shown in **Fig. 10**. As show in **Fig. 10a,** at 00:10:18-00:10:19 UT the satellite observes the bipolar current structure of extreme density consisting of the poleward downward (81 $\mu A/m^2$) and equatoward upward (-66 $\mu A/m^2$) FACs. The paired up- and downward FACs are of relatively comparable values, thus they are balanced and likely closed

locally. In **Fig. 10a** the original 1 s values are superimposed to the smoothed curve, which reveals that the bipolar structure is located at the edge of the mesoscale downward FACs.

The bipolar current structure is accompanied by plasma perturbations. A narrow peak in Ne up to $77 \cdot 10^3$ $cm^{-3}$ (**Fig. 10b**) and an increase of Te up to $\sim 10^4$ K on average (**Fig. 10c**), that is $\sim$50% above their ambient

values, are observed almost simultaneously with a pair of extreme FACs. (It should be noted that the Te values presented here are based on the current processing of the satellite data and may be still uncalibrated. However, it hardly affects the relatively small-scale perturbations.) The elevated Te is observed in a wider region slightly poleward of the enhanced Ne. The plasma disturbances are clearly seen in $U_{sc}$, which is

proportional to $-k \cdot Te$ ($k$ is the Boltzmann constant). Note that the level of noise for the $U_{sc}$ channel is much lower compared to that for the Te channel (0.4% and 2% for $U_{sc}$ and Te, respectively). **Fig. 10d** shows that a reduction of $U_{sc}$ starts at 00:09:56 UT, then peaks at 00:10:08 (-12 V) and 00:10:20 UT (-8 V), the average decrease is -5 V. The region, where the Te and $U_{sc}$ are perturbed is several times wider than region occupied by the pair of extreme FACs..

If the localized increase in Ne indicates conductance enhancement (likely due to precipitating electrons), the observed plasma and current perturbations are similar to those associated with auroral arcs (Opgenoorth et al. 1990; Lyons, 1992; Lewis et al., 1994; Johnson et al., 1998; Aikio et al., 1993; Juusola, et al., 2016). In particular, Aikio et al. (2002) studied the current system of arcs in the evening sector, where the background electric field is northward. It was shown that for arcs located within the northward convection electric field currents flow downward on the equatorward side of the arcs, then poleward, and upward from the arcs. The arcs are associated with an enhanced northward-directed electric field region on the equatorward side of the arc. An enhancement in the electric field starts already several tens km equatorward of the arc edge.

During the storm under consideration the bipolar FAC pattern observed at 00:10 UT is located in the morning sector, where the background electric field is expected to be southward. This is confirmed by the SuperDARN-based convection model (http://vt.superdarn.org/tiki-index.php?page=ASCIIData), which predicts in the region of the SwB observations the magnitude of the southward and westward component of about 6.5 mV/m and 0.5 mV/m, respectively. As mentioned in Section 2.1, unfortunately the in-situ *Swarm* electric field is unavailable. Only the reported characteristics of the electric field associated with arcs can be used for qualitative analysis. In particular, for morning side arcs an enhanced southward electric field on the poleward side of the arc is expected. In this case the current pattern consists of a downward FAC on the poleward side of the arc connected to an upward current above the arc by an equatorward ionospheric closure current. This is exactly what is seen in **Fig. 10a**: when SwB flies away the pole, it first observes a positive spike (downward FAC) and then a negative spike (upward FAC). Since the width of the region of enhanced Ne is ~30 km, the arc is relatively narrow. Comparing **Fig. 10a** and **Fig. 10b** one can see that the paired FACs is located on the poleward side of the region of enhanced Ne. Note, that in **Fig. 10b** a sharp increase in Ne up to ~$80 \cdot 10^3$ cm$^{-3}$ is preceded by a weaker spike-like drop down to ~$30 \cdot 10^3$ cm$^{-3}$. A

decrease in Ne (which is usually much less pronounced than an increase due to precipitating electrons) is associated with a downward FAC observed at the opposite boundary of the arc. Elevations of Te may be created by electric fields which can arise within narrow region adjacent to the northern side of the auroral arc as observed by Aikio et al. (2002).

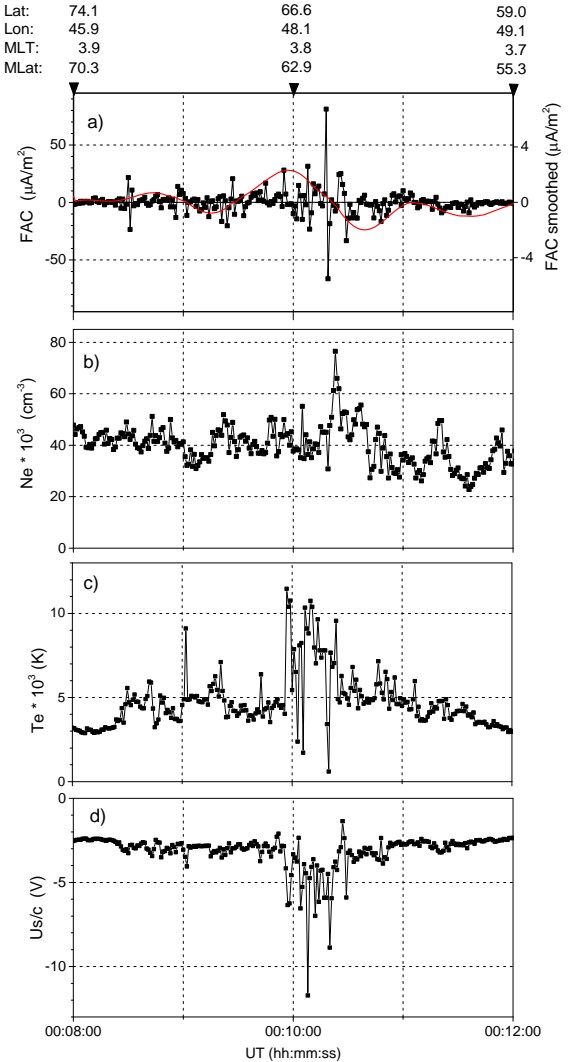

**Figure 10**: The 1 s values of (a) FAC density, (b) Ne, (c) Te, and (d) Us/c measured along the SwB track at 00:08:00 – 00:12:00 UT, 8 September. In the upper plot the 21-point smoothed FAC density is also shown. Geographic and geomagnetic coordinates are shown on the top.

## 5 Discussion

Observations of the LEO *Swarm* multi-satellite mission are used in order to identify various characteristics of the storm-time FACs for the severe event of 6–9 September 2017. During the storm main phase two major substorms occurred, so that the FAC system evolved under conditions of the storm-substorm interplay. In mid-September 2017 the separation between the upper and lower *Swarm* satellites was about 6 hours in local time. Within the sectors centered at 04, 10, 16 and 22 MLT the northern and southern polar regions were covered by about 60 tracks along which the 1 Hz measurements of FACs were carried out. These observations made it possible to reveal the evolution of the large scale FAC intensities, the displacement of the FAC equatorward boundaries and the some features of the extreme small-scale FACs.

### 5.1 Large-scale characteristics of FACs

Evolution of large-scale characteristics of FACs during the September 2017 storm is in general agreement with regularities observed previously by *CHAMP* during the intense 2003 geomagnetic storms (Wang et al., 2006). The common feature of all storm-times is the equatorward motion of FACs generally correlating with the storm intensity. During the September 2017 storm the global coverage of the high latitudes by the precise measurement onboard of *Swarm* satellites made it possible to reveal that the FACs was enhanced at all MLTs starting from the time of the first SW shock arrival at the very beginning of 7 September, although the northward IMF and the prolonged period of geomagnetic quietness lasted almost a day. After this quietness a storm abruptly commenced at ~22: UT on 7 September. During the two-step main phase FACs exhibit three pronounced enhancements, the evolution of FACs depends on the MLT sectors. On the dayside FACs strengthen after the sudden commencement and in response of the first drop of SYM-H, while the response to the second drop of SYM-H is relatively weak. On the night side the current intensities follow mainly the substorm dynamics as monitored in terms of the AL index, promptly respond to the onset of storm-time substorms and strengthen at the peak of substorm. At the same time, during the period between the major substorms, when AL is fully recovered, but SYM-H is not, FACs stay considerably enhanced.

The September 2017 storm is characteristics of a considerable equatorward expansion of the FACs region as low as 50° MLat on both hemispheres. The latitudinal displacement of FACs is more gradual and smooth than the changes in current intensity. For comparison, during the 2003 storms the minimum latitude of peak current density are limited to 52-56° MLat (Wang et al., 2006). It should be noted that these authors

defined the latitudinal positions of peak current density but not the most equatoward boundary of the FAC region, thus the actual FAC region may expand to lower latitudes. The lowest latitudinal position of the storm-time FACs was found by Fujii (1992). For the storm of March 1989 the equatorward boundary of the FAC system reached as low as 48° MLat.  Similar to the 2003 storms, in 2017 the latitudinal positions of EqB generally follow the SYM-H variations. FACs are shifted further equatorward during the storm-time

substorms. Even a relatively minor substorm occurred prior the storm causes a considerable equatorward displacement of FACs. The lowest latitude of EqB is observed when both the SYM-H and AL indices reach their minimums.

Although the storm of September 2017 is considerably weaker (Dst≈-100 nT) than the storms occurred in

1989 (Dst≈-600 nT) and 2003 (Dst≈-400 nT), the FAC region expands approximately to the same latitudes. This effect may be interpreted in terms of saturation, when the FAC region does not expand lower than ~50° MLat independently of the storm severity. Linear dependence between latitudinal boundaries of the FAC sheets upon the dayside merging electric field, the AE and Dst indices has been reported by Xiong et al. (2014). It was also pointed out that toward high activity a saturation of equatorwards expansion seems to

set in.

In September 2017, prior the storm main phase, when the IMF Bz is northward, the pre-noon EqB is located at higher latitudes (~75° MLat) compared to the other MLT sectors (~65° MLat). Surprisingly, in the course of the storm main phase, no considerable difference between the latitudinal positions of EqB in

different MLT sectors is found. After ~12 UT on September 9, in the late recovery phase (SYM-H is -50 nT), both the day- and nightside EqB recover to their undisturbed position (about 70° MLat). The coherent behavior of EqB is rather unexpected because Wang et al (2006) found that the poleward recovery of FACs on the nightside is slower than on the dayside. Previous analysis of the latitudinal shift of the polar cap boundaries based on the *IMAGE* observations during a magnetic storm has also shown that, if the IMF Bz

turns northward, the dayside boundary recovers much faster than the nightside boundary (Lukianova and

Kozlovsky, 2013). This is because dynamics of the nightside boundary depends on the energy accumulated in the magnetotail during the previous period of the storm main phase. However, it seems that the storm of September 2017 does not show the same regularity. The reason may be that during the storm main phase the two major substorms occurred, so that the energy stored in the tail was released faster. Comparing the evolution of the FAC densities and the equatorial boundary positions during the storm recovery, one can see that the densities decay much faster than the boundaries return to their quiet time positions.

High FAC intensity is associated with the auroral oval. Previous studies based on particle precipitation and optical observations have shown that the oval radius increases when the ring current is intensified during magnetic storms (e.g., Meng, 1982; Yokoyama et al., 1998). Significant variations in the location of the aurora take place during the substorm cycle. Substorms occurring on expanded auroral ovals during magnetic storms are most intense, since they close the most magnetospheric open magnetic flux and the presence of the enhanced ring current increases the open flux threshold at which substorm onset is favoured (Milan et al., 2009). It was also shown that changes in oval radius associated with dayside and substorm driving  occur on timescales of minutes and hours, while changes associated with the ring current are more protracted as the ring current dissipates slowly (Milan, 2009).

The *Swarm* observations, although they are instantaneous, reveals a tendency of the dawn-dusk asymmetry FACs. The dawn-dusk asymmetry is revealed by comparing the up- and downward FACs, which are summed for all crossings over dusk and dawn separately. While the summed FAC intensities are comparable between the two hemispheres, the positive and negative densities on the dusk and dawn are slightly imbalanced and the net current is nonzero. It seems that the dusk-side downward (R2) FACs are larger than the dusk-side upward (R1) and the dawn-side R1 and R2 currents. The observed imbalance in FACs is likely related to an intensification of partial ring current, which is connected to R2 FAC on the dusk. Strengthening of partial ring current may also lead to asymmetric dusk-side inflation of the geomagnetic field lines. The dawn-dusk asymmetry in strength and the equatorward displacement of R1 and R2 at the peak of the major storm on August, 2000, has been reported by Anderson and Korth (2007). This study was utilized the global distributions of FACs generated at a 10 min cadence separately for the Northern and Southern Hemisphere by the AMPERE project  which is based on the fleet of Iridium satellites. Although the *Swarm* observations unable to provide the instantaneous global FAC distribution, the responses of FACs in certain

MLT sectors on the dawn side are different from those on the dusk side. Note that the results in Table 2 are calculated by using the 1 Hz FAC values and their averages do not necessary represent the large-scale R1/R2 FACs. Nevertheless, for the storm of September 2017, the dawn–dusk asymmetry is manifested in the enhanced average density of the downward FACs on the dusk side. This feature is consistent with the global observations by AMPERE, from which the asymmetry of large-scale FACs can be identified. At the same time, almost no difference in the equatorward shift of the dusk and dawn side FACs is observed by *Swarm*.

## 5.2 Small-scale FACs

Due to their large amplitudes small-scale FACs play an important role for the energy input to the upper atmosphere. In several previous studies, the FACs associated with arcs were estimated as 1-10 $\mu A/m^2$ (Bythrow and Potemra, 1987; Elphic et al., 1998; Janhunen et al., 2000; Luhr et al., 2016). Larger range of current densities, varying between 4 and >40 $\mu A/m^2$, has been observed (Aikio et al., 2002) and even more intense small-scale FACs, up to hundred of $\mu A/m^2$, at the edges of arcs have been measured by MEO satellites (Marklund et al. 1982; Bythrow et al. 1984). Such a large range of the FACs estimates is likely related to its different scales (and different techniques), because for arcs with very sharp electron density gradients, the FACs associated with ionospheric currents flow in narrow regions at arc edges. If the real widths are smaller, the current densities are expected to be larger.

Filamentary structures of high densities are always presented in the *Swarm* observations. The narrow high-density currents are averaged out when integrated over a FAC region, so that multilayer structures of steady large-scale FACs of the R1/R2 type depicted by Iijima and Potemra (1978) can be revealed after a proper smoothing. From a statistical study of the temporal and spatial-scale characteristics of different FAC types derived with the *Swarm* satellites Luhr et al. (2015) have shown that small-scale, up to some 10 km FACs are carried predominantly by kinetic Alfven waves. A persistent period of small-scale FACs of order 10 s, while large-scale FACs can be regarded stationary for more than 1 min. Neubert and Christiansen (2003) studied the morphology of very small-scale FACs from a survey of *Oersted* satellite 25 Hz data. These FACs are distributed in a broad region around the pre-noon and cusp region, and in the pre-midnight sector. It was found that at the considered time scale, instantaneous currents may reach the largest values up to 1000 $\mu A/m^2$, while the average current densities reach a maximum of 10 $\mu A/m^2$. McGranaghan et al.

(2017) demonstrated a local time dependence in the relationships between large (>250 km) and small FAC scales (10–150 km width, density is up to 0.5 μA/m$^2$). It was found that linear relationships exist near dawn and dusk local times, while at noon and midnight local times no similar regularity is seen. The results are based at all available data from the *Swarm* satellites and the AMPERE irrespective of the level of
geomagnetic activity.

During the September 2017 storm one of the *Swarm* satellites managed to observe a pair of the most intense small-scale 7.5 km width FACs of opposite polarity, the magnitude of which are approximately +80 and -70 μA/m$^2$. These up- and downward FACs are adjusted to each other and separated in a fraction of
degree in MLat. The bipolar FAC structure occurs in the region approximately between R1 and R2, just prior of the abrupt substorm intensification in the vicinity of the newly developed ionospheric WEJ. The polarity reversal captured by the *Swarm* data for two consecutive seconds implies a quite localized current closure through the ionosphere mostly via Pedersen horizontal currents. Although without optical and electric field data one could not make a strict conclusion, the small-scale bipolar FAC pattern accompanied
by a localized enhancements in Ne and Te are likely associated with mesoscale discrete aurora. One-to-one correspondence of small-scale FACs with localized electron precipitation events has been previously observed (e.g. Fukunishi et. al., 1991). The SwB observations are in agreement with the disturbances expected for the acrs occurred on the morning side, where the ambient electric field is southward. The observed features are resemble to those reported by Kozlovsky et al (2007) and Aikio et al. (2002) but
bearing in mind that the latter are related to the evening sector, where the background electric field is northward. Based on *Swarm*/*THEMIS* ASI observations Wu et al. (2017) has associated multiple auroral arcs with up/down current pairs. For these arcs unipolar and multipolar FAC systems with current densities of about a few μA/m$^2$ have been observed. Arcs in unipolar FAC systems have a typical width of 10–20 km and a spacing of 25–50 km. Arcs in multipolar systems are wider and more separated. In the bipolar
structure of extreme intensity observed by SwB in September 8, the current density exceeds the values observed by Wu et al. (2017) at least by a factor of ten, while the spatial extend of FACs is smaller. This difference implies to the existence of sharp electron density gradients at arc edges. Usually, the arcs consist of auroral rays and bright spots moving along the arcs and these spatial irregularities may produce the extreme small-scale FACs. This study has shown that under disturbed conditions, FACs forming the arc
current system may reach hundred of μA/m$^2$ on the spatial scale of less than ten kilometers.

Statistically, the bipolar structures dominate in the pre-noon. In the post-midnight MLTs they are observed less frequently. While the interpretation of the bipolar structure in the terms of the meso-scale arc pattern seems reasonable, the small-scale FACs are often a result of reconnection processes distributed over the dayside magnetopause and even in the tail for negative Bz. In contrast to the post-midnight, in the pre-noon sector, where cusp/cleft currents are expected, the bipolar structures are quite frequent. This may be a signature of the plasma injections which are accompanied by pairs of FACs generated due to flux transfer event (FTE) formation (Southwood, 1987) or multiple reconnection at the magnetopause. Magnetic topologies associated with FTEs were previously observed by the MEO satellites (Marchaudon et al., 2004; 2006; Pu et al., 2013). The small-scale field-aligned currents are possibly a consequence of turbulence and instabilities associated with the process of opening previously closed magnetospheric field lines and merging them with the interplanetary magnetic field (Watermann et al., 2009). The regularity presented in **Fig. 8** shows that during the September 2017 magnetic storm the bipolar structures dominate exactly in the region where the signatures of FTEs and the reconnection lines formed at the magnetopause are expected. At the same time, a pair of the most intense FACs is observed on the night side.

## 6 Conclusion

Characteristics of FACs inferred from the 1 Hz *Swarm* observations during the severe magnetic storm of 6-9 September 2017 are presented. This storm is the two-step one with about 22-hr preliminary phase and the intense substorms occurred in the course of the storm main phase. The satellites cross the pre-midnight, pre-noon, pre-dusk and pre-midnight sectors. The following features of the storm-time FACs are found.

Evolution of the current intensities and the latitudinal position of the equatorward boundaries of the FAC region are mainly controlled by a storm-substorm interplay. The FACs become enhanced starting from the SW shock arrival despite of the prolonged period of the northward IMF. The night-time FAC densities primarily follow the substorm development while the dayside FACs are intensified in response to the SW shock and then stay enhanced. At the peak of substorm, the FAC densities averaged over a track within a given MLT sector, reach 3 $\mu A/m^2$, while the undisturbed level is about 0.2 $\mu A/m^2$.

The equatorward displacement of FAC sheets correlates with the storm intensity as monitored by the SYM-H index. The correlation coefficients for the main and recovery phases are about 0.9, while in the course of the main phase the rate of equatorward expansion of FACs is slower than their poleward displacement during the recovery phase. This is likely due to the relatively fast decrease in substorm activity. The minimum latitude of the equatorward FAC boundaries is limited to 49-50° MLat. Although the storm of September 2017 is relatively weak (Dst is about -100 nT), the FAC region expands approximately to the same latitudes as those observed for the much severe storms.

The filamentary structures of high-density FACs are always presented in the *Swarm* observations. A bipolar structure (i.e. the adjacent upward and downward small-scale FACs), ~80 $\mu A/m^2$, 7.5 km width, is observed in the vicinity of the newly developed westward electrojet just prior the substorm onset. Simultaneous plasma perturbations indicate that the FAC pattern is likely associated with mesoscale auroral arc.

*Data availability*: The data used for the publication of this research are freely available from the *Swarm* Science Team web site (ftp://swarm-diss.eo.esa.int). Data selected for the analysis are available upon request (RL).

*Competing interests*: The author declare that she has no conflict of interest concerning this paper.

**Acknowledgement**

*Swarm* data are available through the European Space Agency Online platform (ftp://swarm-diss.eo.esa.int), after registration. We acknowledge the *Swarm* Science Team for providing the level 2 data and the *Swarm* visualization tool (https://vires.services/). The OMNI data on the solar wind, interplanetary magnetic field and geomagnetic indices are obtained from NASA/GSFC's Space Physics Data Facility's CDAweb service (http://omniweb.gsfc.nasa.gov/). This research was supported by the RSF (grant 16-17-00121).

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

**Figure captions**

Figure 1: Polar maps of the SwA and SwB orbits on the northern and southern hemispheres on 6-9 September 2017 in the MLT-Mlat framework. Circles are drawn every 10° down to 50° MLat. Symbol *
and ** indicates, respectively, the entry and exit crossing of the boundary MLat=50°.

Figure 2: From top to bottom: IMF Bz and By, SW speed and density, AL and SYM-H indices on 6-9 September 2017 (5-min values).

Figure 3: (a) 1 s and (b) smoothed FACs measured by SwB in the northern polar region between 23:50 UT, 7 September, and 00:13 UT, 8 September. Downward (upward) current is positive (negative).

Figure 4: Average FAC densities in the four local time sectors covered by the *Swarm* data on 6-9 September, 2017. The left column of plots corresponds to the northern hemisphere (NH) and the right column corresponds to the southern hemisphere (SH). The upper plots (a-d) and the lower plots (e-h) show the dayside and nightside FACs, respectively. The SYM-H and AL indices are added in the plots (a, b) and (e, f), respectively. The centered MLT (10, 16, 22 and 04) is shown in the right upper corner of each plot. The downward and upward FACs (and the corresponding error bars) are shown in red and blue, respectively.

Figure 5: MLat of the FAC equatorward boundaries (EqB) on the Northern (a) and Southern (b) hemispheres for the sectors centered at around 04, 10, 16 and 22 MLT. EqB for each sector is shown by dots of different colors; blue dots representing the nightside (~22 MLT) EqB are connected by a line. The SYM-H and AL index (black line) is added to the upper and lower plot, respectively. The vertical lines mark the beginning of the main and recovery phases.

Figure 6: Correlations between the SYM-H index and the latitudinal position of the nightside (~22 MLT) EqB: black dots and open triangles correspond to the main and recovery phase, respectively.

Figure 7: The largest downward (positive) and upward (negative) 1 s current densities for four MLT sectors on 6-9 September. The vertical solid lines mark the beginning of the storm main phase at 22:00 on 7 September (the time when SYM-H attains its stable negative values <-20 nT; the period of SYM-H<-20 lasts till the end of September 9), the peaks of the first and second major substoms (the time when AL attains its minimum).

Figure 8: Correlations between magnetic latitudes, at which the up- and downward peak FACs are observed: (a) dusk, 16 MLT; (b) pre-noon, 10 MLT; (c) pre-midnight, 22 MLT; (d) post-midnight/early morning, 04 MLT.

Figure 9: The 10 s IL index at 00:00-00:50 UT, September 8. Time of the extreme FAC observation is shown by grey line.

Figure 10: The 1 s values of (a) FAC density, (b) Ne, (c) Te, and (d) Us/c measured along the SwB track at 00:08:00 – 00:12:00 UT, 8 September. In the upper plot the 21-point smoothed FAC density is also shown. Geographic and geomagnetic coordinates are shown on the top.

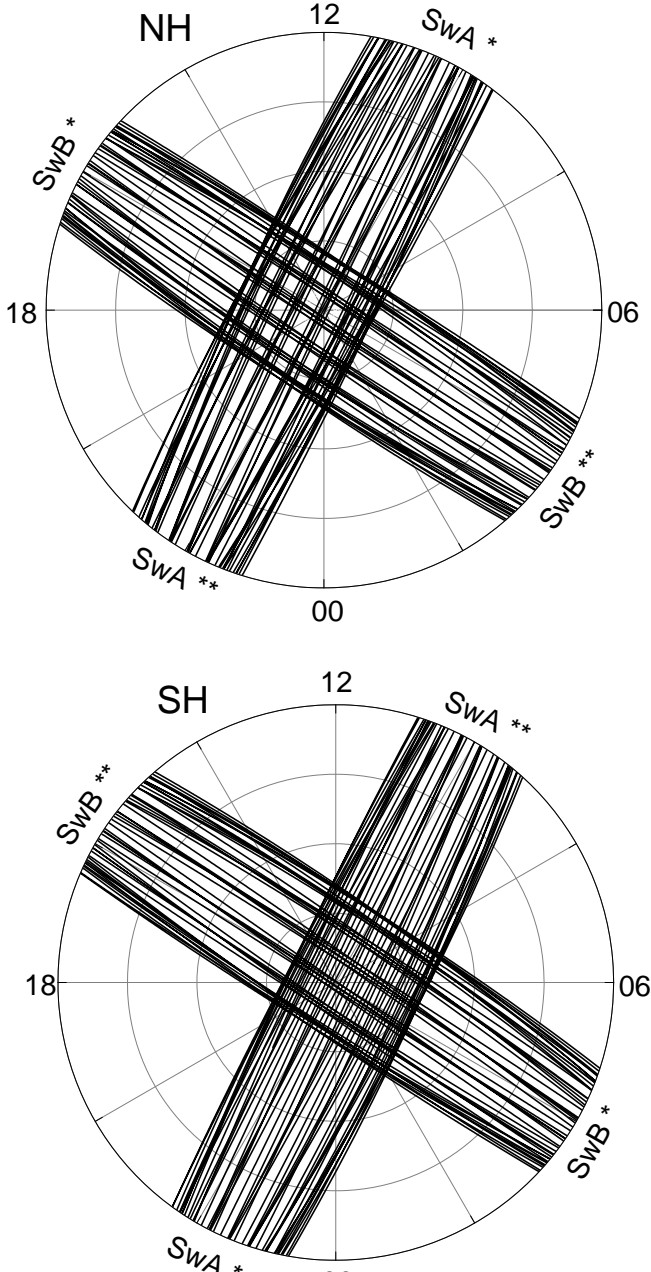

Fig. 1

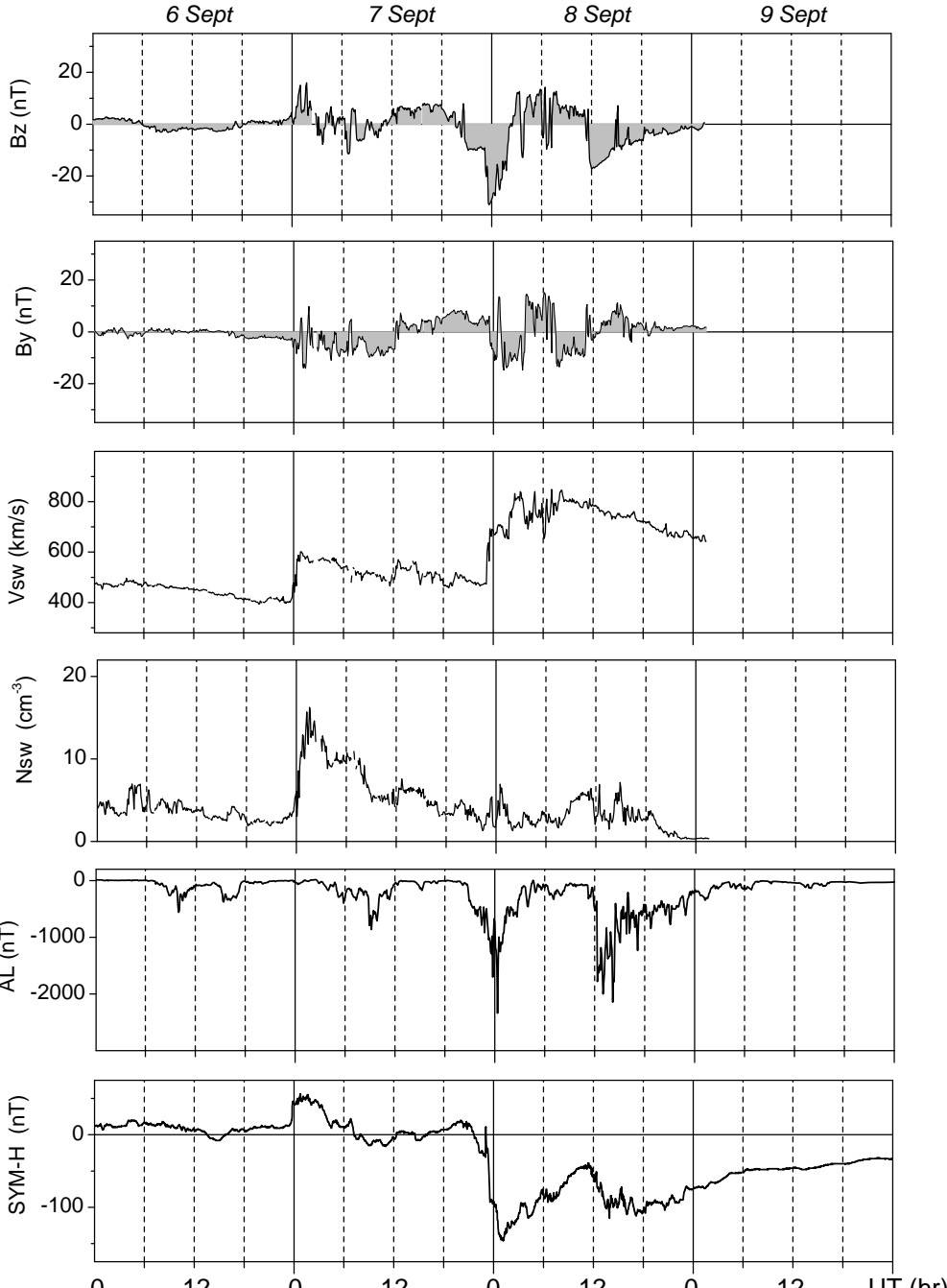

Fig. 2

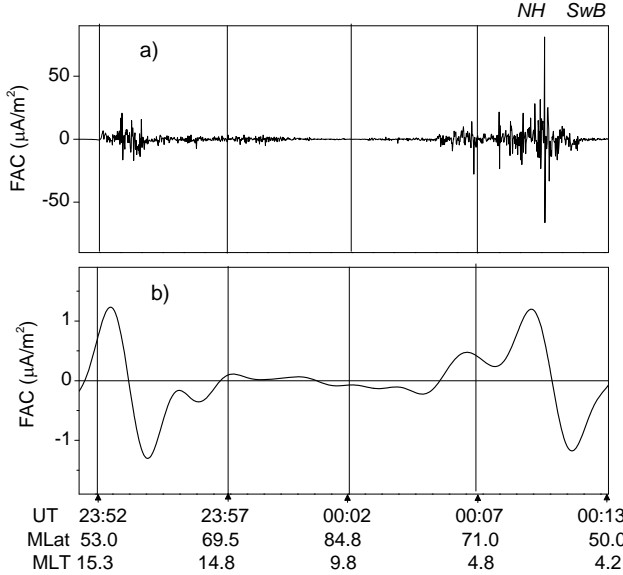

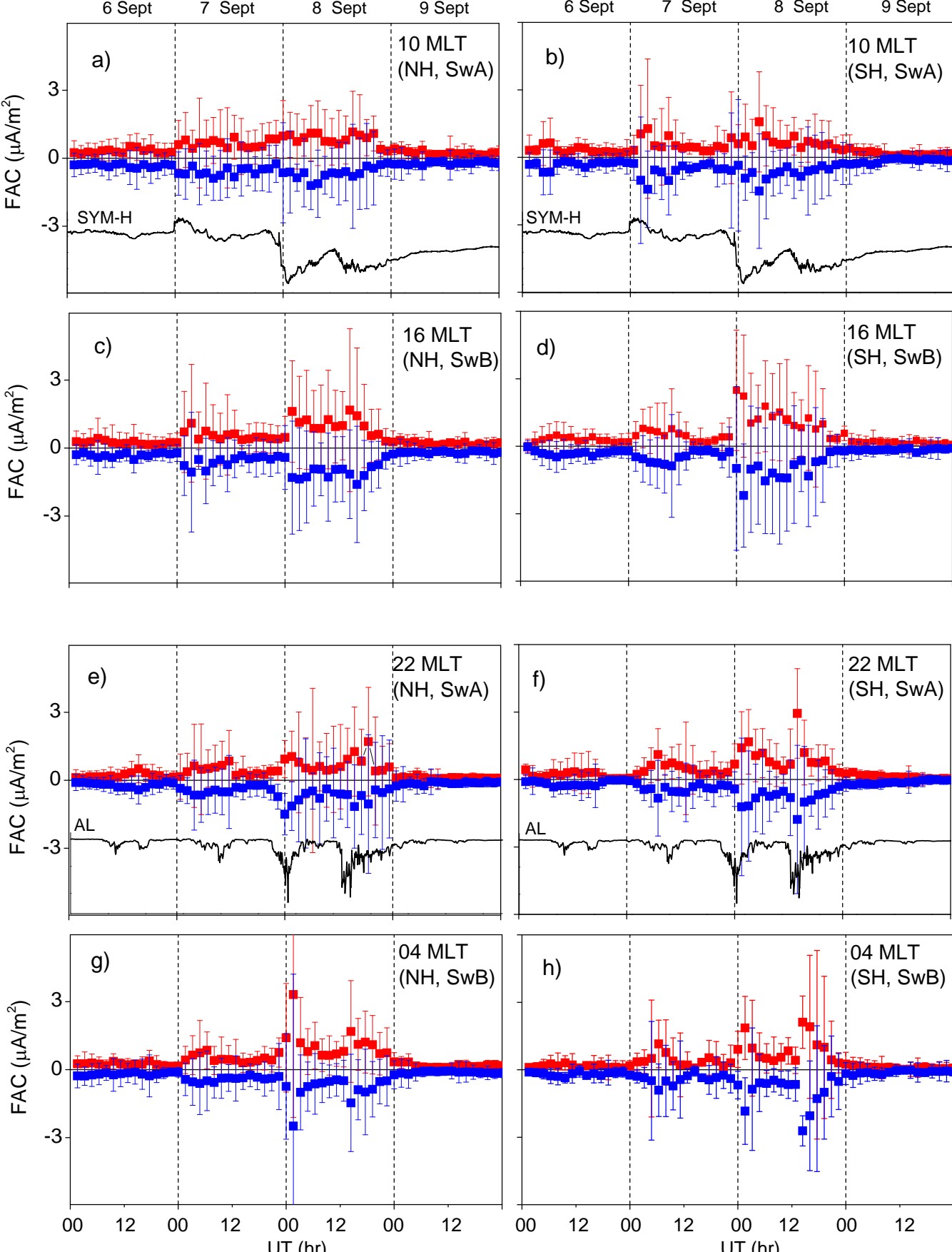

Fig. 4

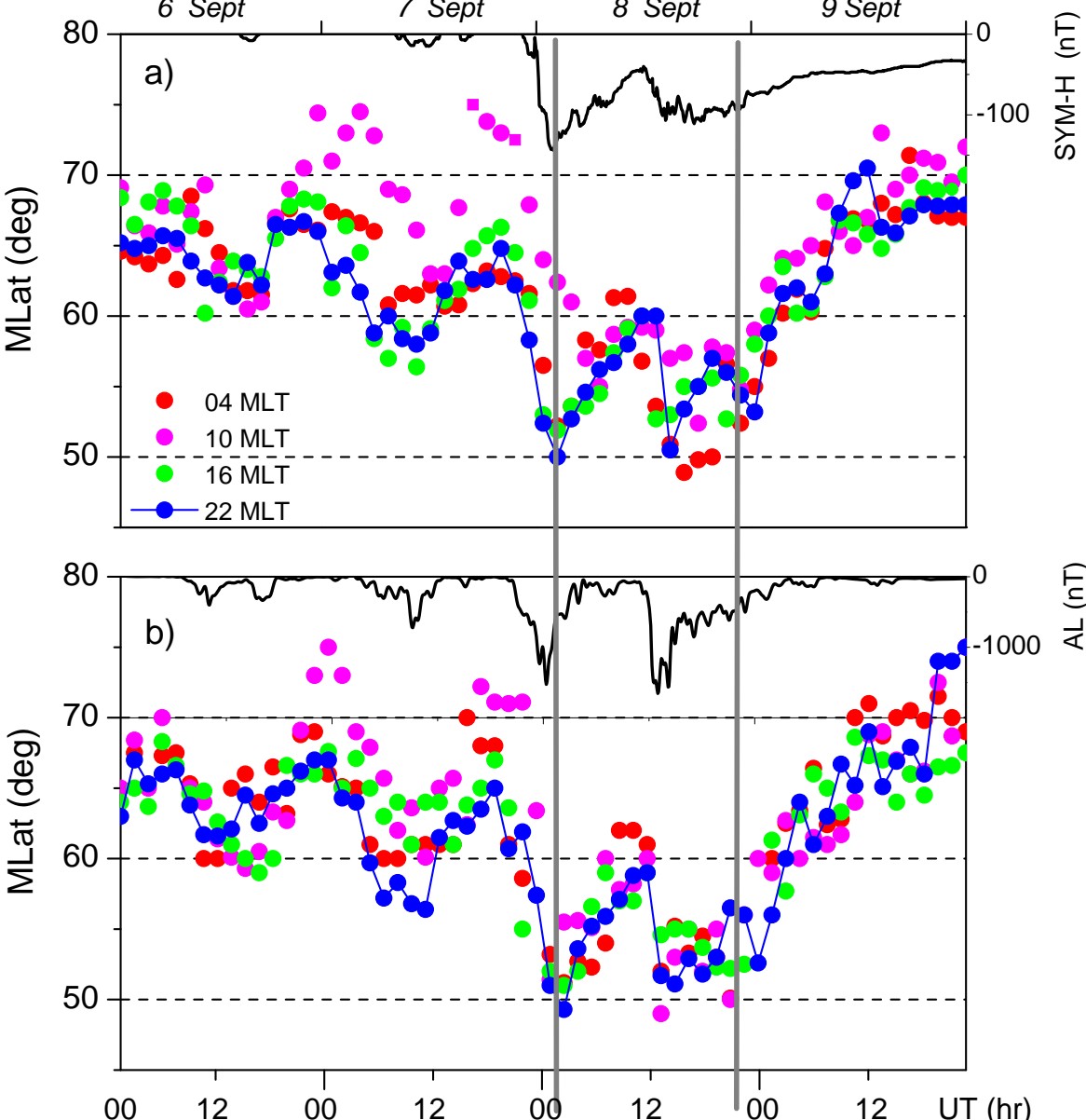

Fig. 5

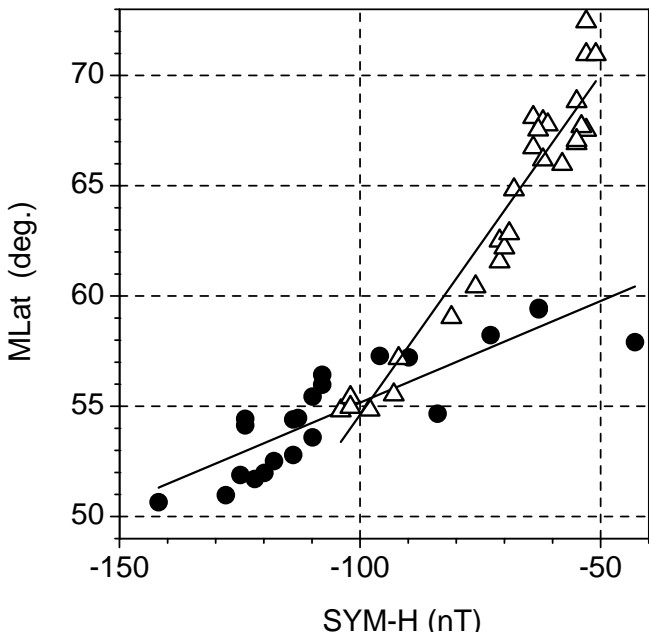

Fig. 6

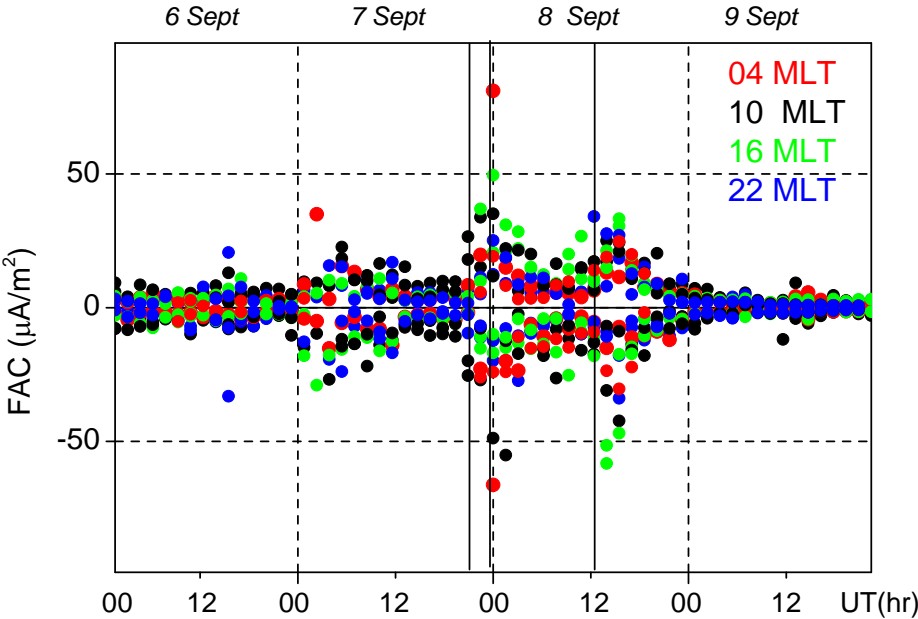

Fig. 7

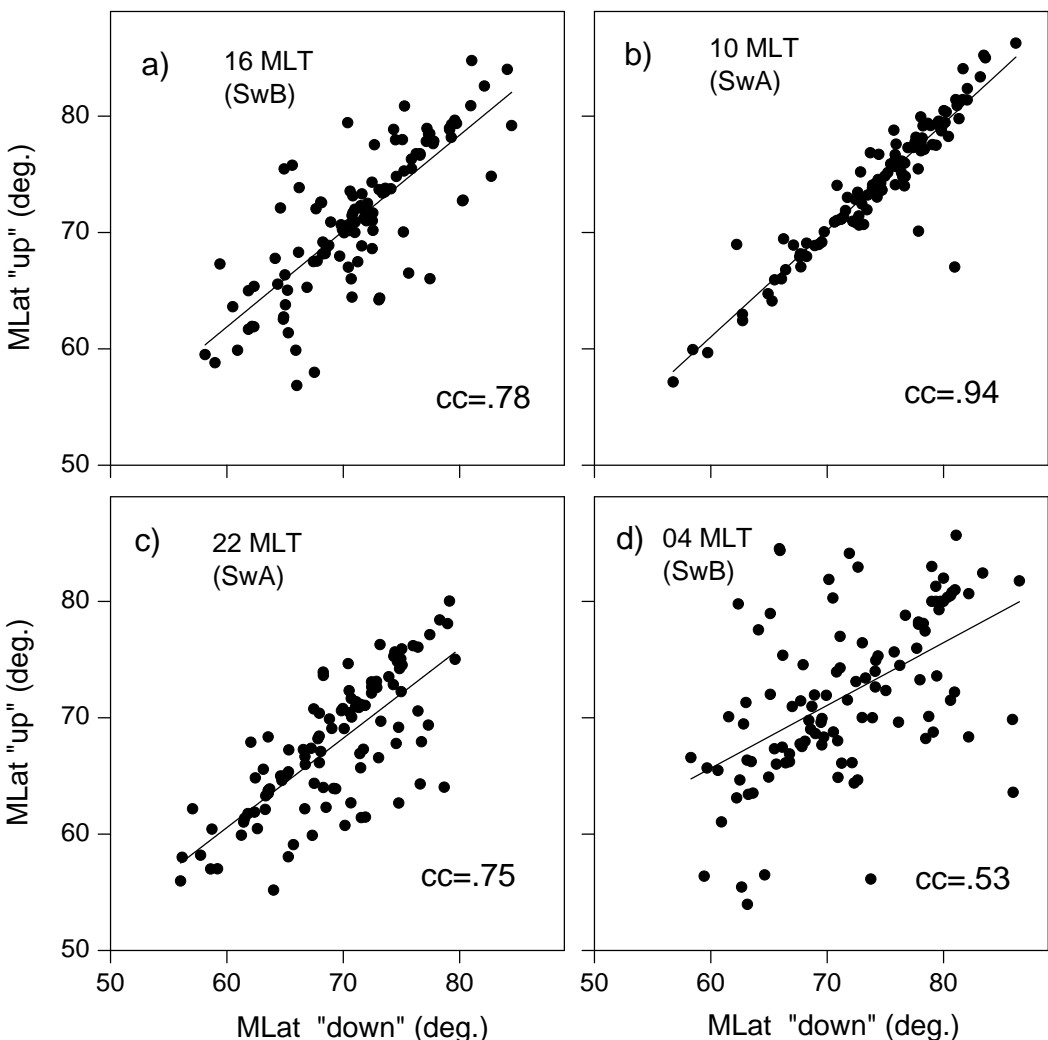

Fig. 8

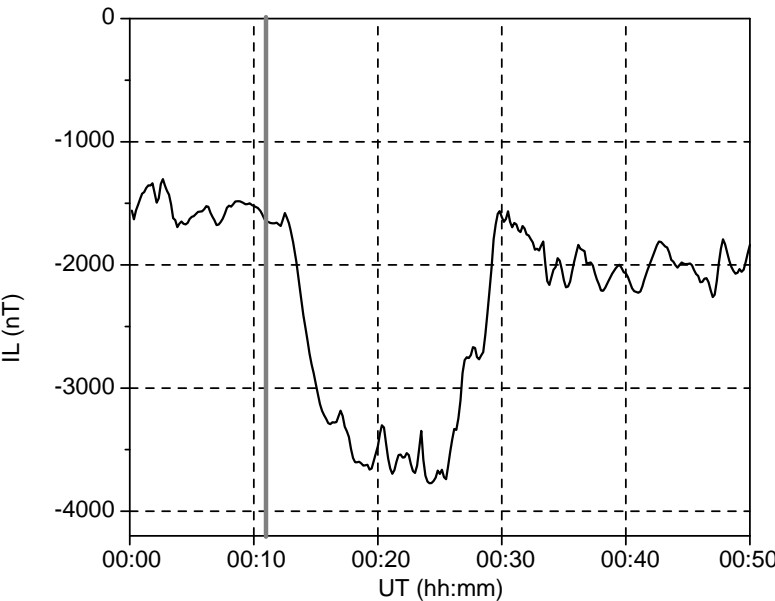

Fig. 9

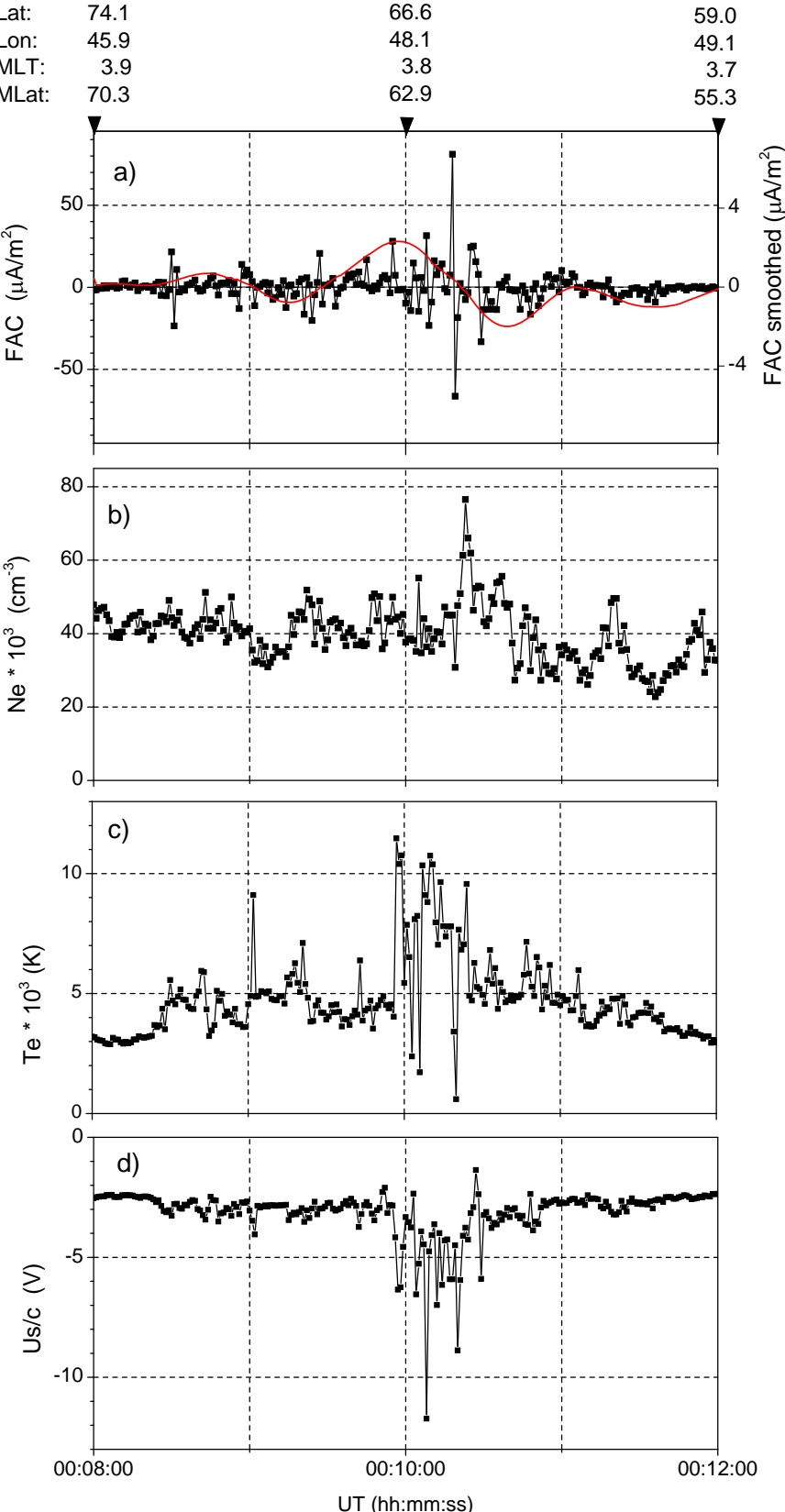

Fig. 10