# Peer review of "Swarm field-aligned currents during a severe magnetic storm of September 2017"

_Annales Geophysicae, 2019_

## Referee Comment (RC1) · Anonymous Referee #1 · 23 Apr 2019

This paper characterizes the field-aligned currents, as described by the single-satellite Swarm data product, during a magnetic storm in September 2017. The paper describes the orbits during the storm, time series of averaged FACs, the FAC equatorial boundaries and the FAC peaks. In addition, a brief description is provided of an event with particularly strong peak values. In some cases the descriptions that are provided in the text are not justified by the data that is presented in the figures. The paper does not present any science questions that it aims to answer, or present any new ideas or discoveries. I therefore recommend that the paper is rejected.

In case the author considers a resubmission, I recommend that the following major issues are addressed:

The introduction consists of two very general paragaraphs, and does not contain any

clear motivation for the study. It is claimed that it is "of interest to analyze [storm FACs'] unique characteristics", but it is not explained why.

The main dataset is the 1 second FAC estimates from Swarm. The assumptions involved in deriving this should be clearly described. There are several alternative techniques which could be mentioned, especially techniques that utilize the Swarm A-C conjunction. How does the results depend on the choice of technique?

There are no references where one can find more information about the instruments (not just the magnetometers, plasma measurements are also used in the paper without proper introduction).

It is claimed in the abstract and in the conclusions that R1/R2 currents are composed of small-scale currents. This is never really shown in the data. How do you know that R1/R2 is not a large-scale current system with small-scale currents superimposed?

The word saturation is used to describe the lower limit of the equatorward boundary. It is never explained what is meant. Reference is made to Xiong et al.'s definition, but the data is never presented in such a way that we can compare with how they define it.

In figures 4, 5, and 6, reference is made to external parameters which are shown with rather coarse resolution in Figure 1. It is very difficult to follow the description when one has to go back and forth between the figures to check. It would help to plot SYM-h together with the panels in 4, 5, 6, and also mark the time of substorm onset.

The time of substorm onset is never mentioned in the paper I believe, and this is quite crucial. For example, it is stated that the FACs propagate equatorward during substorms, but you would expect something different: An expansion during the growth phase, and then a contraction. The way that the figures are presented, it is very difficult to see if this is the case.

I think the description of dawn/dusk asymmetries is an example of not choosing the right tool for the job. If you want to investigate global dawn/dusk asymmetries, why not

use AMPERE, which provides global FAC maps, instead of Swarm which only gives in-situ measurements?

In two cases (l. 25, p. 18 and l. 25 p. 19) reference is made to analyses that are not shown. If it is not shown, it should not be included unless it is completely trivial and easy to check for the reader, which does not seem to be the case here.

———————————————————

---

## Referee Comment (RC2) · Anonymous Referee #2 · 29 Apr 2019

MS angeo-2019-40

Swarm field-aligned currents during a severe magnetic storm of September 2017 by Renata Lukianova

The paper presents the results of the analysis of FAC dynamics during the September 2017 geomagnetic storm as observed by the Swarm satellites at LEO. The analysis is based on the Swarm 1 Hz single satellite FAC product. FACs (density, dawn-dusk asymmetry) observed in four different MLT sectors (midnight, noon, dawn and dusk) are compared both at large and small scales. The movement of the equatorial boundary of the FACs associated with enhanced storm-time and substorm-time geomagnetic activity is also investigated. It is found that the most intense small-scale currents show up during substorms in the post-midnight sector and that the night-time FAC dynamics

is mainly controlled by substorms. Another finding of the paper is the dawn-dusk asymmetry manifested in the enhanced dusk-side downward R2 FACs. Moreover, a bipolar FAC structure is identified prior to a substorm onset that is attributed by the author to a mesoscale auroral arc.

General comments

In general, the paper presents a case study, interesting new observations, however, in some cases it is not clear whether the presented observation and result is new or just confirmation of previous findings. This should be clarified and emphasized in the discussion section. The study of the dawn-dusk asymmetry comparing the dusk-side and dawn-side portions of the same orbit (practically in 1D), especially at small scales, is questionable since this approach ignores the 3D distribution of the current system. Based on the presented figures and tables the calculation of the MLT is very suspicious (see below in the specific comments). Since the MLT information is essential for the whole analysis that may fundamentally affect all the results and conclusions, the paper cannot be accepted before this serious issue is fixed. There are a series of other smaller issues (listed below) that have to be considered before a possible acceptance.

Specific comments

p 3 l 1-10 The description of the storm evolution needs some correction and complementation. Give the time of the shock arrival on 6 Sep. The trigger of the first substorm is first identified as the southward turning Bz at 18:30 on 7 Sep, then a few lines later, as the second shock arriving at 22:00. In contrast with your description, SYM-H was not recovered on 8 Sep.

p 4 l 4 and 6 Description of the Swarm constellation has to be revised. The orbit inclination for SwB is different from that of the other two. As a consequence, the separation between B and (A and C) is increasing. In Sep 2017 it was already close to 6 h (and not 1.5 h) as you can easily check on your Fig. 2. p 4 l 6 The slow drift is in MLT not in longitude. p4 l 7 Since any orbit consists of two parts separated in local time by 12 h,

you only need around 4 and a half (and not 7-10) month of data to cover all local times.

p 4 l 8 The description of the derivation of the FAC product is very inaccurate ("from the measured magnetic field variations, which results from FACs, the current densities are computed. . ."). You may just describe the Swarm single-satellite FAC product mentioning its limitations.

p4 Description of the plasma product is missing.

Figure 2 a) From the polar plot presenting the orbits in the MLT-mlat system, it is suspicious that your MLT calculation is not correct. 2 h change in MLT within a few days is not realistic.

Figure 2 b) I cannot make sense of this figure: daily variation of MLT as a function of MLT. On the x-axis, MLT runs from 0-24. And it is not a straight line!

Your MLT calculation has to be revisited and clearly described in the paper.

Table 1 The same MLT issue.

Section 4.1 FAC densities You may want to rename this section in relation to the title of Section 4.4 'Small-scale FACs'.

Figure 3 b) and p 6 l 12-13 What I see is R1 upward and R2 downward.

p 7 l 5-6 This wording ('determined by averaging the positive (negative) FAC densities from a current free location at the lowest and highest MLat of each crossing') is confusing, rephrase. What is the advantage of using the average densities instead of the 'total' (integrated?) densities? As you mention, the two correlates. Does it mean that the variation of the range of FAC latitudes is not significant?

p 7 l 9 precession?

p 7 l 11-18

p 9 l 7 'The largest FACs are observed' > 'The corresponding/associated FACs are the

largest. . .' (after all the shocks and substorms already mentioned the reader is getting lost)

p 9 l 15 'there is no' > 'we could not find any'

p 10 l 10 The definition of the EqB is not clear: "at least eight values before and after the central point do not exceed 0.1 uA/m^2" Do you mean the smoothed values? Before and after? Central point of what? Estimate the scale of the considered FACs.

p 10 l 19 "considerable" > "moderate"

p 11 l 8 "is seen only . . .. unaffected" > "is the largest . . . less affected"

Figure 5 MLT values given in the figure caption and in the legend are different

p 12 l 6 "resolved spatial scale" > "spatial resolution"

p 13 l 5 and 7 Reference to Fig 7a and 7b are exchanged

p 14 l 10 FFT?? Isn't it just a boxcar smoothing?

p 14 l 17 20000-40000 cm-3 seems a bit low for topside Ne, please confirm

p 14 l 19 Note, that as far as I know, the Swarm Te values are still uncalibrated. If still so, please make a note.

p 15 l 13 ". . . a decrease in Ne (which is usually much less pronounced than a decrease . . ." ?

p 15 l 14 "are created" > "may be created"

p 17 l 24 "is associated" > "is likely associated"

p 18 l 5 "It confirms the fact" If it is a fact, why does it need confirmation? Your statement that large-scale FACs are composed of more intense small-scale FACs is not supported by your analysis. You also mention that others found that small-scale FACs are mostly associated with Alfvén-waves.

p 18 l 15 The scale length range in the brackets is for small-scales?. What scale was taken as a large scale? The given density value (0.5) is for large scale? Please clarify.

p 19 l 3 Is your definition of EqB of large-scale FACs is comparable to that of the cited paper?

p 19 l 15 Image > IMAGE (the name is an acronym)

p 19 l 22 equatorial > equatorward

p 20 l 9 'indication' this asymmetry is well-known, you may say, your observation is in accordance with this.

Technical corrections

p 6 l 11 Aan > An

p 7 l 4 is > are

p 9 l 5 and 7 (also elsewhere) 'in the northern hemisphere', 'in the night side', 'in the day side' > 'on the night side', 'on the dayside', 'on the northern hemisphere'

p 9 l 13 'coherence' > 'correlation'

p 14 l 7 (and elsewhere) 1-sec FAC > 1 s FAC or 1 Hz FAC

p 14 l 21 0.4 and 2% > 0.4% and 2%

---

## Referee Comment (RC3) · Anonymous Referee #3 · 7 May 2019

The paper uses Swarm A, C and B data, mainly the 1-s FAC data product, to study the characteristics of an intense storm during 6-9 September 2017. Evolution of the current intensities and the equatorward displacement of FACs are analyzed while the satellites cross the pre-midnight, pre-noon, dusk and dawn sectors in both hemispheres. The main results given by the author are: (1) The equatorward boundaries of FACs mainly follow the dynamics of ring current (as monitored in terms of the SYM-H index). The minimum latitude of the FAC boundaries is limited to 50° MLat, below which saturation occurs. (2) The FAC densities are very variable and may increase dramatically, especially in the nightside ionosphere during the storm-time substorms. (3) The dawn–dusk asymmetry is manifested in the enhanced dusk-side R2 FACs in both hemispheres. (4) Filamentary high-density structures are always observed confirming that a substantial

fraction of R1/R2 FACs is composed of many small-scale currents.

The main problems with the paper are the following:

- It is not focused

- Many important details, especially related to the data and its analysis, are missing

- Many results given in the abstract are not new

- Throughout the text, many claims are not justified by references

The most interesting topic, which has not been so extensively studied (Wang et al., 2006 have studied one strong storm) is related to point (1). However, this point should be studied more carefully in the paper, since I can only partially agree with the given conclusion. Points (2) and (4) related to variability and existence of filamentary currents are well-known even though it is interesting, that the data product gives such high extreme values for one orbit (up to 80 microA/m2).

In summary, the paper needs such a big major revision that resubmission might be the best choice. However, the present manuscript has clear potential. The revised version should be more focused and the interesting thing would be the general behavior of currents during the different phases of the strong storm and the related substorms. At the moment, the section Discussion uses a lot of time on small-scale current dynamics, which is somewhat de-focusing. The author should pay special attention to references and maybe also search for recent Swarm papers.

Detailed comments:

Introduction:

- The first paragraph is written in a bit loose way, e.g. Are FACs flowing only from boundary layers? Why FAC system is evolved only by dayside (not nightside) reconnection? FAC may exceed its nominal level – what is meant by nominal level?

- In the second paragraph it is said that Wang et al. (2006) and Anderson and Korth (2008) have studied storms, but no results are given

Please swap Sections 2 and 3, it would be more logical

Section 2:

- Maybe the Clilverd et al. (2008) paper should be referred to?

Section 3:

- Here one should shortly explain how 1-s FAC data products are derived from the original magnetometer data

- Please explain what coordinate system for MLAT is used and how MLAT and MLT are derived

- SwB is not separated by 1.5 h in LT from SwA and C, but this difference depends on time

Section 4.1

- First sentence in Sect. 4.1.: give a reference

- Figure 3: Define FAC positive values (up- or downward current)

- Figure 4: It would be more informative for the reader to see in the upper right corner the mean MLT value (or text "pre-noon" etc) than the track identifier. One could also add standard deviations to the mean values by error bars (and expand the horizontal width of the figure)

Section 4.2

- Table 2 would need more explanation. Which MLATs are included in the calculation? What are the uncertainty limits behind these numbers? Has the author checked from the Southern hemisphere, which are the highest MLATs that the satellites reach and does that affect the estimates?

- Line 29: "From the FAC values presented in columns 5 and 6 one can see that in both hemispheres the dusk side downward current is stronger than all the other currents. This predominance implies an additional amplification of the storm-time R2 FAC on the dusk side, which is related to the partial ring current." This would need more discussion and definitely a reference.

Section 4.3

- "pre-storm time". It would be good to define from the beginning, what is the onset of the storm time, and maybe mark that in all the figures.

- l. 18 "Comparing Fig. 1 and Fig. 4 one can see that EqB more closely follows the variation of SYM-H." I agree that since end of Sept 7, the boundaries seem to follow SYM-H, but not before that. Maybe the author could check the correlation to AE-index as well?

Section 4.4

- "The current intensity vary inversely with scale". Please give a reference.

- It is unclear how Figure 7 is composed. What are the horizontal and vertical axes? Is the figure even needed in this paper?

- In this section suddenly Te and Usc are discussed without anywhere properly explained, how it has been derived (which instrument, references etc)

Section 5.1

- "The considerably elevated Te within the arc and just poleward of the arc is associated with a local amplification of electric field." I don't understand this sentence. To my understanding, electric field data is not used in this study. Furthermore, why Te enhancement would be associated with enhanced EF?

Section 5.2

- Reference to Wang et al. (2004) is not found from the list. maybe it should be 2006?

---

## Author Comment (AC1) · 5 Jun 2019

I thank the reviewers for valuable comments and constructive critique. All comments were carefully considered and addressed. Answers to all the questions are presented below. Corresponding changes have been made in the revised manuscript.

The comments and answers are numbered according to the Referee number and the order of comments. The changes in the revised manuscript are indicated in red.

Referee #1

Comment 1.1 The introduction consists of two very general paragraphs, and does not contain any clear motivation for the study. It is claimed that it is "of interest to analyze [storm FACs'] unique characteristics", but it is not explained why.

[Figure]

Reply to Comment 1.1 The introduction has been extended and the motivation for the study has been formulated as follows.

"It is the purpose of this paper to characterize the magnitude of the large- and smaller scale FACs as well as their reaction to the magnetic storm development. The Swarm observations are used in order to identify various characteristics of the storm-time FACs for the event of 6–9 September 2017, which was one of the two most severe magnetic storms of the recent solar cycle 24 (the previous event was the St-Patrick storm on 17 March 2015). The September 2017 event is of particular interest because it was a two-step storm during which two intense substorms occurred and the FAC system is affected by the storm-substorm interplay. In this paper we investigate the time evolution of FAC intensities, the displacement of FAC equatorward boundaries and the extreme small/medium-scale ($\sim$7.5 km width) currents."

Comment 1.2 The main dataset is the 1 second FAC estimates from Swarm. The assumptions involved in deriving this should be clearly described. There are several alternative techniques which could be mentioned, especially techniques that utilize the Swarm A-C conjunction. How does the results depend on the choice of technique?

Reply to Comment 1.2 The Swarm data base contains FACs computed using as the single–satellite as the dual-satellite approaches. In the present study the single-satellite FACs are used because this technique is identical for all three satellites. Since FACs from A, B and C satellites are used, we need the similar method of derivation for all of them. If the dual-satellite method is applied, the resulted FACs are seem to be slightly weaker, than those obtained by the single-satellite method. The following additions explaining the single–satellite as the dual-satellite approaches have been included to section 2.1 Instrumentation.

"FACs can be detected by their magnetic perturbations in the orthogonal plane which are obtained after subtracting the Earth's main magnetic field model from the total measured values. From single spacecraft the FAC density can be estimated based on

one magnetic component with a techniques invoking Ampere's law under assumptions about the infinite current sheet geometry and the orthogonal crossing of the current sheet. This method was used for the previous one-satellite missions, such as Magsat and Ørsted (Christiansen et al., 2002). It is also applied to each Swarm satellite separately. The dual-satellite estimation method calculates current density from curl(B) measured simultaneously at 4 locations was adapted for SwA and SwC data, where measurements separated along-track will be used to create a 'tetrahedron' (Ritter and Lühr, 2006). The curl(B) method provides more reliable current density estimates, as it does not require any assumptions on current geometry and orientation. The FAC output of both the dual-satellite method and the single-satellite method are considered to be in reasonable agreement (Ritter et al., 2013). Both algorithms are implemented in the Level-2 processor to generate the Swarm products that are produced automatically by ESA's processing centre as soon as all input data are available. The products are provided using the dual-satellite method on the lower pair of satellites SwA and SwC, and the single-satellite solution for each of the Swarm spacecraft individually. The 1-sec values of FAC densities are available via the on-line Swarm data portal (ftp://swarm-diss.eo.esa.int) as Level 2 data products (Swarm Level 2 Processing System Consortium, 2012). In the present study the single-satellite FACs are used in order to apply the similar method to SwB and SwA/SwC data."

Comment 1.3 There are no references where one can find more information about the instruments (not just the magnetometers, plasma measurements are also used in the paper without proper introduction).

Reply to Comment 1.3 More information about the magnetometers and the plasma probe along with the appropriate references has been added to section 2.1 Instrumentation. The following para describes the mission instruments and provides references.

"The ESA Swarm mission is a constellation consisting of the three identical satellites (hereafter SwA, SwB and SwC, respectively), all are at the low-altitude polar orbit (Friis-Christensen et al., 2008). …. The main module is the high-sensitivity vector (fluxgate

type) and scalar magnetometers for determining the magnitude and direction of the total vector and variations of the geomagnetic field with an accuracy of more than 0.5 nT (Merayo et al., 2008). Magnetometers make it possible to carry out measurements in a wide range, including the main magnetic field and the variations of external magnetic field generated by FACs. ….….. Each Swarm satellite is also equipped with the Electric Field Instrument which includes the Langmuir plasma probe to provide measurements of electron density, electron temperature and spacecraft potential (Knudsen et al., 2003). These data are available at 2 Hz sampling rate as the standard product of the Swarm data base."

Comment 1.4 It is claimed in the abstract and in the conclusions that R1/R2 currents are composed of small-scale currents. This is never really shown in the data. How do you know that R1/R2 is not a large-scale current system with small-scale currents superimposed?

Reply to Comment 1.4 Figure 3 shows that the R1/R2 are composed of small-scale currents. This figure depicts the original 1-s FACs (small scale) measured along the track, from which large scale currents are revealed after a smoothing procedure.

Comment 1.5 The word saturation is used to describe the lower limit of the equatorward boundary. It is never explained what is meant. Reference is made to Xiong et al.'s definition, but the data is never presented in such a way that we can compare with how they define it.

Reply to Comment 1.5 This word has been eliminated from the abstract and conclusion. As for the reference to Wang et al. (2006), these authors used the definition "saturation" likely in the sense that the limits of the equatorward boundary were observed not lower than at $50°$ MLat.

Comment 1.6 In figures 4, 5, and 6, reference is made to external parameters which are shown with rather coarse resolution in Figure 1. It is very difficult to follow the description when one has to go back and forth between the figures to check. It would

help to plot SYM-h together with the panels in 4, 5, 6, and also mark the time of substorm onset.

Reply to Comment 1.6 SYM-H and AL plots have been added to Fig. 4. SYM-H plots and substorm onset time have been added to Fig. 5. Substorm onset time has been marked on Fig. 6.

Comment 1.7 The time of substorm onset is never mentioned in the paper I believe, and this is quite crucial. For example, it is stated that the FACs propagate equatorward during substorms, but you would expect something different: An expansion during the growth phase, and then a contraction. The way that the figures are presented, it is very difficult to see if this is the case.

Reply to Comment 1.7 Now the time of the two storm-time substorm onsets are indicated in Figs 4-6. As far as an expansion during the growth phase, and then a contraction is concerned, the effect is not seen in this case. It is likely because we deal with the equatorward but not the poleward boundary. It is rather the poleward boundary of FACs that is closely related to the polar cap (open-close) boundary. The poleward boundary is expected to react to the substorm development. But in the parameter presented here, indeed, it is difficult to reveal the effect of expansion/contraction. The following comment has been added to Section 4 Dynamics of the equatorward boundary.

"From Fig. 5 one can see that during the pre-storm time FACs are observed mainly poleward of 60° MLat in both hemispheres. Upon arrival of the SW shock at the very end of September 7, EqB is shifted equatorward, then tends to recover, and then drops again following the second intensification of the storm. At the very beginning of 8 September EqB is found at its lowest position of 50° MLat. The EqB drops abruptly and simultaneously with the peak of the first storm-time substorm and with the lowest drop of SYM-H down to -150 nT. The second substorm reaches its peak slightly before the second minimum of SYM-H (at 12:50 and 13:55, respectively). During the second activation of the storm the EqB is shifted again as low as 50° MLat (although SYM-H

is only -100 nT). As seen from Fig. 5, the evolution of EqB tends to follow the gradual change of SYM-H rather than an abrupt drop of AL related to the substorm onset (see also Fig. 2). Unlike the current density, which exhibits sharp spike-like increases, temporal variations of EqB are relatively smooth. Note that almost no notable difference in evolution in the day- and nightside EqBs is observed during the main and recovery phases. An equatorward expansion of the FAC region during the substorm growth phase, and then a contraction are not resolved."

Comment 1.8 I think the description of dawn/dusk asymmetries is an example of not choosing the right tool for the job. If you want to investigate global dawn/dusk asymmetries, why not use AMPERE, which provides global FAC maps, instead of Swarm which only gives in-situ measurements?

Reply to Comment 1.8 Yes, AMPERE provides global FAC maps which are suitable to study the asymmetry. However, the AMPERE products are not considered here because the present paper concentrates the Swarm data and intends to reveal the storm-time effects solely in these in-situ measurements. Although the picture is not global, the expected asymmetry can be seen. Joint analysis of the AMPERE and Swarm data in order to reveal an asymmetry may be a subject of future work. The appropriate references for the previous AMPERE results are included.

Comment 1.9 In two cases (l. 25, p. 18 and l. 25 p. 19) reference is made to analyses that are not shown. If it is not shown, it should not be included unless it is completely trivial and easy to check for the reader, which does not seem to be the case here.

Reply to Comment 1.9 Following to this comment, in both cases the para referred to the preliminary, not shown analysis has been eliminated.

Please also note the supplement to this comment:
https://www.ann-geophys-discuss.net/angeo-2019-40/angeo-2019-40-AC1-supplement.pdf

---

## Author Comment (AC2) · 5 Jun 2019

I thank the reviewers for valuable comments and constructive critique. All comments were carefully considered and addressed. Answers to all the questions are presented below. Corresponding changes have been made in the revised manuscript.

The comments and answers are numbered according to the Referee number and the order of comments. The changes in the revised manuscript are indicated in red.

Referee #2

General comments In general, the paper presents a case study, interesting new observations, however, in some cases it is not clear whether the presented observation and result is new or just confirmation of previous findings. This should be clarified and Printer-friendly version

emphasized in the discussion section.

The study of the dawn-dusk asymmetry comparing the dusk-side and dawn-side portions of the same orbit (practically in 1D), especially at small scales, is questionable since this approach ignores the 3D distribution of the current system.

Based on the presented figures and tables the calculation of the MLT is very suspicious (see below in the specific comments). Since the MLT information is essential for the whole analysis that may undamentally affect all the results and conclusions, the paper cannot be accepted before this serious issue is fixed. There are a series of other smaller issues (listed below) that have to be considered before a possible acceptance.

**Reply General comments**

All sections have been considerably modified in order to better structuring and emphasis of the new findings.

Yes, the dawn-dusk asymmetry can not be resolved in 1D distribution. This issue was also mentioned in the Referee #1 comments. AMPERE product is more relevant to this kind of analysis (The appropriate references for the previous AMPERE results are included). However, it does not exclude a possibility to reveal signatures of such asymmetry even in the Swarm measurements during a severe magnetic storm. The present paper concentrates the Swarm data and intends to reveal the storm-time effects solely in these in-situ measurements. Although the picture is not global, some signatures of the expected asymmetry can be seen. Joint analysis of the AMPERE and Swarm data in order to reveal an asymmetry may be a subject of future work.

Concerning the MLT problem please see Reply to Comment 2.5

Comment 2.1 p 3 I 1-10 The description of the storm evolution needs some correction and complementation. Give the time of the shock arrival on 6 Sep. The trigger of the first substorm is first identified as the southward turning Bz at 18:30 on 7 Sep, then a few lines later, as the second shock arriving at 22:00. In contrast with your description,
**SYM-H was not recovered on 8 Sep.**

Reply to Comment 2.1 Corrected. The description of the storm evolution has been modified as follows. "The arrival of the first shock late on 6 September (23:50 UT) results in a sudden increase in all parameters except the AL index. Since IMF Bz turns northward, this initial disturbance is only weakly geoeffective as a result. At 20:40 UT on 7 September, IMF Bz turns southward that triggers a substorm growth phase and a ring current build up. The second shock arrived at ~23:40 UT on 7 September, with the SW speed up to 800 km/s and strongly negative Bz and By. This shock causes an abrupt drop of SYM-H down to -150 nT and a spike-like decrease of AL down to -2200 nT. After 03:00, 8 September the IMF Bz becomes positive, AL gradually approaches to zero and SYM-H starts to recover until the next southward turn of Bz. At ~06 UT on 8 September another strongly negative Bz period is seen, and the SW speed remains high (>700 km/s). This causes the second substorm (AL is -2000 nT) and ring current intensification (SYM-H is -100 nT). A steady recovery occurs in the AL index throughout 9 September, while the SYM-H gradually increases from -75 to -35 nT. The SW parameters are not available for this day."

Comment 2.2 p 4 I 4 and 6 Description of the Swarm constellation has to be revised. The orbit inclination for SwB is different from that of the other two. As a consequence, the separation between B and (A and C) is increasing. In Sep 2017 it was already close to 6 h (and not 1.5 h) as you can easily check on your Fig. 2. p 4 I 6 The slow drift is in MLT not in longitude. p4 I 7 Since any orbit consists of two parts separated in local time by 12 h, you only need around 4 and a half (and not 7-10) month of data to cover all local times.

Reply to Comment 2.2 Corrected. The description of the orbits has been modified as follows. "SwA and SwC fly in a tandem separated by  $1-1.4^{\circ}$  in longitude and the maximal differential delay in orbit is  $\sim 3$  s. The orbit period is about 93 min and slightly different between SwA/SwC and the upper satellite SwB, so that their along-orbit separation in local time gradually changes. Their orbital planes also gradually drift apart and
the separation angle increases by  ${\sim}20^{\circ}$  longitude per year. Slowly drifting in longitude, the orbits cover all the local time sectors over about 130 days."

Comment 2.3 p 4 I 8 The description of the derivation of the FAC product is very inaccurate ("from the measured magnetic field variations, which results from FACs, the current densities are computed: : :"). You may just describe the Swarm single-satellite FAC product mentioning its limitations.

Reply to Comment 2.3 The description of the derivation of the FAC product has been modified and extended in order to describe the dual- and single-satellite approach. In the revised version the following additional comments have been included to section 2.1 Instrumentation.

"The main module is the high-sensitivity vector (fluxgate type) and scalar magnetometers for determining the magnitude and direction of the total vector and variations of the geomagnetic field with an accuracy of more than 0.5 nT (Merayo et al., 2008). Magnetometers make it possible to carry out measurements in a wide range, including the main magnetic field and the variations of external magnetic field generated by FACs. FACs can be detected by their magnetic perturbations in the orthogonal plane which are obtained after subtracting the Earth's main magnetic field model from the total measured values. From single spacecraft the FAC density can be estimated based on one magnetic component with a techniques invoking Ampere's law under assumptions about the infinite current sheet geometry and the orthogonal crossing of the current sheet. This method was used for the previous one-satellite missions, such as Magsat and Ørsted (Christiansen et al., 2002). It is also applied to each Swarm satellite separately. The dual-satellite estimation method calculates current density from curl(B) measured simultaneously at 4 locations was adapted for SwA and SwC data, where measurements separated along-track will be used to create a 'tetrahedron' (Ritter and Lühr, 2006). The curl(B) method provides more reliable current density estimates, as it does not require any assumptions on current geometry and orientation. The FAC output of both a dual-satellite and a single satellite method are considered to be in a
reasonable agreement (Ritter et al., 2013). Both algorithms are implemented in the Level-2 processor to generate the Swarm products that are produced automatically by ESA's processing center as soon as all input data are available. The products are provided using the dual-satellite method on the lower pair of satellites SwA and SwC, and the single-satellite solution for each of the Swarm spacecraft individually. The 1-sec values of FAC densities are available via the on-line Swarm data portal (ftp://swarm-diss.eo.esa.int) as Level 2 data products (Swarm Level 2 Processing System Consortium, 2012). In the present study the single-satellite FACs are used in order to apply the similar method to SwB and SwA/SwC data."

Comment 2.4 p4 Description of the plasma product is missing.

Reply to Comment 2.4 Added as follows: "Each Swarm satellite is also equipped with the Electric Field Instrument which includes the Langmuir plasma probe to provide measurements of electron density, electron temperature and spacecraft potential (Knudsen et al., 2003). These data are available at 2 Hz sampling rate as the standard product of the Swarm data base."

Comment 2.5 Figure 2 a) From the polar plot presenting the orbits in the MLT-mlat system, it is suspicious that your MLT calculation is not correct. 2 h change in MLT within a few days is not realistic. Figure 2 b) I cannot make sense of this figure: daily variation of MLT as a function of MLT. On the x-axis, MLT runs from 0-24. And it is not a straight line! Your MLT calculation has to be revisited and clearly described in the paper. Table 1 The same MLT issue.

Reply to Comment 2.5 Figure 1a is seemed to be correct. (Note that in the revised version Figure 1a is Figure 2a because Section "Swarm satellites" and Section "Space weather conditions on 6–9 September 2017" was swapped as recommended by Referee #3.) The polar plots of the orbits do not imply a systematic drift in MLT as large as 2 hr within a few days. While the daily total data coverage is expanded in MLT, it is centered in certain MLT hour (sf, e.g., Cherniak and Zakharenkova Earth, Planets and
Space (2016) 68:136, DOI 10.1186/s40623-016-0506-1).

Figure 1b (former Figure 2b) was erroneous: it contained the erroneous x-axis title. The correct x-axis title is UT.

Projections of the passes are recalculated to the MLT and MLat domain according to coordinate definitions by Laundal and Richmond (2017) as MLT = UT +( $\varphi$  + $\Phi$ N)/15, where  $\varphi$  is the magnetic longitude,  $\Phi$ N is the geographic longitude of the North Centered Dipole pole and UT is the universal time specified in hours.

Comment 2.6 Section 4.1 FAC densities You may want to rename this section in relation to the title of Section 4.4 'Small-scale FACs'.

Reply to Comment 2.6 Section 4.1 can be renamed, however I'd prefer to keep the present title because it is more general and actually both the small (1 Hz) and large (averaged) currents are considered.

Comment 2.7 Figure 3 b) and p 6 I 12-13 What I see is R1 upward and R2 downward.

Reply to Comment 2.7 The description of Fig. 3 has been made more precise. "The 1-s values are presented in Fig. 3a, while Fig.3b depicts the 21-point smoothed curve from which it can be seen that the satellite approaching the pole from the dusk observes first the downward (positive) R2 and then the upward (negative) R1 current. In the near-pole region (above approximately 72° MLat) FACs are almost absent. Then the satellite move equatorward in the early morning local times."

Comment 2.8 p 7 l 5-6 This wording ('determined by averaging the positive (negative) FAC densities from a current free location at the lowest and highest MLat of each crossing') is confusing, rephrase. What is the advantage of using the average densities instead of the 'total' (integrated?) densities? As you mention, the two correlates. Does it mean that the variation of the range of FAC latitudes is not significant?

Reply to Comment 2.8 Fist, the sentence about correlation has been eliminated because it is not confirmed by presentation. The advantage of using the average densiANGEOD
ties instead of the 'total' (summed) densities is related mainly to the different length of the satellite track located within the FAC region.

Comment 2.9 p 7 l 9 precession?

Reply to Comment 2.9 Replaced by "daily variation"

Comment 2.10 p 9 l 7 'The largest FACs are observed' > 'The corresponding/associated FACs are the largest: : :' (after all the shocks and substorms already mentioned the reader is getting lost)

Reply to Comment 2.10 Rephrased to avoid duplicated wording

Comment 2.11 p 9 l 15 'there is no' > 'we could not find any'

Reply to Comment 2.11 Corrected

Comment 2.12 p 10 I 10 The definition of the EqB is not clear: "at least eight values before and after the central point do not exceed 0.1 uA/mËĘ2" Do you mean the smoothed values? Before and after? Central point of what? Estimate the scale of the considered FACs.

Reply to Comment 2.12 To determine the EqB the original 1 Hz data are used. In the sliding window the "central" point moves along the track with 1-s sampling. At each step 10 FACs values preceded and followed the central point are considered. From them at least 8 should be <0.1. If the criterion is fulfilled, the most poleward point without FAC is selected as EqB.

Comment 2.13 p 10 | 19 "considerable" > "moderate" p 11 | 8 "is seen only : : :. unaffected" > "is the largest : : : less affected"

Reply to Comment 2.13 Corrected

Comment 2.14 Figure 5 MLT values given in the figure caption and in the legend are different
Reply to Comment 2.14 Corrected

Comment 2.15 p 12 l 6 "resolved spatial scale" > "spatial resolution" p 13 l 5 and 7 Reference to Fig 7a and 7b are exchanged

Reply to Comment 2.15 Corrected

Comment 2.16 p 14 l 10 FFT?? Isn't it just a boxcar smoothing?

Reply to Comment 2.16 The integrated (build-in) FFT procedure is used. Removed from the text to avoid the ambiguousness.

Comment 2.17 p 14 l 17 20000-40000 cm-3 seems a bit low for topside Ne, please confirm

Reply to Comment 2.17 These values are what is available via the Swarm data portal

Comment 2.18 p 14 l 19 Note, that as far as I know, the Swarm Te values are still uncalibrated. If still so, please make a note.

Reply to Comment 2.18 The data were downloaded from the Swarm data portal. There is no clear indication that they are still uncalibrated. Even if so, it is hardly affect the relatively small-scale perturbations.

Comment 2.19 p 15 l 13 ": : : a decrease in Ne (which is usually much less pronounced than a decrease : : :" ? p 15 l 14 "are created" > "may be created" p 17 l 24 "is associated" > "is likely associated"

Reply to Comment 2.19 Corrected

Comment 2.20 p 18 I 5 "It confirms the fact" If it is a fact, why does it need confirmation? Your statement that large-scale FACs are composed of more intense small-scale FACs is not supported by your analysis. You also mention that others found that small-scale FACs are mostly associated with Alfvén-waves.

Reply to Comment 2.20 These sentences have been refrased as follows: "This implies
that a substantial fraction of R1/R2 currents is composed of many small-scale FACs.

Comment 2.21 p 18 I 15 The scale length range in the brackets is for small-scales?. What scale was taken as a large scale? The given density value (0.5) is for large scale? Please clarify.

Reply to Comment 2.21

In the cited paper the large scale implies >250 km length and the density value (0.5) is for the small scale. Corrected.

Comment 2.22 p 19 I 3 Is your definition of EqB of large-scale FACs is comparable to that of the cited paper?

Reply to Comment 2.22

Wang et al. (2006) defined the latitudinal positions of peak current density but not the most equatoward boundary of the FAC region, thus the actual FAC region may expand to lower latitudes (below the reported 52-56° MLat. A note has been made.

Other comments and Technical corrections p 19 I 15 Image > IMAGE (the name is an acronym) p 19 I 22 equatorial > equatorward p 20 I 9 'indication' this asymmetry is well-known, you may say, your observation is in accordance with this. p 6 I 11 Aan > An p 7 I 4 is > are p 9 I 5 and 7 (also elsewhere) 'in the northern hemisphere', 'in the night side', 'in the day side' > 'on the night side', 'on the dayside', 'on the northern hemisphere' p 9 I 13 'coherence' > 'correlation' p 14 I 7 (and elsewhere) 1-sec FAC > 1 s FAC or 1 Hz FAC p 14 I 21 0.4 and 2% > 0.4% and 2%

**Reply Corrected**

Please also note the supplement to this comment: https://www.ann-geophys-discuss.net/angeo-2019-40/angeo-2019-40-AC2supplement.pdf

---

## Author Comment (AC3) · 5 Jun 2019

I thank the reviewers for valuable comments and constructive critique. All comments were carefully considered and addressed. Answers to all the questions are presented below. Corresponding changes have been made in the revised manuscript.

The comments and answers are numbered according to the Referee number and the order of comments. The changes in the revised manuscript are indicated in red.

Referee #3

Comment 3.1 - The first paragraph is written in a bit loose way, e.g. Are FACs flowing only from boundary layers? Why FAC system is evolved only by dayside (not nightside) reconnection? FAC may exceed its nominal level – what is meant by nominal level?

[Figure]

Reply to Comment 3.1 The first para of Introduction has been rewritten in order to address the issues pointed out by Referee. The revised version reads as follows. "Field-aligned currents (FACs) provide electrodynamic coupling of the solar wind-magnetosphere-ionosphere system. FACs flow along the high-conducting geomagnetic field lines between different magnetospheric domains and the high latitude ionosphere. This current system is driven by the internal magnetospheric circulation of plasma and magnetic field within the global reconnection cycle (Dangey, 1961) and by additional viscous-like interaction at the flanks of magnetosphere (Axford, 1964). Configuration of FACs is primarily controlled by the interplanetary magnetic field (IMF) orientation. Other parameters of the solar wind (velocity, density, IMF strength) and the ionospheric conductivity also play a role (e.g. Christiansen et al., 2002; Ridley 2007; Korth et al., 2002).

Schematic distribution of large-scale FACs has been established by Iijima and Potemra (1976) based on the Triad satellite data. Subsequent space missions allowed constructing comprehensive empirical models of FAC parameterized by the IMF direction and strength, by season, and by hemisphere (Weimer, 2001; Papitashvili et al., 2002; Green at al., 2009). The ionospheric projection of the 3D FAC system consists of a pair of sheets elongated along the auroral oval, namely, Region 1 (R1) and Region 2 (R2), with opposite current flow directions in the morning and evening local time sectors and additional current sheets (R0) located on the dayside poleward of R1/R2. R1 is downward (flows into the ionosphere) and upward (flow from the ionosphere) on the dawn and dusk side, respectively. R1 currents, if reside on closed field lines of the Earth's magnetic field, are believed to originate in either the boundary layer or in the plasma sheet (Ganushkina et al., 2015). R2 FAC is a diversion of the partial ring current to the ionosphere driven by pressure gradients in the inner magnetosphere (Cowley, 2000). R0 current is connected to the dayside magnetopause and its polarity strongly depends on the IMF By component. On the Northern Hemisphere, the R0 current flows predominantly out of the ionosphere for positive IMF By and into the ionosphere for negative IMF By (Iijima et al., 1978; Papitashvili et al., 2002). Additional (NBZ) current associated with the sunward ionospheric flow may appear inside the polar cap, if IMF Bz is northward (Iijima et al., 1984)."

Comment 3.2 In the second paragraph it is said that Wang et al. (2006) and Anderson and Korth (2008) have studied storms, but no results are given

Reply to Comment 3.2 A brief description of the results obtained by the authors cited has been added to the forth para (former 2nd) as follows. "Utilizing the magnetic field measurements by CHAMP satellite Wang et al. (2006) investigated the northern and southern hemisphere dayside and nightside FAC characteristics during the extreme October and November 2003 magnetic storms. It was shown that as Dst decreases, the FAC region expand equatorward, with the shift of FACs on the dayside controlled by the southward IMF. For both case studies, on the southern (late spring) hemisphere the minimum latitude of the FACs is limited to 50° magnetic latitude (MLat) for large negative values of Bz (The minima are the same, although in October the IMF Bz drops dawn to -28 nT, while in November it reaches -50 nT.) On the northern (late autumn) hemisphere the equatorward boundaries of the FAC region are located at 55-60° MLat. Using the global maps from the Iridium constellation Anderson et al. (2005) studied the FACs intensities during severe magnetic storms which occurred during the solar cycle 23 with a particular attention to the evolution of FACs in the course of the storm of August 2000. The results revealed the dawn–dusk asymmetry of the R1/R2 current sheets, with an increase primarily found on the duskside. It was shown that under disturbed conditions the total current is not linearly related to the interplanetary electric field, with the intensity constrained to be below 20 MA (Anderson and Korth, 2007)."

Comment 3.3 Please swap Sections 2 and 3, it would be more logical

Reply to Comment 3.3 Done. Also, Section 2 Swarm satellites has been divided into two subsections: 2.1 Instrumentation (descriptions of the methods used for FAC derivation and the EFI instrument have been added). and 2.2 Orbits on 6-9 September 2017.

Comment 3.4 Section 2: - Maybe the Clilverd et al. (2008) paper should be referred to?

Reply to Comment 3.4 What paper by Clilverd et al. (2008) does the Referee mean? In the 1st para of Section 3 Space weather conditions on 6–9 September 2017 (former Section 2) I refer to several recent papers in which the different effects of this storm were analyzed.

Comment 3.5 Section 3: - Here one should shortly explain how 1-s FAC data products are derived from the original magnetometer data

Reply to Comment 3.5 The description of FAC data products has been added to Section 2 Swarm satellites 2.1 Instrumentation (former Section 3) as follows.

"The mission has a multi-instrument payload. The main module is the high-sensitivity vector (fluxgate type) and scalar magnetometers for determining the magnitude and direction of the total vector and variations of the geomagnetic field with an accuracy of more than 0.5 nT (Merayo et al., 2008). Magnetometers make it possible to carry out measurements in a wide range, including the main magnetic field and the variations of external magnetic field generated by FACs. FACs can be detected by their magnetic perturbations in the orthogonal plane which are obtained after subtracting the Earth's main magnetic field model from the total measured values. From single spacecraft the FAC density can be estimated based on one magnetic component with a techniques invoking Ampere's law under assumptions about the infinite current sheet geometry and the orthogonal crossing of the current sheet. This method was used for the previous one-satellite missions, such as Magsat and Ørsted (Christiansen et al., 2002). It is also applied to each Swarm satellite separately. The dual-satellite estimation method calculates current density from curl(B) measured simultaneously at 4 locations was adapted for SwA and SwC data, where measurements separated along-track will be used to create a 'tetrahedron' (Ritter and Lühr, 2006). The curl(B) method provides more reliable current density estimates, as it does not require any assumptions on current

geometry and orientation. The FAC output of both a dual-satellite and a single satellite method are considered to be in a reasonable agreement (Ritter et al., 2013). Both algorithms are implemented in the Level-2 processor to generate the Swarm products that are produced automatically by ESA's processing center as soon as all input data are available. The products are provided using the dual-satellite method on the lower pair of satellites SwA and SwC, and the single-satellite solution for each of the Swarm spacecraft individually. The 1 s values of FAC densities are available via the on-line Swarm data portal (ftp://swarm-diss.eo.esa.int) as Level 2 data products (Swarm Level 2 Processing System Consortium, 2012). In the present study the single-satellite FACs are used in order to apply the similar method to SwB and SwA/SwC data."

Comment 3.6 - Please explain what coordinate system for MLAT is used and how MLAT and MLT are derived

Reply to Comment 3.6 The location of the satellite is presented in an geographic co-ordinate system NEC (x North, y East, z Center), where the x and y components lie in the horizontal plane, pointing northward and eastward, respectively, and z points to the centre of gravity of the Earth. For the purpose of present study all projections of the passes are recalculated to the magnetic local time (MLT) and MLat domain according to coordinate definitions from Laundal and Richmond (2017). The corrected geomagnetic (CGM) coordinates with the Definite/International Geomagnetic Reference Field (DGRF/IGRF) Models is used.

Comment 3.7 - SwB is not separated by 1.5 h in LT from SwA and C, but this difference depends on time

Reply to Comment 3.7 This previous erroneous statement has been corrected. The corresponding para in Section 2.1 Instrumentation reads as follows. SwA and SwC fly in a tandem separated by 1-1.4° in longitude and the differential delay in orbit is ∼3 s. The orbit period is about 93 min and slightly different between SwA/SwC and the upper satellite SwB, so that their along-orbit separation in local time gradually changes. Their

orbital planes also gradually drift apart and the separation angle increases by ~20° longitude per year. Slowly drifting in longitude, the orbits cover all the local time sectors over about 130 days.

Comment 3.8 - First sentence in Sect. 4.1.: give a reference

Reply to Comment 3.8 The references to (Weimer, 2001; Papitashvili et al., 2002; Green at al., 2009) have been added.

Comment 3.9 - Figure 3: Define FAC positive values (up- or downward current)

Reply to Comment 3.9 Definition has been added to the figure caption: "Downward (upward) current is positive (negative)".

Comment 3.10 Figure 4: It would be more informative for the reader to see in the upper right corner the mean MLT value (or text "pre-noon" etc) than the track identifier. One could also add standard deviations to the mean values by error bars (and expand the horizontal width of the figure)

Reply to Comment 3.10 Figure 4 has been re-plotted. Standard deviations, the centered MLT (instead of the track identifier) and SYM-H and AL indices have been added. Shading has been eliminated to avoid a overloading of the figure. The description of the figure has been modified accordingly. The revised version reads as follows.

"To demonstrate the temporal evolution of FACs in Fig. 4 the FAC intensities for each MLT sector are presented separately for the northern (Fig. 4a, c, e, g) and southern (Fig. 4b, d, f, h) hemispheres. Red (blue) point is determined by averaging the downward (upward) FAC density from a current-free location at the lowest and highest MLat of each crossing. The upper plots (a-d) and the lower plots (e-h) show the data from day side and night side, respectively. For comparison the evolution of the FAC intensities with the storm development the SYM-H and AL indices are shown in the plots (a, b) and (e, f), respectively. Overall, FACs shown in Fig. 4 exhibit three pronounced enhancements, which are of different intensity depending on the MLT sectors. (Note, that

the measured FAC densities do not exhibit any systematic changes associated with the daily variation of the orbit.) The first, smaller enhancement in the very beginning of September 7 is seen on the day side (Fig. 4a-d). This increase of FAC intensity is associated with the SW dynamic pressure front impinges the magnetosphere causing a positive excursion of SYM-H. Unlike the day side, on the nightside FACs (Fig. 4e-h) respond to the shock with a considerable delay, so that FACs are peaked at about 06 UT, 7 September. A moderate substorm occurred in the middle of September 7 also contributes to the increase in FAC intensity.

The two higher peaks occur in the course of the storm main phase at the very beginning and in the middle of September 8. The intensity of a particular peak varies in different MLT sectors being more pronounced on the night side. On the day side, in the sector centered at 10 MLT (Fig. 4a, b) FACs are enhanced during the whole period of 7-8 September with only a relatively modest intensification at about 06 UT on 8 September. In the sector centered at 16 MLT (Fig. 4c, d) an abrupt increase up to 1-2 1 $\mu$A/m2 is observed at the very beginning of 8 September with association with a first deep drop of SYM-H. On the nigh side, a strong increase of FAC is also observed (Fig. 4e-h). However, the current intensities increase more gradually, although finally they reach the higher values. The nightside FACs follow the evolution of AL and start to increase as the substorm growth phase begins at $\sim$22 UT, September 7. In the sector centered at 04 MLT, northern hemisphere (Fig. 4g) a narrow peak up to 3 $\mu$A/m2 is observed in the very beginning of 8 September (for a particular crossing the average density exceeds 6 $\mu$A/m2 as seen from the standard deviation)."

Comment 3.11 Table 2 would need more explanation. Which MLATs are included in the calculation? What are the uncertainty limits behind these numbers? Has the author checked from the Southern hemisphere, which are the highest MLATs that the satellites reach and does that affect the estimates?

Reply to Comment 3.11 MLAT for 40°–90° is accounted. It is difficult to estimate the uncertainty behind the numbers presented in Table 2. The number itself has no uncertainty because this is a result of the straightforward summation. At the same time, as pointed out by the Referee, there may be a lot of indirect factors which may lead as to the under- as to the overestimation. The following additional explanation for Table 2 is suggested. "It should be noted that the numbers presented in Table 2 contain uncertainties. Although these numbers are the result of the straightforward summation, there may be indirect factors which lead as to the under- as to the overestimation. For example, the highest MLATs that the satellites reach mayt affect the estimates. However, if the under- as to the overestimation approximately compensate each other, the tendency of the prevalence of the dawn side R2 is revealed."

Comment 3.12 - Line 29: "From the FAC values presented in columns 5 and 6 one can see that in both hemispheres the dusk side downward current is stronger than all the other currents. This predominance implies an additional amplification of the storm-time R2 FAC on the dusk side, which is related to the partial ring current." This would need more discussion and definitely a reference.

Reply to Comment 3.12 The following addition has been made. "As pointed out by Anderson and Korth (207) this shift may result from a strong dusk side ion pressure leading to asymmetric dusk-side inflation of the magnetic field consistent with a partial, dusk side, ring current during storm main phase (Liemohn et al., 2001)."

Comment 3.13 - "pre-storm time". It would be good to define from the beginning, what is the onset of the storm time, and maybe mark that in all the figures.

Reply to Comment 3.13 The pre-storm time is defined as the time before the SYM-H attains its stable negative values <20 nT at 22:00 on 7 September. For easier comparison of the FAC evolution with the storm phases the SYM-H is shown in Figures 4 and 5. In Figure 6 the time before the SYM-H attains its stable negative values <20 nT at 22:00 on 7 September is marked.

Comment 3.14 - l. 18 "Comparing Fig. 1 and Fig. 4 one can see that EqB more closely follows the variation of SYM-H." I agree that since end of Sept 7, the boundaries seem

to follow SYM-H, but not before that. Maybe the author could check the correlation to AE-index as well?

Reply to Comment 3.14 I do not think that the correlation between AE and EqB would help to resolve the dependence of EqB on any single parameter. For easier comparison the SYM-H index has been added to Fig. 5. More explanation on the SYM-H and EqB coherence has been added to Section 4.3 as the following.

"Even visual comparison of the SYM-H and EqB evolutions reveals a coherent behavior of these two parameters. At the same time, during a period preceding the storm main phase (before end of 7 September, when SYM-H is mainly positive) EqB is located much lower than during the end of recovery phase (after ∼12 UT on 9 September, when SYM-H is still negative). From Fig. 5 one can see that during the pre-storm time (before the SYM-H attains its stable negative values <-20 nT at 22:00 on 7 September) FACs are observed mainly poleward of 60° MLat on both hemispheres. Moderate equatorward shifts of EqB are associated with the modest substorms occurred before the storm main phase in the middle of 6 and 7 September. Prior the storm main phase, on both hemispheres the prenoon (04 MLT) EqB is found considerably poleward compared to the EqB location at other MLTs. The effect is well seen during the two time intervals: from ∼22 UT, Sept 6 till 06 UT, Sept 7 and after 12 UT, Sept 7. Both intervals are dominated by the northward IMF (sf. Fig. 2), so that a shrinking of the polar cap and a poleward shift of the auroral oval is expected. With regard to the position of FACs, the displacement of its equatorward boundary is the largest only in the pre-noon sector, while the other local times remain less affected.

Upon arrival of the SW shock at the very end of September 7, EqB is shifted equatorward, then tends to recover, and then drops again following the second intensification of the storm. At the very beginning of 8 September EqB is found at its lowest position at 50° MLat. The EqB drops abruptly and simultaneously with the peak of the first storm-time substorm just between 7 and 8 September and with the lowest drop of SYM-H down to -150 nT. The second substorm reaches its peak slightly before the second

minimum of SYM-H (at 12:50 and 13:55, respectively). During the second activation of the storm the EqB is shifted again as low as 50° MLat (although SYM-H is only -100 nT). As seen from Fig. 5, the evolution of EqB tends to follow the gradual change of SYM-H rather than an abrupt drops of AL related to the substorm onset (see also Fig. 2 for AL). Unlike the current density, which exhibits sharp spike-like increases, temporal variations of EqB are relatively smooth. During the late recovery phase at the second part of September 9, EqB is shifted poleward as high as 70° MLat. As far as the day-night asymmetry is concerned, almost no notable difference in evolution on the day- and nightside EqBs is observed during the main and recovery phases. An expansion of the FAC region during the substorm growth phase, and then a contraction are not resolved."

Comment 3.15 "The current intensity vary inversely with scale". Please give a reference.

Reply to Comment 3.15 The references to (Neubert and Christiansen, 2003; McGranaghan et al., 2017) have been added.

Comment 3.16 It is unclear how Figure 7 is composed. What are the horizontal and vertical axes? Is the figure even needed in this paper?

Reply to Comment 3.16 When for each crossing of the region of MLat>50° within a certain MLT sector the minimum (i.e. the peak upward current) and maximum (i.e. the peak downward current) of 1 s FACs were selected, it appears that in some cases the adjacent upward and downward currents (call them the bipolar structure) are observed, while in other cases the min/max are separated in latitude. In Fig. 7, the correlations between the MLats, at which the most intense small-scale FACs of opposite polarities are observed, are presented separately for the four MLT sectors. The x-axis (y-axis) corresponds to the MLat at which the downward (upward) 1 s FAC is observed.The x-axis (y-axis) corresponds to the MLat at which the downward (upward) 1 s FAC is observed.

Fig. 7 seems curious because it demonstrates the occurrence of the bipolar structures likely associated with the reconnection formed at the magnetopause. In this relation more explanations have been added as follows.

"In contrast to the early morning local times, in the pre-noon sector, where cusp and cleft currents are expected, the bipolar structures are quite frequent. Here, these structures may represent a signature of the cusp plasma injections which are accompanied by pairs of FACs generated due to flux transfer event (FTE) formation (Southwood, 1987) or localised reconnection at the magnetopause. Magnetic topologies associated with FTEs were previously observed by the MEO satellites (Marchaudon et al., 2004; 2006; Pu et al., 2013) but not by the LEO satellites. Neubert and Christiansen (2003) reported small-scale currents primarily found in the cusp and pre-noon region with densities 1–2 orders of magnitude larger than R1 and R2 currents. The dependence on IMF Bz and the SW turbulence was found by these authors suggesting that currents are a result of reconnection processes distributed over the dayside magnetopause and even in the tail for negative Bz. The bipolar structures were not resolved in the observations by Neubert and Christiansen (2003). The correlations presented in Fig. 7 may be interpreted in such a way that the bipolar structures dominate exactly in the region where the signatures of FTE and the multiple reconnection lines formed at the magnetopause are expected."

Comment 3.17 - In this section suddenly Te and Usc are discussed without anywhere properly explained, how it has been derived (which instrument, references etc)

Reply to Comment 3.17 A brief description of the plasma instrument has been added to Section 2.1 Instrumentation as follows. "Each Swarm satellite is also equipped with the Electric Field Instrument which includes the Langmuir probe to provide measurements of plasma parameters: electron density, electron temperature and spacecraft potential (Knudsen et al., 2003). These data are available at 2 Hz sampling rate as the standard product of the Swarm data base. Combination of the vector magnetometer and the plasma analyzer makes it possible to study the plasma disturbances associated with

FACs."

Comment 3.18 - "The considerably elevated Te within the arc and just poleward of the arc is associated with a local amplification of electric field." I don't understand this sentence. To my understanding, electric field data is not used in this study. Furthermore, why Te enhancement would be associated with enhanced EF?

Reply to Comment 3.18 Yes, the electric field data is not used in this study because these data is unavailable. Thus I can only referee to previous observations, e.g. by Aikio et al. (2002). It is not necessary the Te enhancement would be associated with enhanced EF.

Comment 3.19 Reference to Wang et al. (2004) is not found from the list. maybe it should be 2006?

Reply to Comment 3.19 Wang et al. (2006) is correct.

Please also note the supplement to this comment:
https://www.ann-geophys-discuss.net/angeo-2019-40/angeo-2019-40-AC3-
supplement.pdf
* * *
[Figure]

**Supplement:**

[revised manuscript text omitted]
. Combination of the vector magnetometer and the plasma analyzer makes it possible to study the plasma disturbances associated with FACs. The location of the satellite is presented in an geographic coordinate system NEC (*x* North, *y* East, *z* Center), where the *x* and *y* components lie in the horizontal plane, pointing northward and eastward, respectively, and *z* points to the centre of gravity of

20   the Earth. For the purpose of present study all projections of the passes are recalculated to the magnetic local time (MLT) and MLat domain according to coordinate definitions by Laundal and Richmond (2017).

**2.2 Orbits on 6-9 September 2017**

The polar projection of the satellite orbits as of September 6-9, 2017 on the northern hemisphere is shown in **Fig. 1a**. For mid-September 2017 the projections of the SwB passes are centered in the pre-midnight, pre-noon, dusk and dawn sectors. The successive trajectories (14-15 trajectories per day) are almost parallel to each other and slightly shifted in local time. The satellite SwA (orbits of SwC are very similar) enters the

[revised manuscript text omitted]

To demonstrate the temporal evolution of FACs in **Fig. 4** the FAC intensities for each MLT sector are presented separately for the northern (**Fig. 4a, c, e, g**) and southern (**Fig. 4b, d, f, h**) hemispheres. Red (blue) point is determined by averaging the downward (upward) FAC density from a current-free location at the lowest and highest MLat of each crossing. The upper plots (a-d) and the lower plots (e-h) show the

10   data from day side and night side, respectively. For comparison the evolution of the FAC intensities with the storm development the SYM-H and AL indices are shown in the plots (a, b) and (e, f), respectively. Overall, FACs shown in **Fig. 4,** exhibit three pronounced enhancements, which are of different intensity depending on the MLT sectors. (Note, that the measured FAC densities do not exhibit any systematic changes associated with the daily variation of the orbit.) The first, smaller enhancement in the very

15   beginning of September 7 is seen on the day side (Fig. 4a-d). This increase of FAC intensity is associated with the SW dynamic pressure front impinges the magnetosphere causing a positive excursion of SYM-H. Unlike the day side, on the nightside FACs (Fig. 4e-h) respond to the shock with a considerable delay, so

that FACs are peaked at about 06 UT, 7 September. A moderate substorm occurred in the middle of September 7 also contributes to the increase in FAC intensity.

The two higher peaks occur in the course of the storm main phase at the very beginning and in the middle of September 8. The intensity of a particular peak varies in different MLT sectors being more pronounced on the night side. On the day side, in the sector centered at 10 MLT (Fig. 4a, b) FACs are enhanced during the whole period of 7-8 September with only a relatively modest intensification at about 06 UT on 8 September. In the sector centered at 16 MLT (Fig. 4c, d) an abrupt increase up to 1-2 1 $\mu A/m^2$ is observed at the very beginning of 8 September with association with a first deep drop of SYM-H. On the nigh side, a strong increase of FAC is also observed (Fig. 4e-h). However, the current intensities increase more gradually, although finally they reach the higher values. The nightside FACs follow the evolution of AL and start to increase as the substorm growth phase begins at ~22 UT, September 7. In the sector centered at 04 MLT, northern hemisphere (Fig. 4g) a narrow peak up to 3 $\mu A/m^2$ is observed in the very beginning of 8 September (for a particular crossing the average density exceeds 6 $\mu A/m^2$ as seen from the standard deviation).

The next increase in FACs occurs at ~12 UT on September 8 in association with the second major substorm and the second drop of SYM-H. The corresponding peaks are observed on the night side(the largest average values up to 3 $\mu A/m^2$ are detected on the southern hemisphere (**Fig. 4f, h**), while on the day side the response of FACs to this substorm intensification is weak, if any, although the current densities remain elevated throughout the day. All FACs fall to pre-storm levels by September 9.

Comparison of the evolution of FAC intensity with the SW and geomagnetic parameters during the period of 6-9 September reveals that the storm-time FACs are, on average, by several times larger than the quiet-time ones. Better correlation exists between the night side FACs and the substorm activity (AL index). Although FACs are considerably enhanced during a storm main phase in all MLT sectors, the correlation between the current densities and SYM-H is not strightforward. Also we could not find any simple relation with any isolated SW input, such as the IMF or the SW dynamic pressure.

[Figure]

**Figure 4:** Average FAC densities in the four MLT sectors covered by the *Swarm* data on 6-9 September, 2017. The left column of plots corresponds to the southern (SH) hemisphere and the right corresponds to the northern (NH) hemisphere. The upper plots (a-d) show the dayside data and the lower plots (e-h) show the nightside data. The SYM-H and AL indices are shown in the plots (a,b) and (e, f), respectively. The centered MLT is shown in the right upper corner of each plot. The downward and upward FAC is shown in red and blue, respectively.

**4.2 Dawn-dusk asymmetry**

[revised manuscript text omitted]

Upon arrival of the SW shock at the very end of September 7, EqB is shifted equatorward, then tends to recover, and then drops again following the second intensification of the storm. At the very beginning of 8 September EqB is found at its lowest position at 50° MLat. The EqB drops abruptly and simultaneously with the peak of the first storm-time substorm just between 7 and 8 September and with the lowest drop of SYM-H down to -150 nT. The second substorm reaches its peak slightly before the second minimum of SYM-H (at 12:50 and 13:55, respectively). During the second activation of the storm the EqB is shifted again as low as 50° MLat (although SYM-H is only -100 nT). As seen from **Fig. 5**, the evolution of EqB tends to follow the gradual change of SYM-H rather than an abrupt drops of AL related to the substorm onset (see also **Fig. 2** for AL). Unlike the current density, which exhibits sharp spike-like increases, temporal variations of EqB are relatively smooth. During the late recovery phase at the second part of September 9, EqB is shifted poleward as high as 70° MLat. As far as the day-night asymmetry is concerned, almost no notable difference in evolution on the day- and nightside EqBs is observed during the main and recovery phases. An expansion of the FAC region during the substorm growth phase, and then a contraction are not resolved.

[Figure]

**Figure 5**: Magnetic latitude of the FAC equatorward boundaries on the northern (top) and southern (bottom) hemispheres for four MLT sectors  centered at approximately 04, 10, 16 and 22 MLT (see **Table 1**) on 6-9 September 2017. The SYM-H index is shown by black line in each plot. Three vertical solid lines mark successively the beginning of the storm main phase at 22:00 on 7 September (the time when SYM-H attains its stable negative values <-20 nT), the peaks of the first and second major substoms (the time when AL attains its minimum)

**4.4 Small-scale FACs**

FACs appear on a wide range of scales from large-scale sheet-like currents of hundreds kilometers width to very small-scale filamentary currents of hundreds meters width. The current intensity vary inversely with scale so that large-scale currents are typically a few $\mu A/m^2$, whereas the smaller scale (down to 10 km) are

a few tens μA/m$^2$ (Neubert and Christiansen, 2003; McGranaghan et al., 2017). The standard *Swarm* time-series provide the spatial resolvution of ~7.5 km. The quasi-instantaneous amplitudes of these small scale component of FACs are often much larger than the stationary R1/R2 FACs. To obtain the time-series of peak current densities, the largest positive and negative 1 s values were selected from each crossing of a given MLT time sector irrespective of the hemisphere. The time-series of peak values are presented in **Fig. 6**. From the figure one can see that the small-scale peaks may be more than an order of magnitude larger than the FACs averaged over a track. During the disturbed period, starting with the compression of the magnetosphere on September 7, the amplitude of FACs tends to increase. Two intense substorms occurring during the storm main phase are accompanied by an additional strengthening of small-scale FACs. Both the up- and downward currents strengthen with increased geomagnetic activity and attain their extremes of ~80 μA/m$^2$) near the midnight between September 7 and 8.

[Figure]

**Figure 6**: The largest downward (positive) and upward (negative) 1 s current densities for four MLT sectors on 6-9 September. Three vertical solid lines mark successively the beginning of the storm main phase at 22:00 on 7 September (the time when SYM-H attains its stable negative values <-20 nT), the peaks of the first and second major substorms (the time when AL attains its minimum)

When for each crossing of the region of MLat>50° within a certain MLT sector the minimum (i.e. the peak upward current) and maximum (i.e. the peak downward current) of 1 s FACs were selected, it appears that in some cases the adjacent upward and downward currents (called hereafter the bipolar structure) are observed, while in other cases the min/max are separated in latitude. In **Fig. 7**, the correlations between the MLats, at which the most intense small-scale FACs of opposite polarities are observed, are presented separately for the four MLT sectors. The x-axis (y-axis) corresponds to the MLat at which the downward (upward) 1 s FAC is observed. It is seen that in the pre-noon sector (**Fig. 7b**) the correlation between the latitudinal positions of the up- and downward FACs is high (the correlation coefficient is 0.94). This is indicative of a large population of the paired, closely adjacent small-scale currents of opposite polarity. In the post-noon sector (**Fig. 7a**) the correlation coefficient decreases down to 0.78. Slightly weaker correlation (cc=0.75) is observed in the pre-midnight sector (**Fig. 7c**). At the early morning hours (**Fig. 7d**) the correlation is poor (cc=0.53). Despite some spatiotemporal ambiguity inherent in single satellite observations, it can be speculated that in the early morning local time sector the up- and downward currents appear less frequently in pair but rather are separated by a distance greater than 8 km.

In contrast to the early morning local times, in the pre-noon sector, where cusp and cleft currents are expected, the bipolar structures are quite frequent. Here, these structures may represent a signature of the cusp plasma injections which are accompanied by pairs of FACs generated due to flux transfer event (FTE) formation (Southwood, 1987) or localised reconnection at the magnetopause. Magnetic topologies associated with FTEs were previously observed by the MEO satellites (Marchaudon et al., 2004; 2006; Pu et al., 2013) but not by the LEO satellites. Neubert and Christiansen (2003) reported small-scale currents primarily found in the cusp and pre-noon region with densities 1–2 orders of magnitude larger than R1 and R2 currents. The dependence on IMF Bz and the SW turbulence was found by these authors suggesting that currents are a result of reconnection processes distributed over the dayside magnetopause and even in the tail for negative Bz. The bipolar structures were not resolved in the observations by Neubert and Christiansen (2003). The correlations presented in **Fig. 7** may be interpreted in such a way that 
[revised manuscript text omitted]
. During the period between the substorms, when the AL index is recovered, FACs stay considerably enhanced. Clausen et al. (2013) utilized the data from the Advanced Magnetosphere and Planetary Electrodynamics Response Experiment (AMPERE) to study temporal and spatial dynamics of the R1/R2 currents during isolated substorms. It was shown that the dayside R1 currents are stronger than their nightside counterpart during the substorm growth phase, whereas after

expansion phase onset, the nightside R1 currents dominate. For the second storm-time substorm occurred on 8 September the Swarm observations are in agreement with this regularity. The response of the FAC intensities to the first storm-time substorm is seemed to develop rapidly and the dayside FACs do not exceed the nightside FACs during the growth phase. The effect is likely related to a large amount of energy entered to the magnetosphere from the solar wind due to the viscous-like interaction even if the IMF is northward.

The September 2017 storm is characteristics of a considerable equatorward expansion of the FACs region. For the 2003 storms, the minimum latitude of peak current density are limited to 52-56° MLat (Wang et al., 2006), while for the 2017 storm the EqB expand as low as 50° MLat on both hemispheres. It should be noted that Wang et al. (2006) 
[revised manuscript text omitted]

*Data availability*: The data used for the publication of this research are freely available from the *Swarm* Science Team web site (ftp://swarm-diss.eo.esa.int). Data selected for the analysis are available upon request (RL).

*Competing interests*: The author declare that she has no conflict of interest concerning this paper.

**Acknowledgement**

*Swarm* data are available through the European Space Agency Online platform (ftp://swarm-diss.eo.esa.int), after

5 registration. We acknowledge the *Swarm* Science Team for providing the level 2 data  and the *Swarm* visualization tool (https://vires.services/). The OMNI data on the solar wind, interplanetary magnetic field and geomagnetic indices are obtained from NASA/GSFC's Space Physics Data Facility's CDAweb service (http://omniweb.gsfc.nasa.gov/). This research was partly supported by the RFBR (grant 17-05-00475).

---

## Author Response (AR1)

I thank the reviewers for valuable comments and constructive critique. All comments were carefully considered and addressed. Answers to all the questions are presented below. Corresponding changes have been made in the revised manuscript.

Comments and answers are numbered according to the Referee number and the order of comment. If changes were made in the revised manuscript, their location (page and lines) are indicated in grey shading in the Reply. In the revised manuscript changes made are marked in red. Figs. 4 and 5 have been re-plotted as recommended. Sections 2 and 3 have been swapped.

While revising I tried to make the presentation more clear and consistent. This resulted in some additional changes which are also marked in red (for these changes no locations are provided in Reply) and one additional simple figure (fig.8).

**Referee #1**

**Comment 1.1**
The introduction consists of two very general paragraphs, and does not contain any clear motivation for the study. It is claimed that it is "of interest to analyze [storm FACs'] unique characteristics", but it is not explained why.

**Reply to Comment 1.1**
The introduction has been extended in order to provide more information on the physical processes related to FACs and to place the present study into context. Motivation for the study has been formulated as follows (p. 3, l. 32 - p.4, ll. 1-7).

"It is the purpose of this paper to characterize the magnitude and position of the large- and smaller scale FACs as their response to the magnetic storm development. The *Swarm* observations are used in order to identify various characteristics of the storm-time FACs for the event of 6–9 September 2017, which was one of the two most severe magnetic storms of the recent solar cycle 24 (the previous event was the St-Patrick storm on 17 March 2015). The September 2017 event is of particular interest because it was a two-step storm during which two major substorms occurred and the FAC system is affected by the storm-substorm interplay. In this paper we investigate the time evolution of the large scale FAC intensities, the displacement of the FAC equatorward boundaries and the extreme small scale currents."

**Comment 1.2**
The main dataset is the 1 second FAC estimates from Swarm. The assumptions involved in deriving this should be clearly described. There are several alternative techniques which could be mentioned, especially techniques that utilize the Swarm A-C conjunction. How does the results depend on the choice of technique?

**Reply to Comment 1.2**
The Swarm data base contains FACs computed using as the single-satellite as the dual-satellite approaches. In the present study the single-satellite FACs are used because this technique is identical for all three satellites. Since FACs from A, B and C satellites are used, we need the similar method of derivation for all of them. In general, if the dual-satellite method is applied, the resulted FACs are seem to be slightly weaker, than those obtained by the single-satellite method. The following additions explaining the single-satellite as the dual-satellite approaches have been included to section 2.1 Instrumentation (p. 4, ll. 25-29 - p.5, ll. 1-15).

"FACs are detected by their magnetic perturbations in the orthogonal plane which are obtained after subtracting the main magnetic field model from the total measured values. From single spacecraft the FAC density can be estimated based on one magnetic component with a techniques invoking Ampere's law under assumptions about the infinite current sheet geometry and the orthogonal crossing of the current sheet. This method was used for the previous one-satellite missions, such as Magsat and Ørsted (Christiansen et al., 2002). It is also applied to each *Swarm* satellite separately. The dual-satellite estimation method calculates current density from curl(B) measured quasi-simultaneously at 4 locations is adapted for SwA and SwC data, where measurements separated along-track are used to create a 'tetrahedron' (Ritter and Lühr, 2006). The curl(B) method provides more reliable current density estimates, as it does not require any assumptions on current geometry and orientation. The FAC output of both a dual-satellite and a single satellite methods are considered to be in a reasonable agreement (Ritter et al., 2013). However, a high degree of coherence is typical at auroral latitudes, while in the polar cap the results based ondual-spacecraft technique as more reliable (Luhr et al., 2016). Both algorithms

are implemented to generate the *Swarm* products that are produced automatically by ESA's processing center as soon as all input data are available. The products are provided using the dual-satellite method on the lower pair of satellites SwA and SwC, and the single-satellite solution for each of the Swarm spacecraft individually. The 1 s values (1 Hz sampling rate) of FAC densities are available via the on-line *Swarm* data portal (ftp://swarm-diss.eo.esa.int) as Level 2 data products (Swarm Level 2 Processing System Consortium, 2012). In the present study the single-satellite FACs are used in order to apply the similar method to SwB and SwA/SwC data."

**Comment 1.3**

There are no references where one can find more information about the instruments (not just the magnetometers, plasma measurements are also used in the paper without proper introduction).

**Reply to Comment 1.3**

More information (including references) about the magnetometers and the plasma probe along with the appropriate references has been added to section 2.1 Instrumentation. The following para describes the mission instruments and provides references.

(p. 4, l. 10-11):
"The ESA Swarm mission is a constellation consisting of the three identical satellites (hereafter SwA, SwB and SwC, respectively), all are at the low-altitude polar orbit (Friis-Christensen et al., 2008)."

….

(p. 4, ll. 21-25):
"The mission has a multi-instrument payload. The main module is the high-sensitivity vector (fluxgate type) and scalar magnetometers for determining the magnitude and direction of the total vector and variations of the geomagnetic field with an accuracy of more than 0.5 nT (Merayo et al., 2008). Magnetometers make it possible to carry out measurements in a wide range, including the Earth's main magnetic field and the variations of external magnetic field generated by FACs."

…….

(p.5, ll. 17-22):
"Each satellite is also equipped with the Electric Field Instrument which includes the Langmuir probe to provide measurements of ionospheric plasma parameters: electron density, electron temperature and spacecraft potential (Knudsen et al., 2003). The plasma data are available at 2 Hz sampling rate as the standard product of the *Swarm* data base. Unfortunately, due to technical problems, measurements of the electric field and ions are rather rare. Nevertheless, the combination of data provided by a magnetometer and a plasma analyzer on electrons makes it possible to identify perturbations associated with FACs."

**Comment 1.4**

It is claimed in the abstract and in the conclusions that R1/R2 currents are composed of small-scale currents. This is never really shown in the data. How do you know that R1/R2 is not a large-scale current system with small-scale currents superimposed?

**Reply to Comment 1.4**

Fig. 3 shows that the R1/R2 are composed of small-scale currents. This figure depicts the original 1-s FACs (small scale) measured along the track (Fig. 3a), from which large scale currents (Fig. 3.b) are revealed after a smoothing procedure.

**Comment 1.5**
The word saturation is used to describe the lower limit of the equatorward boundary. It is never explained what is meant. Reference is made to Xiong et al.'s definition, but the data is never presented in such a way that we can compare with how they define it.

**Reply to Comment 1.5**
To avoid the ambiguity the word "saturation" has been eliminated from the abstract and conclusion. As for the reference to Wang et al. (2006), these authors used the definition "saturation" likely in the sense that the limits of the equatorward boundary were observed not lower than at 50° MLat. In the previous version "saturation" was used in the same sense.

**Comment 1.6**
In figures 4, 5, and 6, reference is made to external parameters which are shown with rather coarse resolution in Figure 1. It is very difficult to follow the description when one has to go back and forth between the figures to check. It would help to plot SYM-h together with the panels in 4, 5, 6, and also mark the time of substorm onset.

**Reply to Comment 1.6**
For easier visual comparison the SYM-H and AL have been added to Fig. 4. The SYM-H and substorm onsets have been added to Fig. 5. In Fig. 6 the substorm onsets has been marked.

**Comment 1.7**
The time of substorm onset is never mentioned in the paper I believe, and this is quite crucial. For example, it is stated that the FACs propagate equatorward during substorms, but you would expect something different: An expansion during the growth phase, and then a contraction. The way that the figures are presented, it is very difficult to see if this is the case.

**Reply to Comment 1.7**
Now the time of the two storm-time substorm onsets (AL minima) are indicated in Figs 4-6. As far as an expansion during the growth phase, and then a contraction is concerned, the effect is almost not seen for the event under consideration. It is likely because we deal with the equatorward but not the poleward boundary. It is rather the poleward boundary of FACs that is closely related to the polar cap (open-close) boundary displacement in the course of substorm development. However, in the parameter presented here (the equatoward boundary and the current intensity), it is difficult to reveal the effect of expansion/contraction. The following comment has been added to Section 4 *Dynamics of the equatorward boundary* (p. 15, ll. 16-29 - p.16, ll. 1-2).

"Upon arrival of the SW shock at the very end of September 7, EqB is abruptly shifted equatorward, then tends to recover until the middle of September 8, and then drops again following the second intensification of the storm. At the very beginning of 8 September EqB is found at its lowest position at 50° MLat. A drop of EqB occurs simultaneously with the peak of the first storm-time substorm just between 7 and 8 September and with the lowest SYM-H (below -150 nT). The second substorm reaches its peak slightly before the second minimum of SYM-H (at 12:50 and 13:55, respectively). During this second activation the EqB is shifted again as low as 50° MLat (although SYM-H is only -100 nT). As seen in Fig. 5, the evolution of EqB tends to follow the gradual change of SYM-H rather than an abrupt drops of AL related to the substorm onset (see also Fig. 2 for AL). Unlike the nightside current density, which exhibits several spike-like increases in accordance with AL and the dayside density, which is enhanced

throughout the storm, temporal variations of EqB are relatively smooth and tend to follow the SYM-H ndex. In the end of September 9, during the late recovery phase of the storm, EqB is shifted poleward as high as 70° MLat. As far as the day-night asymmetry is concerned, almost no difference in evolution on the day- and nightside EqBs is observed during the main and recovery phases. Because the storm-time substorms are relatively short-lived, an expansion of the FAC region during the growth phase, and then a contraction are difficult to resolve with the *Swarm* data."

**Comment 1.8**
I think the description of dawn/dusk asymmetries is an example of not choosing the right tool for the job. If you want to investigate global dawn/dusk asymmetries, why not use AMPERE, which provides global FAC maps, instead of Swarm which only gives in-situ measurements?

**Reply to Comment 1.8**
Yes, AMPERE provides global FAC maps which are suitable to study the asymmetry. However, the AMPERE products are not considered here because the present paper concentrates the Swarm data and intends to reveal the storm-time effects solely in these in-situ measurements. Although the picture presented here is not global, the expected asymmetry can be revealed. Joint analysis of the AMPERE and Swarm data in order to reveal an asymmetry and place the Swarm in situ observations in context may be a subject of future work. The appropriate references for the previous AMPERE results are included (p. 25, l. 15-20).

**Comment 1.9**
In two cases (l. 25, p. 18 and l. 25 p. 19) reference is made to analyses that are not shown. If it is not shown, it should not be included unless it is completely trivial and easy to check for the reader, which does not seem to be the case here.

**Reply to Comment 1.9**
Following to this comment, in both cases the text referred to the preliminary, not shown analysis has been eliminated.

**General comments**
In general, the paper presents a case study, interesting new observations, however, in some cases it is not clear whether the presented observation and result is new or just confirmation of previous findings. This should be clarified and emphasized in the discussion section.

The study of the dawn-dusk asymmetry comparing the dusk-side and dawn-side portions of the same orbit (practically in 1D), especially at small scales, is questionable since this approach ignores the 3D distribution of the current system.

Based on the presented figures and tables the calculation of the MLT is very suspicious (see below in the specific comments). Since the MLT information is essential for the whole analysis that may undamentally affect all the results and conclusions, the paper cannot be accepted before this serious issue is fixed. There are a series of other smaller issues (listed below) that have to be considered before a possible acceptance.

**Reply General comments**

All sections have been considerably modified in order to better structuring and emphasis of the new findings.

Yes, the dawn-dusk asymmetry can not be resolved in 1D distribution. This issue was also mentioned in the Referee #1 comments. AMPERE product is more relevant to this kind of analysis. However, it does not exclude a possibility to reveal signatures of such asymmetry even in the instantaneous measurements during a severe magnetic storm. The present paper concentrates the Swarm data and intends to reveal the storm-time effects solely in these in-situ measurements. Although the picture is not global, some signatures of the expected asymmetry can be revealed. Joint analysis of the AMPERE and Swarm data can be a subject of future work.

Concerning the MLT problem please see Reply to Comment 2.5

**Comment 2.1**
p 3 l 1-10 The description of the storm evolution needs some correction and complementation. Give the time of the shock arrival on 6 Sep. The trigger of the first substorm is first identified as the southward turning Bz at 18:30 on 7 Sep, then a few lines later, as the second shock arriving at 22:00. In contrast with your description, SYM-H was not recovered on 8 Sep.

**Reply to Comment 2.1**
The description of the storm evolution has been modified as follows (p. 7, ll. 15-18 – p.8. ll. 1-8). "Two SW shock events impact the magnetosphere. The arrival of the first shock late on 6 September (23:50 UT) results in a sudden increase in all parameters except the AL index. Since IMF Bz turns northward, this initial disturbance is only weakly geoeffective as a result. At 20:40 UT, 7 September, IMF Bz turns southward that triggers a substorm growth phase and a ring current build up. The second shock arrived at ~23:40 UT on 7 September, with the SW speed up to 800 km/s and strongly negative Bz and By. This shock causes an abrupt drop of SYM-H down to -150 nT and a spike-like decrease of AL down to -2200 nT. After 03 UT, 8 September, the IMF Bz becomes positive, AL gradually approaches to zero and SYM-H starts to recover until the next southward turn of Bz. At ~06 UT on 8 September another strongly negative Bz period is seen, and the SW speed remains high (>700 km/s). This causes the second substorm (AL is -

2000 nT) and ring current intensification (SYM-H is -100 nT). A steady recovery occurs in the AL index throughout 9 September, while the SYM-H gradually increases from -75 to -35 nT. The SW parameters are not available for this day."

**Comment 2.2**
p 4 l 4 and 6 Description of the Swarm constellation has to be revised. The orbit inclination for SwB is different from that of the other two. As a consequence, the separation between B and (A and C) is increasing. In Sep 2017 it was already close to 6 h (and not 1.5 h) as you can easily check on your Fig. 2. p 4 l 6 The slow drift is in MLT not in longitude. p4 l 7 Since any orbit consists of two parts separated in local time by 12 h, you only need around 4 and a half (and not 7-10) month of data to cover all local times.

**Reply to Comment 2.2**
Corrected. In section 2.1 the description of the orbits has been modified as follows (p. 4, ll. 14-19).
"SwA and SwC fly in a tandem separated by 1-1.4° in longitude and the differential delay in orbit is ~3 s. The orbit period is about 93 min and slightly different between SwA/SwC and the upper satellite SwB, so that their along-orbit separation in local time gradually changes. Their orbital planes also gradually drift apart and the separation angle increases by ~20° longitude per year. Slowly drifting in longitude, the orbits cover all the local time sectors over about 130 days."

**Comment 2.3**
p 4 l 8 The description of the derivation of the FAC product is very inaccurate ("from the measured magnetic field variations, which results from FACs, the current densities are computed: : :"). You may just describe the Swarm single-satellite FAC product mentioning its limitations.

**Reply to Comment 2.3**
The description of the derivation of the FAC product has been modified and extended in order to describe the dual- and single-satellite approach. In the revised version the following additional comments have been included to section 2.1 Instrumentation (p. 4, ll. 21-29 – p.5, ll. 1-15).

"The mission has a multi-instrument payload. The main module is the high-sensitivity vector (fluxgate type) and scalar magnetometers for determining the magnitude and direction of the total vector and variations of the geomagnetic field with an accuracy of more than 0.5 nT (Merayo et al., 2008). Magnetometers make it possible to carry out measurements in a wide range, including the Earth's main magnetic field and the variations of external magnetic field generated by FACs. FACs are detected by their magnetic perturbations in the orthogonal plane which are obtained after subtracting the main magnetic field model from the total measured values. From single spacecraft the FAC density can be estimated based on one magnetic component with a techniques invoking Ampere's law under assumptions about the infinite current sheet geometry and the orthogonal crossing of the current sheet. This method was used for the previous one-satellite missions, such as Magsat and Ørsted (Christiansen et al., 2002). It is also applied to each *Swarm* satellite separately. The dual-satellite estimation method calculates current density from curl(B) measured quasi-simultaneously at 4 locations is adapted for SwA and SwC data, where measurements separated along-track are used to create a 'tetrahedron' (Ritter and Lühr, 2006). The curl(B) method provides more reliable current density estimates, as it does not require any assumptions on current geometry and orientation. The FAC output of both a dual-satellite and a single satellite methods are considered to be in a reasonable agreement (Ritter et al., 2013). However, a high degree of coherence is typical at auroral latitudes, while in

the polar cap the results based ondual-spacecraft technique as more reliable (Luhr et al., 2016). Both algorithms are implemented to generate the *Swarm* products that are produced automatically by ESA's processing center as soon as all input data are available. The products are provided using the dual-satellite method on the lower pair of satellites SwA and SwC, and the single-satellite solution for each of the Swarm spacecraft individually. The 1 s values (1 Hz sampling rate) of FAC densities are available via the on-line *Swarm* data portal (ftp://swarm-diss.eo.esa.int) as Level 2 data products (Swarm Level 2 Processing System Consortium, 2012). In the present study the single-satellite FACs are used in order to apply the similar method to SwB and SwA/SwC data."

**Comment 2.4**
p4 Description of the plasma product is missing.

**Reply to Comment 2.4**
Added as follows (p. 5, ll.17-22):
"Each satellite is also equipped with the Electric Field Instrument which includes the Langmuir probe to provide measurements of ionospheric plasma parameters: electron density, electron temperature and spacecraft potential (Knudsen et al., 2003). The plasma data are available at 2 Hz sampling rate as the standard product of the *Swarm* data base. Unfortunately, due to technical problems, measurements of the electric field and ions are rather rare. Nevertheless, the combination of data provided by a magnetometer and a plasma analyzer on electrons makes it possible to identify perturbations associated with FACs."

**Comment 2.5**
Figure 2 a) From the polar plot presenting the orbits in the MLT-mlat system, it is suspicious that your MLT calculation is not correct. 2 h change in MLT within a few days is not realistic.
Figure 2 b) I cannot make sense of this figure: daily variation of MLT as a function of MLT. On the x-axis, MLT runs from 0-24. And it is not a straight line!
Your MLT calculation has to be revisited and clearly described in the paper.
Table 1 The same MLT issue.

**Reply to Comment 2.5**
Figure 1 is seemed to be correct. (Note that in the revised version Fig. 1 is Fig. 2 because Section "Swarm satellites" and Section "Space weather conditions on 6–9 September 2017" was swapped as recommended by Referee #3. Also, Fig. 2a has been eliminated because the same information on the orbits is presented in Table 1). During the day, the orbits are systematically shifted almost parallel to each other, however, in auroral latitudes, they are within the early in the morning, pre-noon, pre-dawn and pre-midnight sectors. (Projection of Swarm satellite passes recalculated to the MLT–MLAT domain during the full day is presented in, e.g., Cherniak and Zakharenkova Earth, Planets and Space (2016) 68:136, DOI 10.1186/s40623-016-0506-1; their Fig.1).

Figure 1b (former Figure 2b, eliminated in the revised version) was erroneous. The correct x-axis title is UT.

MLT was not calculated, it was taken from the Swarm VirES tool. The following comment has been added (p.5, ll. 23-27):
"In each Level 2 data file the location of the satellite is presented in an geographic coordinate system NEC (*x* - North, *y* - East, *z* - Center), where the *x* and *y* components lie in the horizontal plane, pointing northward and eastward, respectively, and *z* points to the centre of gravity of the

Earth. For the purpose of present study all projections of the passes are shown in the magnetic local time (MLT) and MLat domain. The coordinates are available via the on-line Swarm Data Visualisation Tool (VirES)."

**Comment 2.6**
Section 4.1 FAC densities You may want to rename this section in relation to the title of Section 4.4 'Small-scale FACs'.

**Reply to Comment 2.6**
I'd prefer to keep the present title because it is more general and actually both the small (1 Hz) and large (averaged) currents are considered.

**Comment 2.7**
Figure 3 b) and p 6 l 12-13 What I see is R1 upward and R2 downward.

**Reply to Comment 2.7**
The description of Fig. 3 has been made more precise and modified as follows (p. 9, ll. 19-25):
"Fig. 3b depicts the 21-point smoothed curve. It can be seen that the satellite approaching the pole from the dusk observes first the downward (positive) R2 and then the upward (negative) R1 current, both are of ~2 $\mu A/m^2$ density. Above approximately 70° MLat FACs become marginal. When the satellite moves equatorward in the early morning local times, a multi-layer structure is observed, in which the poleward currents are mostly positive, so they may be associated with downward the R1 FAC. The most equatorward currents are negative and thus represent the R2 FAC."

**Comment 2.8**
p 7 l 5-6 This wording ('determined by averaging the positive (negative) FAC densities from a current free location at the lowest and highest MLat of each crossing') is confusing, rephrase. What is the advantage of using the average densities instead of the 'total' (integrated?) densities? As you mention, the two correlates. Does it mean that the variation of the range of FAC latitudes is not significant?

**Reply to Comment 2.8**
The confusing description has been rephrased. In the revised version it sounds as follows (p.10, ll. 6-9):
"To demonstrate the global temporal evolution of FACs, in Fig. 4 the current intensities for the four MLT sectors are presented separately for the northern (Fig. 4 a, c, e, g) and southern (Fig. 4 b, d, f, h) hemispheres. Each red (blue) point is determined by averaging the downward (upward) current densities when the satellite crosses the region filled with FACs."

The advantage of using the average densities instead of the 'total' (summed) densities is related mainly to the different length of the satellite track within the region filled by FAC. Although the average and total densities closely correlate, the correlation is not perfect. This is because the variation of the range of FAC latitudes affects the average density in lesser degree.

**Comment 2.9**
p 7 l 9 precession?

**Reply to Comment 2.9**
The word "precession" has been eliminated. The daily variation of the polar projection of the satellite orbit is described in more detail as follows (p. 6, ll. 7-9):
"During a day, the successive projections are systematically shifted almost parallel to each other, however, in auroral latitudes, they stay mainly within the same sectors."

Also, Table 1 has been modified: MLT range within which the satellite crosses the boundary of 50° (70°) MLat has been added. Fig. 1b has been eliminated because it duplicates Table 1.

**Comment 2.10**
p 9 l 7 'The largest FACs are observed' > 'The corresponding/associated FACs are the largest: : :' (after all the shocks and substorms already mentioned the reader is getting lost)

**Reply to Comment 2.10**
The description of Fig. 4 has been considerably modified for easier comparison and explanation of the storm/substorm development and FAC evolution. AL and SYM-H indices have been added to the figure. In the revised version the description of Fig. 4 does not contain a confused phrase mentioned in the comment.

**Comment 2.11**
p 9 l 15 'there is no' > 'we could not find any'

**Reply to Comment 2.11**
Corrected

**Comment 2.12**
p 10 l 10 The definition of the EqB is not clear: "at least eight values before and after the central point do not exceed 0.1 uA/m^2" Do you mean the smoothed values? Before and after? Central point of what? Estimate the scale of the considered FACs.

**Reply to Comment 2.12**
I agree that the description of the procedure of EqB determination was confusing. This was likely because in the previous version the sliding window moved along the track in accordance with the actual satellite track, i.e., entering or leaving the FAC region. In the first case 1-second values "after" the central point of the sliding window should be close to zero. In the second case, values "before" the central point should be close to zero.

In the revised version more simple procedure has been applied. The description has been modified accordingly (p. 14, ll. 14-18):
"The EqB is determined as the lowest MLat at which FACs are terminated. The procedure of the 20-point sliding window (the scale is about 150 km) moving along a track from the equator to the pole is applied to the 1 s FAC values and the corresponding MLats. EqB is selected as the lowest MLat of the window if 90% of FAC values within the window exceed $|0.1|$ $\mu A/m^2$."

**Comment 2.13**
p 10 l 19 "considerable" > "moderate"
p 11 l 8 "is seen only : : :. unaffected" > "is the largest : : : less affected"

**Reply to Comment 2.13**
Corrected

**Comment 2.14**
Figure 5 MLT values given in the figure caption and in the legend are different

**Reply to Comment 2.14**
Corrected

**Comment 2.15**
p 12 l 6 "resolved spatial scale" > "spatial resolution"
p 13 l 5 and 7 Reference to Fig 7a and 7b are exchanged

**Reply to Comment 2.15**
Corrected

**Comment 2.16**
p 14 l 10 FFT?? Isn't it just a boxcar smoothing?

**Reply to Comment 2.16**
A boxcar (build-in to the Mathlab/Origin) FFT procedure is used. "FFT" is removed from the text.

**Comment 2.17**
p 14 l 17 20000-40000 cm-3 seems a bit low for topside Ne, please confirm

**Reply to Comment 2.17**
These values are what is available via the Swarm data portal

**Comment 2.18**
p 14 l 19 Note, that as far as I know, the Swarm Te values are still uncalibrated. If still so, please make a note.

**Reply to Comment 2.18**
The data were downloaded from the Swarm data portal. There is no clear indication that they are still uncalibrated. Even if so, it is hardly affect the relatively small-scale perturbations.

**Comment 2.19**
p 15 l 13 ": : : a decrease in Ne (which is usually much less pronounced than a decrease : : :" ?
p 15 l 14 "are created" > "may be created"
p 17 l 24 "is associated" > "is likely associated"

**Reply to Comment 2.19**
Corrected

**Comment 2.20**
p 18 l 5 "It confirms the fact" If it is a fact, why does it need confirmation? Your statement that large-scale FACs are composed of more intense small-scale FACs is not supported by your analysis. You also mention that others found that small-scale FACs are mostly associated with Alfvén-waves.

**Reply to Comment 2.20**
These sentences have been refrased as follows: "This implies that a substantial fraction of R1/R2 currents is composed of many small-scale FACs." (p. 26, ll. 4-5).

**Comment 2.21**
p 18 l 15 The scale length range in the brackets is for small-scales?. What scale was taken as a large scale? The given density value (0.5) is for large scale? Please clarify.

**Reply to Comment 2.21**

Corrected (p.26, ll. 16-17). In the cited paper the large scale implies >250 km length and the density value (0.5) is for the small scale.

**Comment 2.22**
p 19 l 3 Is your definition of EqB of large-scale FACs is comparable to that of the cited paper?

**Reply to Comment 2.22**

Wang et al. (2006) defined the latitudinal positions of peak current density but not the most equatoward boundary of the FAC region, thus the actual FAC region may expand to lower latitudes (below the reported 52-56° MLat). A note has been made (p. 24, ll. 4-6).

**Other comments and Technical corrections**
p 19 l 15 Image > IMAGE (the name is an acronym)
p 19 l 22 equatorial > equatorward
p 20 l 9 'indication' this asymmetry is well-known, you may say, your observation is in accordance with this.
p 6 l 11 Aan > An
p 7 l 4 is > are
p 9 l 5 and 7 (also elsewhere) 'in the northern hemisphere', 'in the night side', 'in the day side' > 'on the night side', 'on the dayside', 'on the northern hemisphere'
p 9 l 13 'coherence' > 'correlation'
p 14 l 7 (and elsewhere) 1-sec FAC > 1 s FAC or 1 Hz FAC
p 14 l 21 0.4 and 2% > 0.4% and 2%

**Reply**
Corrected

**Comment 3.1**

- The first paragraph is written in a bit loose way, e.g. Are FACs flowing only from boundary layers? Why FAC system is evolved only by dayside (not nightside) reconnection? FAC may exceed its nominal level – what is meant by nominal level?

**Reply to Comment 3.1**

The first para of Introduction has been rewritten in order to address the issues pointed out by Referee. The revised version reads as follows (p. 1, ll. 27-34 – p .2, ll. 1-17).

"Field-aligned currents (FACs) provide electrodynamic coupling of the solar wind-magnetosphere-ionosphere system. FACs flow along the high-conducting geomagnetic field lines between different magnetospheric domains and the high latitude ionosphere. This current system is driven by the internal magnetospheric circulation of plasma and magnetic field within the global reconnection cycle (Dangey, 1961) and by additional viscous-like interaction at the flanks of magnetosphere (Axford, 1964). Configuration of FACs is primarily controlled by the interplanetary magnetic field (IMF) orientation. Other parameters of the solar wind (velocity, density, IMF strength) and the ionospheric conductivity also play a role (e.g. Christiansen et al., 2002; Ridley 2007; Korth et al., 2002).

Schematic distribution of large-scale FACs has been established by Iijima and Potemra (1976) based on the Triad satellite data. Subsequent space missions allowed constructing comprehensive empirical models of FAC parameterized by the IMF direction and strength, by season, and by hemisphere (Weimer, 2001; Papitashvili et al., 2002; Green at al., 2009). The ionospheric projection of the 3D FAC system consists of a pair of sheets elongated along the auroral oval, namely, Region 1 (R1) and Region 2 (R2), with opposite current flow directions in the morning and evening local time sectors and additional current sheets (R0) located on the dayside poleward of R1/R2. R1 flows into the ionosphere (downward current) and from the ionosphere (upward current) on the dawn and dusk side, respectively. R1 currents, if reside on closed field lines of the Earth's magnetic field, are believed to originate in either the boundary layer or in the plasma sheet (Ganushkina et al., 2015). R2 FAC is considered to be a diversion of the partial ring current to the ionosphere driven by pressure gradients in the inner magnetosphere (Cowley, 2000). R0 current is connected to the dayside magnetopause and its polarity strongly depends on the IMF By component. On the Northern Hemisphere, the R0 current flows predominantly out of the ionosphere for positive IMF By and into the ionosphere for negative IMF By (Papitashvili et al., 2002; Lukianova et al., 2012). Additional (NBZ) current associated with the sunward ionospheric flow may appear inside the polar cap, if IMF Bz is northward (Iijima et al., 1984; Vennerstrøm et al., 2002)."

**Comment 3.2**

In the second paragraph it is said that Wang et al. (2006) and Anderson and Korth (2008) have studied storms, but no results are given.

**Reply to Comment 3.2**

A brief description of the results obtained by the authors cited has been added to the 5th para (former 2[nd]) as follows (p. 3, ll. 16-28).

"Utilizing the magnetic field measurements by CHAMP satellite Wang et al. (2006) investigated the northern and southern hemisphere dayside and nightside FAC characteristics during the

extreme October and November 2003 magnetic storms. It was shown that as Dst decreases, the FAC region expand equatorward, with the shift of FACs on the dayside controlled by the southward IMF. For both case studies, on the southern (late spring) hemisphere the minimum latitude of the FACs is limited to 50° magnetic latitude (MLat) for large negative values of Bz (The minima are the same, although in October the IMF Bz drops dawn to -28 nT, while in November it reaches -50 nT.) On the northern (late autumn) hemisphere the equatorward boundaries of the FAC region are located at 55-60° MLat. Using the global maps from the Iridium constellation Anderson et al. (2005) studied the FACs intensities during severe magnetic storms which occurred during the solar cycle 23 with a particular attention to the evolution of FACs in the course of the storm of August 2000. The results revealed the dawn–dusk asymmetry of the R1/R2 current sheets, with an increase primarily found on the duskside. It was shown that under disturbed conditions the total current is not linearly related to the interplanetary electric field, with the intensity constrained to be below 20 MA (Anderson and Korth, 2007)."

**Comment 3.3**
Please swap Sections 2 and 3, it would be more logical.

**Reply to Comment 3.3**
Done. Also, Section 2 *Swarm satellites* has been divided into two subsections: 2.1 *Instrumentation* (descriptions of the methods used for FAC derivation and the EFI instrument have been added) and 2.2 *Orbits on 6-9 September 2017*.

**Comment 3.4**
Section 2:
- Maybe the Clilverd et al. (2008) paper should be referred to?

**Reply to Comment 3.4**
Unfortunately I did not understand what paper by Clilverd et al. (2008) the Referee advises. In the 1$^{st}$ para of Section 3 Space weather conditions on 6–9 September 2017 (former Section 2) I refer to several recent papers, including Clilverd et al., 2018, in which the different effects of the September 2017 storm were analyzed.

**Comment 3.5**
Section 3:
- Here one should shortly explain how 1-s FAC data products are derived from the original magnetometer data

**Reply to Comment 3.5**
The description of FAC data products has been added to Section 2.1 *Instrumentation* (former Section 3) as follows (p. 4, ll. 21-29 – p. 5, ll. 1-15).

"The mission has a multi-instrument payload. The main module is the high-sensitivity vector (fluxgate type) and scalar magnetometers for determining the magnitude and direction of the total vector and variations of the geomagnetic field with an accuracy of more than 0.5 nT (Merayo et al., 2008). Magnetometers make it possible to carry out measurements in a wide range, including the Earth's main magnetic field and the variations of external magnetic field generated by FACs. FACs are detected by their magnetic perturbations in the orthogonal plane which are obtained after subtracting the main magnetic field model from the total measured values. From single spacecraft the FAC density can be estimated based on one magnetic component with a techniques

invoking Ampere's law under assumptions about the infinite current sheet geometry and the orthogonal crossing of the current sheet. This method was used for the previous one-satellite missions, such as Magsat and Ørsted (Christiansen et al., 2002). It is also applied to each *Swarm* satellite separately. The dual-satellite estimation method calculates current density from curl(B) measured quasi-simultaneously at 4 locations is adapted for SwA and SwC data, where measurements separated along-track are used to create a 'tetrahedron' (Ritter and Lühr, 2006). The curl(B) method provides more reliable current density estimates, as it does not require any assumptions on current geometry and orientation. The FAC output of both a dual-satellite and a single satellite methods are considered to be in a reasonable agreement (Ritter et al., 2013). However, a high degree of coherence is typical at auroral latitudes, while in the polar cap the results based ondual-spacecraft technique as more reliable (Luhr et al., 2016). Both algorithms are implemented to generate the *Swarm* products that are produced automatically by ESA's processing center as soon as all input data are available. The products are provided using the dual-satellite method on the lower pair of satellites SwA and SwC, and the single-satellite solution for each of the Swarm spacecraft individually. The 1 s values (1 Hz sampling rate) of FAC densities are available via the on-line *Swarm* data portal (ftp://swarm-diss.eo.esa.int) as Level 2 data products (Swarm Level 2 Processing System Consortium, 2012). In the present study the single-satellite FACs are used in order to apply the similar method to SwB and SwA/SwC data."

**Comment 3.6**
- Please explain what coordinate system for MLAT is used and how MLAT and MLT are derived

**Reply to Comment 3.6**
The following explanation on the coordinate system has been added (p. 5, ll. 22-27).
"In each Level 2 data file the location of the satellite is presented in an geographic coordinate system NEC (*x* - North, *y* - East, *z* - Center), where the *x* and *y* components lie in the horizontal plane, pointing northward and eastward, respectively, and *z* points to the centre of gravity of the Earth. For the purpose of present study all projections of the passes are shown in the magnetic local time (MLT) and magnetic latitude (MLat) domain. For this the coordinates are available via the on-line Swarm Data Visualisation Tool (VirES)."

**Comment 3.7**
- SwB is not separated by 1.5 h in LT from SwA and C, but this difference depends on time

**Reply to Comment 3.7**
This erroneous statement has been corrected. Now the corresponding para in Section 2.1 *Instrumentation* reads as follows (p. 4, ll. 14-19).

"SwA and SwC fly in a tandem separated by 1-1.4° in longitude and the differential delay in orbit is ~3 s. The orbit period is about 93 min and slightly different between SwA/SwC and the upper satellite SwB, so that their along-orbit separation in local time gradually changes. Their orbital planes also gradually drift apart and the separation angle increases by ~20° longitude per year. Slowly drifting in longitude, the orbits cover all the local time sectors over about 130 days."

**Comment 3.8**
- First sentence in Sect. 4.1.: give a reference

**Reply to Comment 3.8**
The references to (Weimer, 2001; Papitashvili et al., 2002; Green at al., 2009) have been added.

**Comment 3.9**
- Figure 3: Define FAC positive values (up- or downward current)

**Reply to Comment 3.9**
Definition has been added to the figure caption: "Downward (upward) current is positive (negative)".

**Comment 3.10**
Figure 4: It would be more informative for the reader to see in the upper right corner the mean MLT value (or text "pre-noon" etc) than the track identifier. One could also add standard deviations to the mean values by error bars (and expand the horizontal width of the figure)

**Reply to Comment 3.10**
Figure 4 has been re-plotted. Standard deviations, the centered MLT (instead of the track identifier) and SYM-H and AL indices have been added. Shading has been eliminated to avoid a overloading of the figure. The description of the figure has been modified accordingly. The revised version reads as follows (p. 10, ll. 6-16 - p. 12, ll. 1-15).

"To demonstrate the global temporal evolution of FACs, in Fig. 4 the current intensities for the four MLT sectors are presented separately for the northern (Fig. 4 a, c, e, g) and southern (Fig. 4 b, d, f, h) hemispheres. Each red (blue) point is determined by averaging the downward (upward) current densities when the satellite crosses the region filled with FACs. The upper (a - d) and lower (e - h) plots represent the data from the day side (10 and 16 MLT) and night side (04 and 22 MLT), respectively. For easier visual association of the evolution of FACs with the storm development, the SYM-H and AL indices are added in the plots (a, b) representing the day side and in the plots (e, f) representing the night side, respectively. During 6-9 September, FACs shown in Fig. 4, exhibit three pronounced enhancements, which are of different intensity depending on the MLT sectors. (Note that the FAC densities do not show any systematic changes associated with the orbit ocsilllation during the day.) All FACs start to increase in the very beginning of September 7 in association with the SW dynamic pressure front impinges the magnetosphere causing a positive excursion of SYM-H. The dayside FACs increase abruptly (this is especially well seen in Fig. 4 b - c, i.e. at 10 MLT, north, and at 16 MLT, south), while the nightside FACs (Fig. 4 e - h) respond to the shock with a considerable delay. The nightside FACs are peaked in the middle of September 7, when a moderate substorm occurs.

In the very beginning of September 8, in association with the first deep drops of SYM-H and AL, a step-like increase is seen at all MLTs except the prenoon sector. The peak of the day- and nightside FACs reaches 2.5 and 3.5 $\mu A/m^2$ , respectively. For a particular crossing the average density exceeds 5-6 $\mu A/m^2$ as seen from the standard deviation. The dayside FACs (Fig. 4 a - d) stay enhanced during the whole day of 8 September. The nightside FACs (Fig. 4 e - h) more closely follow the evolution of AL, so that the current intensities decrease in accordance with the first storm-time substorm recovery. The next increase in the nightside FACs occurs at ~12 UT on September 8, when the second major substorm occurs and the second drop in SYM-H is observed. On the day side the response of FACs to this substorm is marginal, although the current densities remain elevated throughout the day. All FACs fall to pre-storm levels by September 9."

**Comment 3.11**

Table 2 would need more explanation. Which MLATs are included in the calculation? What are the uncertainty limits behind these numbers? Has the author checked from the Southern hemisphere, which are the highest MLATs that the satellites reach and does that affect the estimates?

**Reply to Comment 3.11**

MLAT for 50°–90° is accounted. It is difficult to estimate the uncertainty behind the numbers presented in Table 2. The number itself has no uncertainty because this is a result of the straightforward summation. At the same time, as pointed out by the Referee, there may be indirect factors which lead to under- or overestimation of the currents. In this connection the following additional explanation for Table 2 has been included (p. 13, ll. 23-26). .

"Although the numbers in Table 2 contain uncertainties related to the lack of global observations, the estimate based on the summed FAC densities from in-situ *Swarm* measurements indicate the existence of the storm-time dawn-dusk asymmetry. Even a limited number of crossings show a clear tendency of the prevalence of the dusk-side R2."

**Comment 3.12**

- Line 29: "From the FAC values presented in columns 5 and 6 one can see that in both hemispheres the dusk side downward current is stronger than all the other currents. This predominance implies an additional amplification of the storm-time R2 FAC on the dusk side, which is related to the partial ring current." This would need more discussion and definitely a reference.

**Reply to Comment 3.12**

The following addition has been made (p. 13, ll. 27-29).

"This shift may result from a strong dusk side ion pressure leading to asymmetric dusk-side inflation of the magnetic field consistent with a partial, dusk side, ring current during storm main phase (Liemohn et al., 2001; Anderson and Korth, 2007)."

**Comment 3.13**

- "pre-storm time". It would be good to define from the beginning, what is the onset of the storm time, and maybe mark that in all the figures.

**Reply to Comment 3.13**

The pre-storm time is defined as the time before the SYM-H attains its stable negative values <20 nT at 22:00 on 7 September. In the revised version, for easier comparison of the FAC evolution with the storm phases the AL and SYM-H indices are added to Fig. 4, SYM-H is added to Fig. 5. In Figs 5 and 6 the time when the SYM-H attains its stable negative values <20 nT at 22:00 on 7 September and the time of AL minima are marked by vertical lines.

**Comment 3.14**

- l. 18 "Comparing Fig. 1 and Fig. 4 one can see that EqB more closely follows the variation of SYM-H." I agree that since end of Sept 7, the boundaries seem to follow SYM-H, but not before that. Maybe the author could check the correlation to AE-index as well?

**Reply to Comment 3.14**

I do not think that the correlation between AE and EqB would help to resolve the dependence of EqB on any single parameter. For easier comparison the SYM-H index has been added to Fig. 5. More explanation on the SYM-H and EqB coherence has been added to Section 4.3 as the follows (p. 15, ll. 1-14).

"Even visual comparison of the SYM-H and EqB evolutions in Fig. 5 reveals generally coherent behavior of these two parameters. In particular, during a period preceding the storm main phase (before 8 September, when SYM-H is mainly positive) EqB is located much lower than during the end of recovery phase (after ~12 UT on 9 September, when SYM-H is still negative). Before the SYM-H attains its stable negative values <-20 nT at 22:00 on 7 September FACs are observed mainly poleward of 60° MLat on both hemispheres. Moderate equatorward shifts of EqB are associated with the modest substorms occurred before the storm main phase in the middle of 6 and 7 September. Prior the main phase, on both hemispheres the prenoon (04 MLT) EqB is found considerably poleward compared to the EqB location at other MLTs. The effect is well seen during the two time intervals: from ~22 UT, September 6 till 06 UT, September 7 and at 12-24 UT, September 7. Both intervals are dominated by the northward IMF (sf. Fig. 2), so that a shrinking of the polar cap and a poleward shift of the auroral oval is expected. With regard to the position of FACs, the displacement of its equatorward boundary is the largest only in the pre-noon sector, while the other local times remain less affected."

**Comment 3.15**
"The current intensity vary inversely with scale". Please give a reference.

**Reply to Comment 3.15**
The references to (Neubert and Christiansen, 2003; McGranaghan et al., 2017) have been added.

**Comment 3.16**
It is unclear how Figure 7 is composed. What are the horizontal and vertical axes? Is the figure even needed in this paper?

**Reply to Comment 3.16**

Fig. 7 seems curious because it demonstrates the bipolar structure (closely adjacent small-scale FACs of opposite polarity) occurrence in different MLTs. On the day- and night side they are likely associated with the reconnection formed at the magnetopause and mesoscale auroral arcs, respectively. To make it more clear how the figure is composed the title of x and y axis has been modified as MLat "down" and MLat "up", respectively.

The following explanation for Fig. 7 has been added to Section 4.4 (p. 18, ll. 7-17 – p. 19, ll. 1-3) and 5.2 (p. 27, ll. 15-28).

"When for each crossing within a certain MLT sector, the minimum (i.e. peak upward current) and the maximum (i.e. peak downward current) 1 s FACs are selected, it appears that in some cases these peaks are observed at very close latitudes, while in other cases the minimum and maximum are spaced in latitude. In Fig. 7, the correlations between the MLats, at which the most intense small-scale FACs of opposite polarities are observed, are presented for each MLT sectors. The x-axis (y-axis) corresponds to the MLat of the downward (upward) peak selected in each crossing. The magnitude of minima and maxima are not accounted. From Fig. 7 one can see that correlation between the latitudinal positions of the up- and downward peaks varies with MLT. The highest correlation coefficient (cc=0.94) is found in the pre-noon sector (Fig. 7b).

This is indicative of a large population of the paired, closely adjacent small-scale currents of opposite polarity (called hereafter the bipolar structure). In the dusk (Fig. 7a) the correlation coefficient decreases down to 0.78. Almost the same correlation (cc=0.75) is observed in the pre-midnight sector (Fig. 7c). At the early morning hours (Fig. 7d) the correlation is much weaker (cc=0.53) implying that the extreme up- and downward currents appear less frequently in pair but rather are spatially (or temporary) separated. Different mechanisms of the small-scale FAC formation on the day- and night side can be the cause of this spatial distribution and variability."

"Statistically, the bipolar structures dominate in the pre-noon. In the post-midnight MLTs they are observed less frequently. While the interpretation of the bipolar structure in the terms of the meso-scale arc pattern seems reasonable, the small-scale FACs are often a result of reconnection processes distributed over the dayside magnetopause and even in the tail for negative Bz. In contrast to the post-midnight, in the pre-noon sector, where cusp/cleft currents are expected, the bipolar structures are quite frequent. This may be a signature of the plasma injections which are accompanied by pairs of FACs generated due to flux transfer event (FTE) formation (Southwood, 1987) or multiple reconnection at the magnetopause. Magnetic topologies associated with FTEs were previously observed by the MEO satellites (Marchaudon et al., 2004; 2006; Pu et al., 2013). The small-scale field-aligned currents are possibly a consequence of turbulence and instabilities associated with the process of opening previously closed magnetospheric field lines and merging them with the interplanetary magnetic field (Watermann et al., 2009). The regularity presented in **Fig. 7** shows that during the September 2017 magnetic storm the bipolar structures dominate exactly in the region where the signatures of FTEs and the reconnection lines formed at the magnetopause are expected. At the same time, a pair of the most intense FACs is observed on the night side."

**Comment 3.17**
- In this section suddenly Te and Usc are discussed without anywhere properly explained, how it has been derived (which instrument, references etc)

**Reply to Comment 3.17**
A brief description of the plasma instrument has been added to Section 2.1 *Instrumentation* (p. 5, ll. 17-22).

**Comment 3.18**
- "The considerably elevated Te within the arc and just poleward of the arc is associated with a local amplification of electric field." I don't understand this sentence. To my understanding, electric field data is not used in this study. Furthermore, why Te enhancement would be associated with enhanced EF?

**Reply to Comment 3.18**
Yes, the electric field data is not used in this study because the Swarm electric field is unavailable. Thus I can only referee to previous observations, e.g. by Aikio et al. (2002) and Kozlovsky et al. (2007). And indeed, it is not necessary the Te enhancement would be associated with enhanced EF.

**Comment 3.19**
Reference to Wang et al. (2004) is not found from the list. maybe it should be 2006?

**Reply to Comment 3.19**
Wang et al. (2006) is correct.

[revised manuscript text omitted]
 and the dayside density, which is enhanced throughout the storm, temporal variations of EqB are relatively smooth and tend to follow the SYM-H ndex. In the end of September 9, during the late recovery phase of the storm, EqB is shifted poleward as high as 70° MLat. As far as the day-night asymmetry is concerned, almost no difference in evolution on the day- and nightside EqBs is observed during the main and recovery phases. Because the storm-time substorms are relatively

short-lived, an expansion of the FAC region during the growth phase, and then a contraction are difficult to resolve with the *Swarm* data.

[Figure]

5 **Figure 5**: Magnetic latitude of the FAC equatorward boundaries on the Northern (top) and Southern (bottom) hemispheres for the sectors centered at around 04, 10, 16 and 22 MLT (MLat for each individual MLT sector is shown by dots of different colors). The SYM-H index (black line) is superimposed on MLat. Three vertical solid lines mark successively 
[revised manuscript text omitted]

---

## Referee Report (RR1)

**Review report for "Swarm Field-aligned Currents During a Severe Magnetic Storm of September 2017"**

**General comment:**

In this paper, the author conducted comprehensive investigations on the evolution of the fieldaligned currents (FACs) at different scales during a recent intense geomagnetic storm by using the Swarm level-2 FAC products. However, some conclusions obtained in this study are not well supported by figures presented in the paper and some conclusions are degraded by the data quality, data coverage and methodology used in this study. In addition, some conclusions do not convey any new ideas. Therefore, this paper may have not reach substantial conclusions that suitable for publication. Meanwhile, conceptual and grammatical mistakes are frequently shown in the manuscript. A major revision is needed before the next submission.

**Major Comments:**

**Comments for the Conclusion #1:**

1) "The FACs become enhanced starting from the SW shock arrival despite of the prolonged period of the northward IMF. The night-time FAC densities primarily follow the substorm development while the dayside FACs are intensified in response to the SW shock and then stay enhanced. At the peak of substorm, the FAC densities averaged over a track within a given MLT sector, reach 3  $\mu$ A/m2, while the undisturbed level is about 0.2  $\mu$ A/m2."

- a) It seems that this sentence is concluded from Figure 4, in which the evolutions of the average upward/downward FACs at four different MLT sectors and at both hemisphere are shown. Indeed, the average values increase after the significant drop of the SYM-H. However, the standard deviations are extremely large in comparison with the averages. If the standard deviations are taken into account, one can say that the FACs do not necessarily increase during the storm.
- b) Meanwhile, Figure 4 does not evidently indicate that "the night-time FAC densities primarily follow the substorm development", since the FAC intensity increases when both SYM-H and AL indices decrease and the FAC intensity decreases when both SYM-H and AL indices increase on the night side. Therefore, the night-side FAC evolutions may be modulated by both the geomagnetic storm and substorm. The data shown in

Figure 4 cannot rule out the important role that the geomagnetic storm plays in the modulation of the evolution of the night-time FAC.

2) "The dawn–dusk asymmetry is manifested on the enhanced dusk side downward (R2) FAC on both hemispheres."

- a) Although Table 2 shows the responses of FACs in certain MLT sectors on the dawn side are different those in certain MLT sectors on the dusk side, it cannot be concluded as "dawn-dusk asymmetry" since the results based on Table 2 are MLT biased. Perhaps it might be different in other MLT sectors. To better study the dawn-dusk asymmetry, data with better MLT coverage, such as AMPERE data, are useful. Without using data with reasonable MLT coverage, the statement associated with the dawn-dusk asymmetry may be problematic and needed to be removed.
- b) It seems that the results in Table 2 are calculated by using 1 Hz FAC data. If so, the upward/downward FACs do not necessary mean R2/R1 (R1/R2) FACs on dawn (dusk) side. Typically, R1/R2 FACs represent large-scale FACs.

**Comments for the Conclusion #2:**

"The equatorward displacement of FAC sheets (in the north and south and at all MLTs) correlates with the storm intensity as monitored by the SYM-H index. The minimum latitude of the equatorial FAC boundaries is limited to 50° MLat. Displacement of FAC sheets is more gradual and occurs with a considerable time delay compared to the changes in current intensity."

a) The first sentence is not an new idea since it has been well studied in previous studies. For example, Wang et al. (2006) stated that "*The response of the equatorward FACs is found to roughly correlate with the IMF*  $B_z$ , *Dst*  $E_m$  and  $\varepsilon$ ". Since the SYM-H index is the high-resolution version of Dst index, the first sentence does not bring anything new to the community. In addition, since only four MLT sectors have been studied in each hemispheres and they do not cover all MLTs, the content in the parenthesis is not precise enough and may need to be removed.

- b) The second sentence also brings nothing new and is not precise enough. For example, after 12 UT on September 8, the equatorward boundary reached <50° MLAT, therefore the statement that "the equatorial FAC boundaries is limited to 50° MLAT" is not precise. In addition, Fujii et al. (1992) stated that "*The equatorward boundary of the FAC system reached as low as 48° MLAT*" although a different storm was studied in their paper. But no new message has been conveyed by the second sentence.
- c) For the third sentence:
  - a. What do you mean by time delay? Delay with what? Did you show it in any figure and provide any quantitative description in the context?
  - b. To study the displacement of the equatorward boundary of FAC you have utilized the 21-s averaged FAC, but to generate Figure 4, I suspect that you have utilized 1-s original data given the very large error bars, so you may not compare the same thing. If you want to substantiate your statement, you may need to use FACs on the same scale (e.g., 150 km or larger scales)

**Comments for the Conclusion #3:**

"The filamentary structures of high-density FACs are always presented in the Swarm observations. A bipolar structure (i.e. the adjacent upward and downward small-scale FACs),  $\sim 80 \ \mu A/m^2$ , 7.5 km width, is observed in the vicinity of the newly developed westward electrojet just prior the substorm onset. Simultaneous plasma perturbations indicate that the FAC pattern is likely associated with mesoscale auroral arc."

- a) Although high-frequency FAC data can be used, cautions are needed when using the high-frequency FAC data. Because the assumptions used to derive the single-satellite FAC data may break down at small scales. Did you apply any data quality control technique for your small-scale FAC data? How? Since you have focused on those very isolated structures, the reliability of data is extremely crucial. Otherwise, your results may be degraded by using unreliable data.
- b) The connection between the "bipolar structure" and "enhancement of the electron density" is not obvious. After a careful inspection, it seems that the strong upward portion

of the bipolar structure actually corresponds to the depletion of the electron density (Figure 9), and does not correspond to the enhancement of the electron density.

**Minor Comments:**

**1) Abstract:**

- a. Page 1, Line 8: Evolution  $\rightarrow$  Evolutions
- b. Page 1, Line 15: "a substantial fraction of R1/R2 FACs is composed of many small-scale currents": May need to be altered, since theR1/R2 FACs are referred to the large-scale currents, which are not necessarily related to the small-scale currents.

**2) Introduction:**

- a. Page 1, Line 29: high latitude  $\rightarrow$  high-latitude
- b. Page 1, Lines 32: Please add some references to support the statement. Also add "the" at the beginning.
- c. Page 2, Line 5: Since the connections between the auroral oval and FACs are still unclear, perhaps you can simplify the sentence to "The large-scale FAC consists of Region 1(R1) and Region 2 (R2) currents …"
- d. Page 2, Line 8: Add "currents/FACs" after "R1/R2" and "R1" and keep it consistent below.
- e. Page 2, Line 19: Please define the spatial scale sizes of "large scale" and "small scale".
- f. Page 2, Line 30: Add some references to support your statement.
- g. Page 3, Line 4: "counterpart"  $\rightarrow$  "counterparts"
- h. Page 3, Line 11: Rephrase the sentence starting at Line 11.
- Page 3, Line 13: "compared to" → "as compared to the"; What do you mean by "stationary"?
- j. Page 3, Line 14: "extreme values are often reached" is not precise.
- k. Page 3, Line 15: "focus"  $\rightarrow$  "have focused"
- 1. Page 3, Line 16: Please add "For example," before "Utilizing"
- m. Page 3, Lines 25~28: Please rephrase the corresponding statements.
- 3) Section 2:
  - a. Page 4, Line 11: "orbit"  $\rightarrow$  "orbits"

- b. Page 4, Line 15: What is the speed of the Swarm satellites?
- c. Page 4, Third paragraph: Please simplify this paragraph and only provide the most important information related to the FAC data used in this study.
- d. Page 6: Where is the Figure 1b? Also please add one plot showing the orbital coverage at southern hemisphere.

**4) Section 3:**

- a. Please check the verb tense (Also in other sections).
- b. Page 7, Line 16: Add the UT to indicate when the IMF Bz turned northward.

**5) Section 4.1:**

- a. Page 9, Lines 23-25: R1/R2 currents typically represent large-scale (e.g., >500 km) FACs. And a 21-point moving window (~150 km) not only captures the large-scale currents but also captures some mesoscale FACs. Thus, the smoothed FAC in Figure 3b has more structures than typical R1/R2 current scheme. If you try to associated the downward/upward currents with R1/R2 currents, a larger moving window (e.g., ~500-km width) is needed. Otherwise, the corresponding discussion does not make too much sense and can be removed.
- b. As mentioned in the major comment, the results shown in Figure 4 are significantly degraded by the large error bars. I think that you may have utilized the original 1-Hz data to calculate the results shown in Figure 4. If that is the case, the large error bars may be related to the very intense small-scale FACs as shown in Figure 3a. From you Section 5.1, it seems that you want to investigate the evolution of the large-scale FAC. If so, you need to use the smoothed FAC data rather the original data to conduct the study, which may give you smaller error bars and improve the results. Did you do in this way? If so, you need to mention it in the context. Otherwise, you can also calculate the standard errors, which are the standard deviation divided by the square root of samples, and use them as the errors bars.
- c. Page 11, second paragraph: The relative importance of substorm and geomagnetic storm in controlling the nightside FAC evolutions cannot be directly distinguished

according to Figure 4. It might be straightforward to show the correlation between FAC and SYM-H/AL to indicate which one plays a more important role.

d. Page 11, last paragraph: The last two sentence can be removed since they are not directly related to figures shown in the paper.

**6) Section 4.2:**

a. Please see the second item of the comment for Conclusion #1.

**7) Section 4.3:**

- a. Page 14, Line 12: Please Add some references after "equatorward"
- b. Page 14, Line 15: "20-point"? You mentioned "21-point" in Line 19 on Page 10.
  Please keep it consistent.
- c. Page 15, Line 6: After 22 UT on September 7, the SYM-H was not stable.
- d. Page 15, Line 9: 04 MLT is probably too prenoon. 04 MLT  $\rightarrow$  10 MLT?
- e. Page 15, Lines 22-24: Please see the third item of the comment for Conclusion #2

**8) Section 4.4:**

- a. Figure 6: Perhaps you could use the shade to highlight the period when SYM-H < -20 nT;</li>
- b. Page 20, Line 1: From the IL index, it is difficult to tell that the substorm was in the growth phase when the bipolar FAC was identified since it seems that the IL index was relatively stable at the time when the bipolar FAC was identified.
- c. Page 20: Line 12: The difference of 15 μA is not trivial in comparison with your peak FACs (20%~25%), so that the downward and upward currents are not comparable. So this whole sentence may need to be removed.
- d. Page 20, Line 13~14: First, FACs at 150-km scale size may not well represent large-scale R1/R2 FACs. Second, From Figure 9a, the bipolar structure is located in the downward FAC rather than between the "large-scale" downward and upward FAC.

**9) Section 5.1:**

- a. First paragraph: I think the content is related to Figure 4, where your results may not really represent the evolution of large-scale FAC, especially you haven't pointed out whether the original data or the smoothed data have been used. Given the large error bars you have presented, it seems that the original data have been used, which are mixtures of FACs on different scales.
- b. Last paragraph: See comments for Section 4.2

**10) Section 5.2:**

a. See the last general comments

**References:**

Fujii, R., Fukunishi, H., Kokubun, S., Sugiura, M., Tohyama, F., Hayakawa, H., Tsyryda, K., and Okada, T.: Field-aligned currents signatures during the March 13-14, 1989 great magnetic storm, J. Geophys. Res., 97, 10 703–10 715, 1992.

H. Wang, H. Lühr, S. Y. Ma, J. Weygand, R. M. Skoug, et al.. Field-aligned currents observed by CHAMP during the intense 2003 geomagnetic storm events. Annales Geophysicae, European Geosciences Union, 2006, 24 (1), pp.311-324.

---

## Referee Report (RR2)

**Review report for Lukianova (2019)**

The author has done substantial improvements of the manuscript through the revision, in particular their conclusion 2. However, the content related to the Conclusion 1 still needs to be improved. Therefore, a minor revision is suggested.

**Detailed comments:**

**Section 4.1**

First paragraph: Perhaps only need to introduce the distribution of tracks and not explain their relationships with the currents and electrojet.

Line 15: What do you mean by empty? Is it due to the range of your y axis (Not supported by your Figure 3b). Perhaps you could remove the expression "while …"

Line 17: Please add the length of the sliding window in km after "51-point"

Figure 4: Perhaps you can use your error bar to represent the standard error (standard deviation divided by the square root of the number) instead of the standard deviation, so that your averages are more meaningful.

The expression "the night-time FAC densities primarily follow the substorm development": Perhaps the evolvements of the nightside FAC are combinations of the modulations related the geomagnetic storm and substorm. The relative importance cannot be simply established only from the similarity of the evolvements of the FAC and AL index. In fact, one can argue that the similarity between the evolvements of FAC and SYM-H are not that bad. Perhaps you need to mention the contributions from both geomagnetic storm and substorm.

**Section 4.2**

Perhaps it is better not to include the content relevant to the "dawn-dusk asymmetry" in the manuscript. First, as mentioned in the last comment, the Swarm data are MLT-biased, and credit of the related content will be significantly degraded. Second, your methodology is questionable. From Table 2, it seems that you just simply added the all upward/downward FAC in a given sector. In fact, you need to take into account the area. Since the area of a latitudinal ring decreases as the latitude increase, so that the measurements at low latitude have larger weights than those obtained at higher latitude.

**Section 4.4:**

What are your definitions of mesoscale and small scale?

**Conclusion:**

Second paragraph, last sentence: Please add the standard deviations or standard errors after the number 3 and 0.2.

---

## Author Response (AR2)

**Response to the Referee' comments**

I thank the reviewers for valuable comments and constructive critique. All comments were carefully considered and addressed. Answers to all the questions are presented below. Corresponding changes have been made in the revised manuscript (marked in cyan). New Figure 6 has been added. Comments and replies are shown in smaller and larger letters, respectively.

- - -

**1#Referee**

This paper considers the evolution of the large-scale FAC morphology during the magnetic storm of September 2017, and relationship between large-scale and small-scale FACs, using observations from Swarm. The author has done a satisfactory job of addressing comments from the previous reviewers, and the manuscript is more focussed and readable as a consequence. I consider the paper to be suitable for publication, subject to the minor comments below.

**Response to the 1#Referee comments**

1) Clausen et al. (2013) not included in the reference list. The reference list is not in alphabetical order.

**Included. The reference list has been ordered.**

2) Section 1: A key aspect of the paper is the latitude of the FACs as controlled by magnetic reconnection at the magnetopause and in the magnetotail. Although the Dungey cycle is mentioned, it might be appropriate to include a reference to the time-dependent expanding/contracting polar cap model, e.g. Cowley and Lockwood (Ann. Geophys., 1992).

**The recommended reference has been included.**

3) p 10, l 15: oscillation misspelt

**Corrected**

4) Section 4.1 and Fig. 4: There are subtle differences in the FAC variations in the NH and SH which do not seem to be commented on in the paper. For instance, the NH FACs seem to be marginally less variable than the SH FACs. The period in question is close to equinox, so ionospheric conductance effects are not expected to play a role. Does the author have any suggestion why this interhemispheric asymmetry manifests itself?

In my view, based on the FACs presented in Fig. 4 it is difficult to make an unambiguous conclusion on the interhemispheric difference in the FAC variations. The greater variability in the SH FACs compared to the NH FACs is not sufficiently evident neither from the average current densities nor from the error bars. Quantitatively (see the figure below), for all local times the errors summed over the NH and SH, are of the same value (the upper plot). There is a systematic difference between the upper and lower

satellites (the lower plot) in such a way that the SwB exhibits slightly larger variability.

5) Section 4.3: There has been some previous work on the latitude of the auroras (and hence FACs) in relation to ring current intensity, e.g. Milan et al. (Ann. Geophys., 2009) and Milan (GRL, 2009), showing that the oval expands to low latitudes during storms, and even for moderate Sym-H levels is expanded lower than quiet times. It would be appropriate to cite these papers, and to discuss in relation to the current observations.

The following additions have been made (p. 25, ll. 8-16):

"High FAC intensity is associated with the auroral oval. Previous studies based on particle precipitation and optical observations have shown that the oval radius increases when the ring current is intensified during magnetic storms (e.g., Meng, 1982; Yokoyama et al., 1998). Significant variations in the location of the aurora take place during the substorm cycle. Substorms occurring on expanded auroral ovals during magnetic storms are most intense, since they close the most magnetospheric open magnetic flux and the presence of the enhanced ring current increases the open flux threshold

at which substorm onset is favoured (Milan, 2009). It was also shown that changes in oval radius associated with dayside and substorm driving occur on timescales of minutes and hours, while changes associated with the ring current are more protracted as the ring current dissipates slowly (Milan et al., 2009)."

Yokoyama, N., Kamide, Y., and Miyaoka, H.: The size of the auroral belt during magnetic storms, Ann. Geophys., 16, 566–573, 1998, www.ann-geophys.net/16/566/1998/.

Meng, C.-I.: Dynamic variation of the auroral oval during intense magnetic storms, J. Geophys. Res., 89, 227–235, 1984.

Milan, S. E.: Both solar wind-magnetosphere coupling and ring current intensity control of the size of the auroral oval, Geophys. Res. Lett., 36, L18101, doi:10.1029/2009GL039997, 2009.

Milan, S.E., Hutchinson, J., Boakes, P.D., and Hubert, B.:Influences on the radius of the auroral oval Ann. Geophys., 27, 2913–2924, www.ann-geophys.net/27/2913/2009/, 2009.

**2#Referee**

General comment: In this paper, the author conducted comprehensive investigations on the evolution of the fieldaligned currents (FACs) at different scales during a recent intense geomagnetic storm by using the Swarm level-2 FAC products. However, some conclusions obtained in this study are not well supported by figures presented in the paper and some conclusions are degraded by the data quality, data coverage and methodology used in this study. In addition, some conclusions do not convey any new ideas. Therefore, this paper may have not reach substantial conclusions that suitable for publication. Meanwhile, conceptual and grammatical mistakes are frequently shown in the manuscript. A major revision is needed before the next submission.

**Response to the 2#Referee comments**

Major Comments:

Comments for the Conclusion #1:

1) "The FACs become enhanced starting from the SW shock arrival despite of the prolonged period of the northward IMF. The night-time FAC densities primarily follow the substorm development while the dayside FACs are intensified in response to the SW shock and then stay enhanced. At the peak of substorm, the FAC densities averaged over a track within a given MLT sector, reach 3  $\mu$ A/m2, while the undisturbed level is about 0.2  $\mu$ A/m2."

a) It seems that this sentence is concluded from Figure 4, in which the evolutions of the average upward/downward FACs at four different MLT sectors and at both hemispheres are shown. Indeed, the average values increase after the significant drop of the SYM-H. However, the standard deviations are extremely large in comparison with the averages. If the standard deviations are taken into account, one can say that the FACs do not necessarily increase during the storm.

Yes, this conclusion is based on Fig. 4, which does show the storm time increase of FACs. The increase of standard deviations indicates as the larger variability and as the larger magnitude of the 1 s values. In Fig. 4 each red (blue) point is determined by averaging the 1-s downward (upward) current densities, when the satellite crosses the region filled with FACs (about 500 1-second measurements per crossing). Thus the increase of the error bar indicates that a satellite flying over the particular FAC region measures (at least during one second) the large-amplitude FAC. The smaller error bar indicates that the 1-Hz FACs are approximately of the same amplitude.

b) Meanwhile, Figure 4 does not evidently indicate that "the night-time FAC densities primarily follow the substorm development", since the FAC intensity increases when both SYM-H and AL indices decrease and the FAC intensity decreases when both SYMH and AL indices increase on the night side. Therefore, the night-side FAC evolutions may be modulated by both the geomagnetic storm and substorm. The data shown in Figure 4 cannot rule out the important role that the geomagnetic storm plays in the modulation of the evolution of the night-time FAC.

Yes, the night-side FAC evolutions are modulated by both the storm and substorm. The conclusion on "the night-time FAC densities primarily follow the substorm development" is based on the comparison between the dayside and nightside FACs. Even visual examination of Figure 4 shows that the

dayside FACs are much less compared with the evolution of AL index and less affected by substorms. The conclusion #1 stays unchanged.

2) "The dawn–dusk asymmetry is manifested on the enhanced dusk side downward (R2) FAC on both hemispheres."

a) Although Table 2 shows the responses of FACs in certain MLT sectors on the dawn side are different those in certain MLT sectors on the dusk side, it cannot be concluded as "dawn-dusk asymmetry" since the results based on Table 2 are MLT biased. Perhaps it might be different in other MLT sectors. To better study the dawn-dusk asymmetry, data with better MLT coverage, such as AMPERE data, are useful. Without using data with reasonable MLT coverage, the statement associated with the dawn-dusk asymmetry may be problematic and needed to be removed.

As mentioned in the last para of section 5.1, the AMPERE data provide more global and reliable estimate. However, it is curious that an indication of the dawn-dusk asymmetry can be inferred even from the instantaneous observations made by Swarm. However, because the estimate based on the Swarm data is approximate and indeed may suffer of the MLT bias, the statement on the dawn-dusk asymmetry has been removed from the Conclusion 2.

b) It seems that the results in Table 2 are calculated by using 1 Hz FAC data. If so, the upward/downward FACs do not necessary mean R2/R1 (R1/R2) FACs on dawn (dusk) side. Typically, R1/R2 FACs represent large-scale FACs.

Yes, the results in Table 2 are calculated by using 1 Hz FAC data and their averages are not necessary a representation of the large-scale R2/R1 FACs. The corresponding comment has been added to the last para of Section 5.1 ("Although the *Swarm* observations unable to provide the instantaneous global FAC distribution, the responses of FACs in certain MLT sectors on the dawn side are different from those on the dusk side. Note that the results in Table 2 are calculated by using the 1 Hz FAC values and their averages do not necessary represent the large-scale R1/R2 FACs. Nevertheless, for the storm of September 2017, the dawn-dusk asymmetry is manifested in the enhanced average density of the downward FACs on the dusk side. This feature is consistent with the global observations by AMPERE, from which the asymmetry of large-scale FACs can be identified.").

Comments for the Conclusion #2:

"The equatorward displacement of FAC sheets (in the north and south and at all MLTs) correlates with the storm intensity as monitored by the SYM-H index. The minimum latitude of the equatorial FAC boundaries is limited to 50° MLat. Displacement of FAC sheets is more gradual and occurs with a considerable time delay compared to the changes in current intensity."

a) The first sentence is not a new idea since it has been well studied in previous studies. For example, Wang et al. (2006) stated that "The response of the equatorward FACs is found to roughly correlate with the IMF Bz, Dst Em and  $\varepsilon$ ". Since the SYM-H index is the high-resolution version of Dst index, the first sentence does not bring anything new to the community. In addition, since only four MLT sectors have

been studied in each hemispheres and they do not cover all MLTs, the content in the parenthesis is not precise enough and may need to be removed.

The Conclusion #2 has been modified in order to emphasize the role of substorms in the SYMH-EqB relationship. A new figure illustrating this relationship has been added. The correlation coefficients for the main and recovery phases are very similar (cc=0.88 and 0.87), while the corresponding regression equations are considerably different. During the storm main phase, the equatorward expansion of EqB is described by the equation MLat=63.1+0.1·SYMH, while during the recovery phase the poleward shift of EqB is described by the expression MLat=79.5+0.3·SYMH. The fast recovery of EqB is mainly due to the fast decrease in substorm activity on September 9. This result is not similar to what was found by Wang et al. (2006), because these authors did not consider the role of substorm activity.

The words "at all MLTs" have been removed to avoid ambiguity. Note, however, that in the previous version "all MLTs" implied the MLTs covered by the orbits.

b) The second sentence also brings nothing new and is not precise enough. For example, after 12 UT on September 8, the equatorward boundary reached  $<50^{\circ}$  MLAT, therefore the statement that "the equatorial FAC boundaries is limited to 50° MLAT" is not precise. In addition, Fujii et al. (1992) stated that "The equatorward boundary of the FAC system reached as low as 48° MLAT" although a different storm was studied in their paper. But no new message has been conveyed by the second sentence.

Here, a notable feature is that for the September 2017 storm the Dst was about -100 nT, while for the events studied previously the Dst was much lower (-400/-600 nT). Despite the large difference in Dst, for all storms the minimum of the equatorward boundary is found at approximately the same latitude, not lower than 48 - 50° MLat. For clarification, the second sentence of conclusion #2 has been replaced by the following: "The correlation coefficients for the main and recovery phases are about 0.9, while in the course of the main phase the rate of equatorward expansion of FACs is slower than their poleward displacement during the recovery phase. This is likely due to the relatively fast decrease in substorm activity. The minimum latitude of the equatorward FAC boundaries is limited to 49-50° MLat. Although the storm of September 2017 is relatively weak (Dst is about -100 nT), the FAC region expands approximately to the same latitudes as those observed for the much severe storms."

c) For the third sentence: a. What do you mean by time delay? Delay with what? Did you show it in any figure and provide any quantitative description in the context?

Because, indeed, no quantitative estimate of time delay is presented, the sentence has been eliminated.

d) To study the displacement of the equatorward boundary of FAC you have utilized the 21-s averaged FAC, but to generate Figure 4, I suspect that you have utilized 1-s original data given the very large error bars, so you may not compare the same thing. If you want to substantiate your statement, you may need to use FACs on the same scale (e.g., 150 km or larger scales)

The 1-s data, without any averaging, were used to generate Figure 4 and Figure 5. To determine the lowest MLat at which FACs were terminated, the 20-point sliding window (but not the 20-point averaging) were applied to the 1 s FAC values in order.

Comments for the Conclusion #3:

"The filamentary structures of high-density FACs are always presented in the Swarm observations. A bipolar structure (i.e. the adjacent upward and downward small-scale FACs), ~80  $\mu$ A/m2, 7.5 km width, is observed in the vicinity of the newly developed westward electrojet just prior the substorm onset. Simultaneous plasma perturbations indicate that the FAC pattern is likely associated with mesoscale auroral arc."

a) Although high-frequency FAC data can be used, cautions are needed when using the high-frequency FAC data. Because the assumptions used to derive the single-satellite FAC data may break down at small scales. Did you apply any data quality control technique for your small-scale FAC data? How? Since you have focused on those very isolated structures, the reliability of data is extremely crucial. Otherwise, your results may be degraded by using unreliable data.

No special data quality control technique for the small-scale FAC data has been applied. The original FAC data from the Swarm data base, as it is, were used. However, the magnetic East and North components were checked. The B-E (and the B-N, to the less degree) shows considerable perturbations which can be interpreted as a signature of FACs. In addition, all storm times during the Swarm operational period were checked. During each storm, the high-amplitude 1-second FACs, similar to those shown in Fig. 10, are presented. During the non-storm periods, no such peaks are observed.

b) The connection between the "bipolar structure" and "enhancement of the electron density" is not obvious. After a careful inspection, it seems that the strong upward portion of the bipolar structure actually corresponds to the depletion of the electron density (Figure 9), and does not correspond to the enhancement of the electron density.

A higher frequency (> 1 Hz) is desirable to determine unambiguously the small-scale FACs. The figure below explains how the 1-second time shift between FACs and the depletion of Ne may originate. The magnetic eastward component shows a positive spike at 00:10:18 (upper plot), from which the automatic procedure calculates the consecutive upward and downward FAC with the time stamp of 00:10:18 and 00:10:19, respectively (middle plot). In this case the current density is calculated as FAC(i)=(B(i+1)-B(i))/dx. (1)

The depletion of Ne has the time stamp of 00:10:19 and thus, indeed, it turns out that the upward portion of the bipolar structure coincides with a drop in Ne (lower plot). However, the FAC can be also calculated as FAC(i+1)=(B(i+1)-B(i))/dx. (2)

Formulas (1) and (2) are equally correct. In case (2) the time stamps for the upward and downward currents are 00:10:19 and 00:10:20, and the downward FAC that corresponds to the Ne depletion.

---

## Author Response (AR3)

**Reply to the Referee's minor comments**

Comments and replies are given below. Corresponding changes have been made in the revised manuscript (marked in cyan).

1) Section 4.1
   First paragraph: Perhaps only need to introduce the distribution of tracks and not explain their relationships with the currents and electrojet.

Reply:

The sentence explaining the relationship of the observed FACs with the R1/R2 currents and electrojets have been removed from the first para of Section 4.1.

2) Line 15: What do you mean by empty? Is it due to the range of your y axis (Not supported by your Figure 3b). Perhaps you could remove the expression "while …"

Reply: Removed from the first para of Section 4.1.

3) Line 17: Please add the length of the sliding window in km after "51-point"

Reply: The corresponding addition has been made (p.9, l. 14-15)

4) Figure 4: Perhaps you can use your error bar to represent the standard error (standard deviation divided by the square root of the number) instead of the standard deviation, so that your averages are more meaningful.

Reply: For each crossing of the FAC region, the corresponding data set consists of about 500 (1 Hz) values. The standard errors have been calculated but Fig. 4 has left unmodified. The magnitude of SE is mentioned in Section 4.1 (p.11, l. 7) and in Conclusion.

5) The expression "the night-time FAC densities primarily follow the substorm development":
   Perhaps the evolvements of the nightside FAC are combinations of the modulations related the geomagnetic storm and substorm. The relative importance cannot be simply established only from the similarity of the evolvements of the FAC and AL index. In fact, one can argue that the similarity between the evolvements of FAC and SYM-H are not that bad. Perhaps you need to mention the contributions from both geomagnetic storm and substorm.

Reply: The sentence "the night-time FAC densities primarily follow the substorm development" has been modified. Now it reads as "The evolvements of the nightside FACs are combinations of the modulations related the geomagnetic storm and substorm. Their densities are more responsive to the substorm development, while the dayside FACs are intensified in response to the SW shock and then stay enhanced." (p. 27, l. 21-23)."

6) Section 4.2

Perhaps it is better not to include the content relevant to the "dawn-dusk asymmetry" in the manuscript. First, as mentioned in the last comment, the Swarm data are MLT-biased, and credit of the related content will be significantly degraded. Second, your methodology is questionable. From Table 2, it seems that you just simply added the all upward/downward FAC in a given sector. In fact, you need to take into account the area. Since the area of a latitudinal ring decreases as the latitude increase, so that the measurements at low latitude have larger weights than those obtained at higher latitude.

Reply: Section 4.2 has been removed.

7) Section 4.4:

What are your definitions of mesoscale and small scale?

Reply: As for the FAC, the small scale corresponds to a spatial resolution of ~ 7.5 km (the 1 Hz data available for Swarm). (p. 4, l. 10-11).

The description of the upper plot of Figure 10 has been modified to avoid the word "mesoscale" <FAC>. (p. 19, l.11-12). Instead, the terminology of R1/R2 is used.

The auroral arc is called mesoscale because the typical width of the mesoscale arc is 8-20 km, while the small-scale arc is of order 1 km (e.g. Knudsen et al., 2001, 10.1029/2000GL011969).

8) Conclusion:

Second paragraph, last sentence: Please add the standard deviations or standard errors after the number 3 and 0.2.

Reply: SE is added (p.27, l. 25)

[revised manuscript text omitted]